# Tissue-resident memory CD8 T cell diversity is spatiotemporally imprinted

Miguel Reina-Campos[1,2,7], Alexander Monell[1,3,7], Amir Ferry[1], Vida Luna[1], Kitty P. Cheung[1], Giovanni Galletti[1], Nicole E. Scharping[1], Kennidy K. Takehara[1], Sara Quon[1], Peter P. Challita[1], Brigid Boland[4], Yun Hsuan Lin[4], William H. Wong[4], Cynthia S. Indralingam[4], Hayley Neadeau[2], Suzie Alarcón[2], Gene W. Yeo[3], John T. Chang[4,5], Maximilian Heeg[1,6,7 ✉] & Ananda W. Goldrath[1,6 ✉]

Tissue-resident memory CD8 T ($T_{RM}$) cells provide protection from infection at barrier sites. In the small intestine, $T_{RM}$ cells are found in at least two distinct subpopulations: one with higher expression of effector molecules and another with greater memory potential[1]. However, the origins of this diversity remain unknown. Here we proposed that distinct tissue niches drive the phenotypic heterogeneity of $T_{RM}$ cells. To test this, we leveraged spatial transcriptomics of human samples, a mouse model of acute systemic viral infection and a newly established strategy for pooled optically encoded gene perturbations to profile the locations, interactions and transcriptomes of pathogen-specific $T_{RM}$ cell differentiation at single-transcript resolution. We developed computational approaches to capture cellular locations along three anatomical axes of the small intestine and to visualize the spatiotemporal distribution of cell types and gene expression. Our study reveals that the regionalized signalling of the intestinal architecture supports two distinct $T_{RM}$ cell states: differentiated $T_{RM}$ cells and progenitor-like $T_{RM}$ cells, located in the upper villus and lower villus, respectively. This diversity is mediated by distinct ligand–receptor activities, cytokine gradients and specialized cellular contacts. Blocking TGFβ or CXCL9 and CXCL10 sensing by antigen-specific CD8 T cells revealed a model consistent with anatomically delineated, early fate specification. Ultimately, our framework for the study of tissue immune networks reveals that T cell location and functional state are fundamentally intertwined.

$T_{RM}$ cells have a pivotal role in the adaptive immune response, offering localized, long-term protection in non-lymphoid tissues through continuous tissue surveillance[2]. $T_{RM}$ cell formation requires the engagement of transcriptional and metabolic programs that induce tissue-specific adaptations[3–6]. These programs, initiated by priming events in lymphoid tissues[7], are reinforced by cellular interactions and the sensing of environmental factors on tissue entry[8,9], such as TGFβ[10,11], which enhances the upregulation of retention molecules including CD103 (encoded by *Itgae*) in epithelial barriers such as the small intestine (SI)[10]. Recent studies have shown that intestinal $T_{RM}$ cells exhibit functional heterogeneity, with at least two distinct states identified in response to acute infections in the SI: a more terminally differentiated $T_{RM}$ cell population and a progenitor-like one[1,4,12,13]. These subpopulations display distinct cytokine production and secondary memory potential, highlighting the nuanced nature of the development and function of $T_{RM}$ cells in tissues to provide long-term immunity. In fact, recent investigations into the architectural structure of the mouse and human SI have uncovered spatially organized transcriptional and metabolic programs that establish complex intestine regionalization[14–16]. How these tissue microenvironments affect $T_{RM}$ cell differentiation and function has not been previously addressed. This is in part because of the

fine granularity needed to capture different CD8 T cell transcriptional states and the variety of the non-immune cellular phenotypes that surround them, which requires highly multiplexed subcellular-resolution imaging technologies. Spatial transcriptomics enables the profiling of hundreds of different mRNA molecules simultaneously in cells across complete tissue sections, in combination with protein and histology read-outs. Here, we exploit spatial transcriptomics and develop a computational framework to characterize the cellular and ligand–receptor interactions that guide intestinal $T_{RM}$ cell differentiation. We further show the feasibility of a multiplexed, optically encoded spatial CRISPR knockout experiment in an in vivo setting to explore cytokine gradients during immune cell differentiation and expand our models of T cell differentiation in the SI.

## A spatial framework for SI $T_{RM}$ cells

To quantitatively and systematically capture the spatial distribution of antigen-specific CD8 T cells in the SI responding to lymphocytic choriomeningitis virus (LCMV) infection, we used T cell receptor transgenic CD8 T cells (P14 CD8 T cells), which specifically recognize the LCMV glycoprotein 33–41 peptide presented by H2-D[b]. After adoptive transfer

[1]School of Biological Sciences, Department of Molecular Biology, University of California, San Diego, La Jolla, CA, USA. [2]La Jolla Institute for Immunology, La Jolla, CA, USA. [3]Department of Cellular and Molecular Medicine, University of California, San Diego, La Jolla, CA, USA. [4]Department of Medicine, University of California, San Diego, La Jolla, CA, USA. [5]Department of Medicine, Veteran Affairs San Diego Healthcare System, San Diego, CA, USA. [6]Allen Institute for Immunology, Seattle, WA, USA. [7]These authors contributed equally: Miguel Reina-Campos, Alexander Monell, Maximilian Heeg. ✉e-mail: maximilian.heeg@alleninstitute.org; agoldrath@alleninstitute.org

of P14 CD8 T cells from mice with a CD45.1 congenic background into CD45.2 wild-type mice and LCMV infection, SIs were collected at different timepoints to analyse the number and intratissue location of transferred cells (Extended Data Fig. 1a). To obtain an unbiased assessment of cell location in the tissue, we implemented a dual-coordinate-axis system based on the proximity of individual P14 CD8 T cells, detected by histological staining of CD45.1 and CD8α, to the nearest epithelial cell or the distance to the base of the muscularis (Fig. 1a and Extended Data Fig. 1a). This approach creates a two-dimensional density representation of the overall distribution of P14 CD8 T cells in the villus, the repetitive functional structure of the SI, which we termed immune allocation plot (IMAP) (Fig. 1a). IMAPs generated over the course of an LCMV infection captured the distribution dynamics of P14 CD8 T cells in the SI, revealing infiltration of the muscularis at an effector timepoint, followed by rapid clearance and a subsequent formation of two spatially separated populations along the crypt–villus axis (Fig. 1b). These data suggest that P14 CD8 T cells dynamically occupy different regions in the SI after infection.

To study the relationship between the gene-expression programs and locations of $T_{RM}$ cells, we adoptively transferred female P14 CD8 T cells to male mice, infected them with LCMV and performed spatial transcriptomic profiling (Xenium, 10x Genomics)[17,18] on mouse SIs over the course of the LCMV infection (6, 8, 30 and 90 days post infection (d.p.i.)). A 350-plex target gene panel was designed using a reference single-nucleus RNA sequencing (RNA-seq) dataset to inform a prioritization algorithm based on predictive deep learning[19] (Extended Data Fig. 1b). Furthermore, we included probes to detect relevant immune gene markers curated from the ImmGen database[20], ligand–receptor pairs[21] and *Xist*, a long non-coding RNA exclusively expressed in female cells that was used to track P14 CD8 T cells in the male hosts (Extended Data Fig. 1b). Recursive feature-elimination modelling showed that 350 genes captured the biological heterogeneity found in the SI using single-nucleus RNA-seq (Extended Data Fig. 1c and Supplementary Table 1). After Xenium processing, tissues were further analysed by immunohistochemical detection of CD8α and cellular membranes with wheat germ agglutinin (WGA), as well as haematoxylin and eosin (H&E) staining, which enabled the assessment of overall tissue structures (Fig. 1c). This analysis provided in situ measurements of 350 genes on 1.8 million cells over four timepoints, with highly correlated biological duplicates ($r \geq 0.93$), capturing an average median count of 98 transcripts per cell and enabling the identification of 36 distinct cell types, including P14 CD8 T cells (CD8α$^+$ and *Cd8a$^+$ Cd8b$^+$ Xist$^+$*) (Fig. 1c,d, Extended Data Fig. 1d–f and Supplementary Table 2). As expected with the course of infection, endogenous CD8αβ T cell and P14 CD8 T cell fractions were the most changed after LCMV infection with a notable increase in B cell frequencies (Fig. 1d and Extended Data Fig. 1f). Similar dynamics were observed by flow cytometry analysis (Extended Data Fig. 1g and Supplementary Fig. 1). *Xist* expression was successfully detected in P14 CD8 T cells across timepoints (Extended Data Fig. 1h), and female CD8 T cells were able to differentiate effectively into SI $T_{RM}$ cells in a male host, as shown by a mixed female:male transfer in the context of LCMV infection (Extended Data Fig. 1j,k). Notably, spatial visualization of phenotypic diversity by Leiden clustering revealed a marked spatial pattern (Fig. 1d), suggesting that our approach was able to capture transcriptional regionalization while simultaneously being able to capture a broad range of cell types in the SI.

## SI $T_{RM}$ cell diversity is spatially defined

To study the transcriptional programs of SI $T_{RM}$ cells as a function of their unique location in the tissue, we generated IMAP representations for each cell type (Extended Data Fig. 2) by calculating coordinates along three main axes of reference for every cell: distance to the muscularis (crypt–villus axis) (Fig. 2a), distance along the gastrointestinal tract (longitudinal axis) (Fig. 2b) and distance to the nearest epithelial

cell (epithelial axis) (Fig. 2c). IMAPs for P14 CD8 T cells across the infection time course revealed a dynamic shift in their location over time, from a relative accumulation in the lower villus and muscularis at 6 d.p.i. to forming two spatially resolved populations along the crypt–villus axis by 90 d.p.i. (Fig. 2d). P14 CD8 T cells were uniformly distributed along the longitudinal axis over time (Extended Data Fig. 3a). We next considered whether the cellular distribution had an effect on P14 CD8 T gene expression programs. Out of the 191 genes detected in P14 CD8 T cells (expressed in more than 5% of these cells), 87 genes (46%) had notable changes in expression along the crypt–villus axis, 76 genes (40%) changed along the epithelial axis and 8 genes (4%) changed along the longitudinal axis (Fig. 2e). These results indicated strong transcriptional imprinting based on the intratissue location of P14 CD8 T cells, influenced primarily by the crypt–villus and epithelial axes. Other immune cells displayed similar correlations with variable expression along the epithelial and crypt–villus axes, as well as epithelial cells, which followed expression gradients consistent with previous studies[14–16,22] (Extended Data Fig. 3b–d and Supplementary Table 3). In P14 CD8 T cells, genes associated with progenitor-like $T_{RM}$ cells, *Tcf7* and *Slamf6*, and short-lived effectors (*Il18r1* and *Klrg1*), were expressed in the lower villus area, whereas genes associated with differentiated $T_{RM}$ cells, such as *Itgae*, *Gzma* and *Gzmb*, were highly expressed around the top of the villus (Fig. 2f–h). Consistently, CD8α$^+$TCF1$^+$ cells were found predominantly in the crypt and lower villus area whereas CD8α$^+$ GZMB$^+$ cells were found predominantly in the top intraepithelial side of the villus as revealed by immunostaining (Extended Data Fig. 3e). Transcriptional gradients were also apparent along the epithelial axis, with progenitor-like $T_{RM}$-cell- and short-lived effector-cell-associated genes preferentially expressed by P14 CD8 T cells in the lamina propria (Fig. 2i). At the population level, *Gzma* and *Gzmb* expression decreased concomitantly with an increase in *Tcf7* expression over time (Extended Data Fig. 3f), as previously shown by single-cell RNA-seq[1,12] (Extended Data Fig. 3g). Despite these overall changes at the population level over time, the gene-expression polarization along the crypt–villus axis was evident at each individual timepoint (Fig. 2f–i). To put our results in the context of our previous observations, the projection of gene signatures for progenitor-like $T_{RM}$ cells and more differentiated $T_{RM}$ cells (clusters 3 and 29, respectively[12]) into the spatial coordinates of P14 CD8 T cells at 90 d.p.i. delineated a population of $T_{RM}$ cells expressing cluster 3 genes preferentially located near the bottom of the villus, and $T_{RM}$ cells enriched for expression of cluster 29 located at the top of the villus (Fig. 2j). Similar results were obtained using signatures derived from long-lived ID3$^+$ $T_{RM}$ cells and shorter-lived effector BLIMP1$^+$ $T_{RM}$ cells[1] (Extended Data Fig. 3h). Together, these data show that the phenotypic and gene-expression heterogeneity of SI CD8 T cells depends largely on their location in the tissue.

## Regionalized intestinal immune signalling

To understand better how the anatomical position in the SI leads to $T_{RM}$ cell heterogeneity, we focused on the composition of the cellular communities around P14 CD8 T cells over time (Fig. 3a). A graphic representation of the cellular connectome of the SI separated four spatial domains: immune (lamina propria), epithelial, muscularis and crypt regions (Fig. 3b). At effector timepoints (6 and 8 d.p.i.), P14 CD8 T cells interacted mostly with immune cells of the lamina propria and fibroblasts, whereas connections with immune cells of the upper villus and enterocytes dominated at later memory timepoints (Fig. 3b), consistent with their location (Fig. 2d). P14 CD8 T cells nearer to the tip of the villus (enriched for expression of cytotoxic molecules) were in closer proximity to enterocytes, and immune cells preferentially located in the top intraepithelial area, such as mucosal-associated invariant T cells and natural killer cells, whereas P14 CD8 T cells nearer to the crypts (enriched for expression of progenitor-like genes) had increased physical proximity to B cells, CD4 T cells, fibroblasts and progenitor

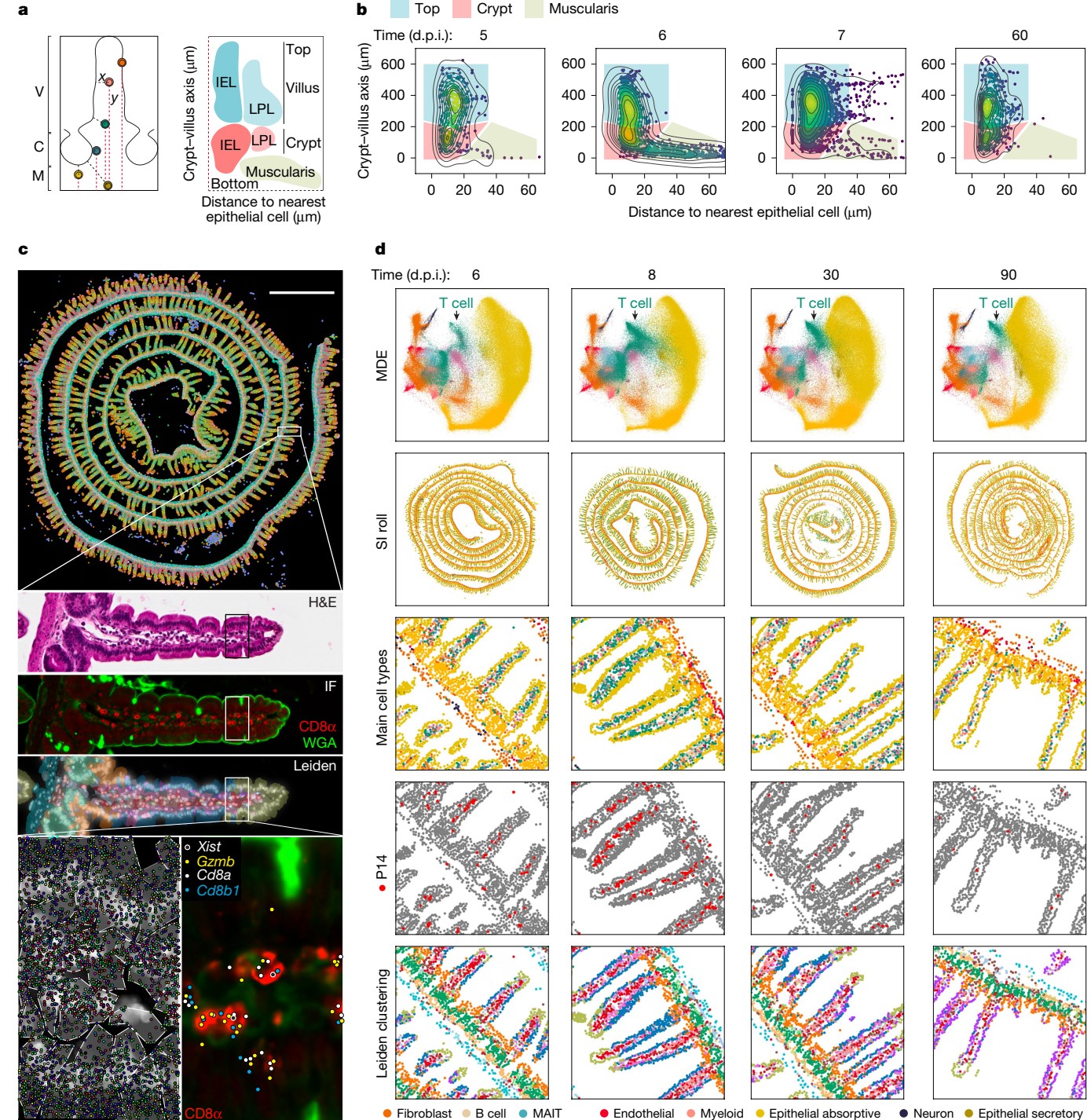

**Fig. 1 | Characterization of the spatial and transcriptional state of antigen-specific CD8 T cells in response to acute viral infection in the mouse SI with spatial transcriptomics. a**, Left, a coordinate system to define morphological axes in the SI. C, crypt; M, muscularis; V, villus. Right, the distance to the nearest epithelial cells and the distance to muscularis form the basis of an IMAP. The top of the villus and the crypt regions house both intraepithelial lymphocytes (IEL) and lamina propria lymphocytes (LPL). **b**, An IMAP representation of P14 T cell localization measured by immunofluorescence (staining at the indicated days after infection. Two biological replicates for *n* = 3 mice per timepoint, representative data from one mouse are shown. The gates for the top of the villus (blue), the crypt (red) and the muscularis (beige) highlight the different regions. The points, representing cell positions, are coloured by kernel density estimates over the IMAP (*x*, *y*) coordinates. **c**, An overview of the Xenium-based spatial transcriptomics data structure: row 1, Xenium output of a mouse intestine (8 d.p.i.), with cells coloured by Leiden cluster;

row 2, a magnification of a villus showing H&E staining; row 3, confocal immunofluorescence images of CD8α and WGA; row 4, Xenium DAPI staining with cell boundary segmentation masks coloured by Leiden cluster; row 5, a further subregion magnification of the same villus depicting Xenium DAPI staining overlaid by cell boundary segmentation and all transcripts assigned to cells (left) and an immunofluorescence image of CD8α and WGA overlaid with transcript locations for *Cd8a*, *Cd8b*, *Gzmb* and female P14-specific *Xist* transcript locations overlaid (right). **d**, An overview of the processed Xenium data of mouse SI at 6, 8, 30 and 90 d.p.i. (columns): row 1, cells in a joint minimum distortion embedding (MDE) of all SI Xenium samples coloured by cell type; row 2, in situ spatial positioning of the cells; rows 3–5, magnifications coloured by cell type (row 3), with P14 clusters/cells highlighted in red (row 4) and coloured by Leiden cluster (row 5). One of two biological replicates for each timepoint is shown. DC, dendritic cell; ILC, innate lymphoid cell; MAIT, mucosal-associated invariant T; NK, natural killer cell.

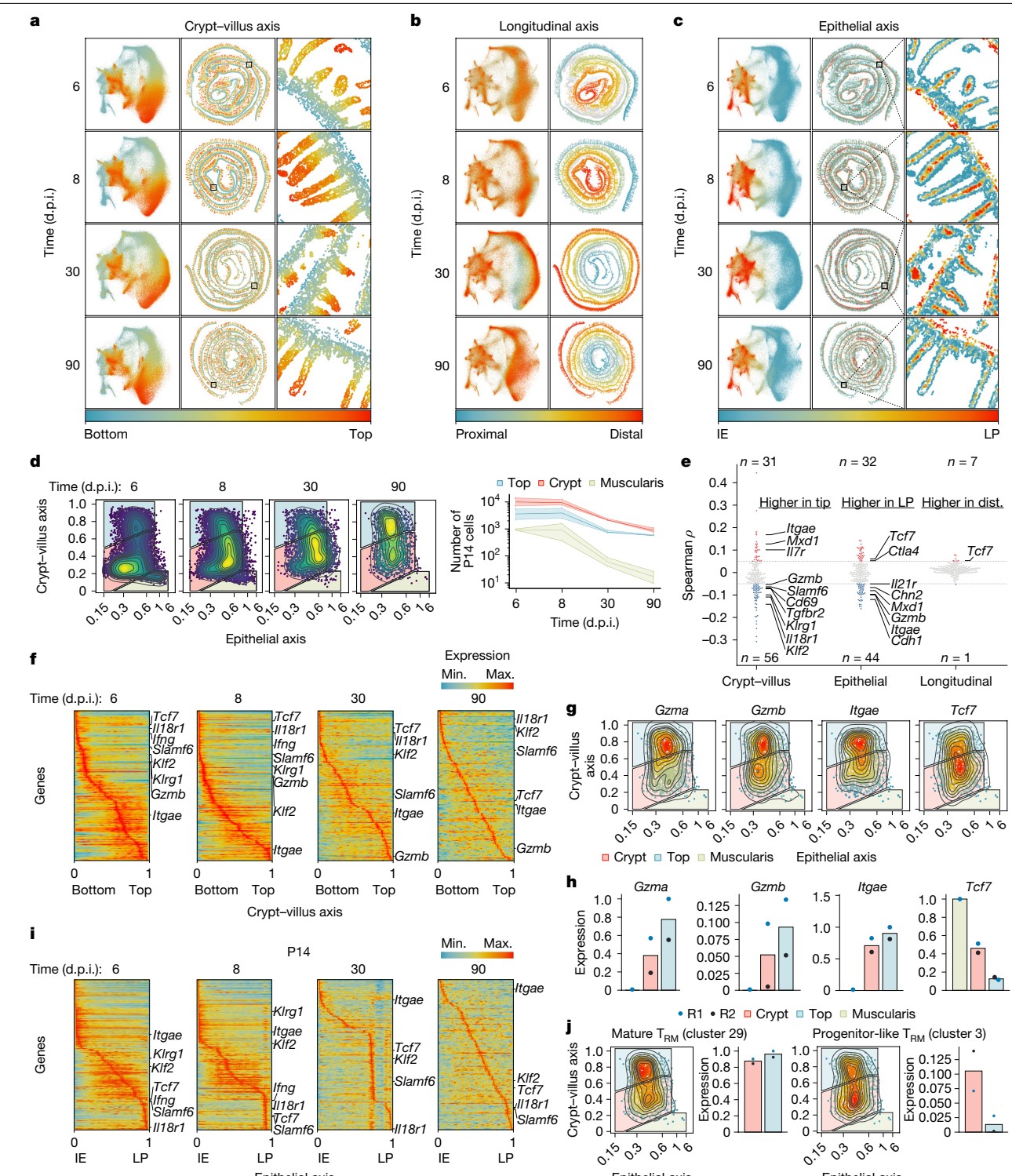

**Fig. 2 | Intestinal regionalization along key axes informs $T_{RM}$ cell diversity in the mouse intestine. a–c**, The spatial position and joint MDE embedding, coloured by their crypt–villus axis position (**a**), longitudinal axis position (**b**) and epithelial axis position (**c**). One of two biological replicates for each timepoint is shown. **d**, IMAPs of P14 CD8 T cells in samples from each timepoint (one of two replicates for each timepoint), with coloured gates dividing the top, crypt and muscularis (left) and the number of P14 CD8 T cells positioned in each gate across timepoints (two biological replicates for each timepoint) (right). Data are mean ± s.e.m. **e**, Combined time-course samples (*n* = 8, four timepoints with two biological replicates each) were pooled to create a swarm plot of Spearman rank correlation coefficients (*ρ*) between each axis and every gene expressed in at least 5% of P14 cells, with select correlated genes annotated. Genes are considered positively correlated (red) when *ρ* > 0.05, negatively correlated

(blue) when *ρ* < −0.05 and not correlated (grey) otherwise. **f**, The convolved gene expression of P14 CD8 T cells along the crypt–villus axis at every timepoint (*n* = 2 pooled biological replicates). **g**, IMAP representations of P14 CD8 T cells at 90 d.p.i. coloured by kernel density estimates weighted by expression counts of select genes (one of two biological replicates is shown). **h**, The expression of each gene in IMAP-gated regions of P14 CD8 T cells at 90 d.p.i. (*n* = 2 replicates, R1 and R2). **i**, The convolved gene expression of P14 CD8 T cells along the epithelial axis at every timepoint (*n* = 2 pooled biological replicates). **j**, IMAP representations of P14 CD8 T cells at 90 d.p.i. (one of two biological replicates is shown) coloured by kernel density estimates weighted by UCell signature enrichment of progenitor-like $T_{RM}$ cells (cluster 3 (ref. 12)) and differentiated $T_{RM}$ cells (terminal state, cluster 29)[12] with signature scores in IMAP-gated regions (*n* = 2). IE, intraepithelial; LP, lamina propria; Max., maximum; Min., minimum.

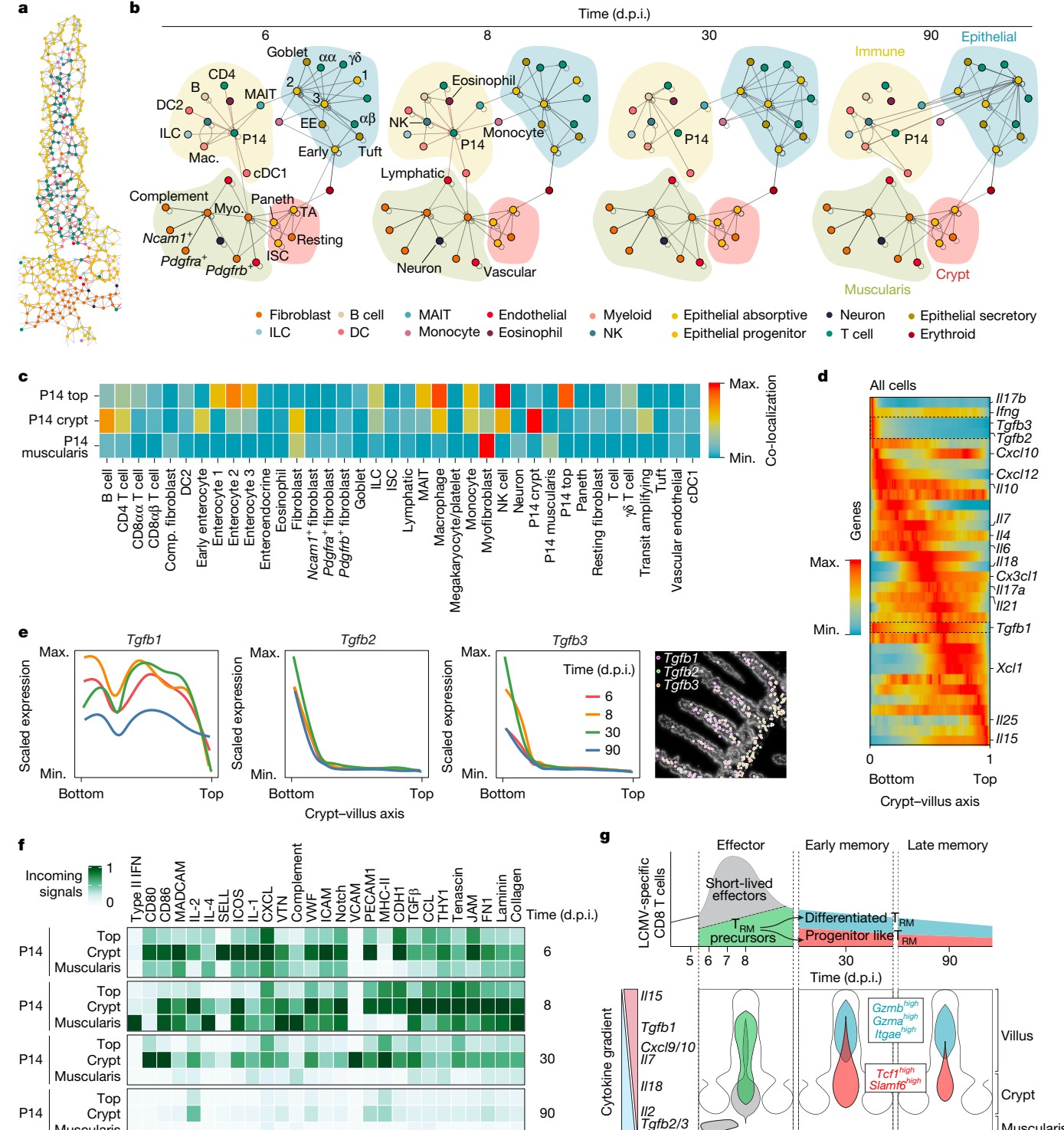

**Fig. 3 | Differential cytokine gradients and cellular communities across intestinal niches. a,b**, A representation of the connectome between cell subtypes at an individual cell resolution in a villus at 8 d.p.i. (**a**) and an aggregated network format in which edges between nodes represent a normalized Squidpy interaction score lying above a 0.1 threshold (10% of the connections) (**b**). Node (*x*, *y*) positions were determined by running a Kamada–Kawai layout algorithm on the Squidpy interaction matrix of the two replicates at 8 d.p.i. Positions were visualized using igraph. For each timepoint, the interaction scores between nodes are averaged across the two biological replicates. **c**, Squidpy interaction scores between cell subtypes and P14-cell regional groupings–top, crypt and muscularis as depicted in Fig. 2d. The colour of the heat-map position reflects the strength of contact. Interaction scores are averaged across the eight samples, and values are row-normalized. **d**, The convolved gene expression

of cytokines along the crypt–villus axis ordered and displayed with scVelo pooled across all time-course samples for all cells (*n* = 8). **e**, Gene expression trends for TGFβ isoforms separated by timepoint (*n* = 2 biological replicates pooled) with representative TGFβ isoform expression depicted spatially at their positions on a villus from a SI (8 d.p.i.). A generalized additive model is used to fit a curve to the expression counts of each ligand along the crypt–villus axis, followed by *z*-score scaling for comparison across trends. **f**, A heat map showing the pathways contributing the most to incoming signalling of each P14-cell regional grouping. The relative strengths of each pathway were calculated using spatial CellChat on *n* = 2 samples from four timepoints. **g**, The spatiotemporal differentiation model for intestinal T_RM cells. ISC, intestinal stem cell; cDC1, conventional type 1 dendritic cell; Comp, complement.

enterocytes (Fig. 3c). These data showed that the spatial polarization of T$_{RM}$ cell states includes a different set of cellular interactions, which could potentially contribute to inducing and maintaining this phenotypic diversity through local signals. To capture the potential for differential signalling along the villus, we focused on cytokine gradients along the crypt–villus axis captured in our Xenium dataset (Fig. 3d). Several key cytokines involved in T$_{RM}$ cell formation and maintenance showed pronounced expression gradients, including *Il10* (ref. 23), *Il7*, *Il21*, *Il15* (refs. 24,25) and TGFβ isoforms[10] (Fig. 3d,e). An analysis of incoming signalling patterns in P14 CD8 T cells highlighted the MAD-CAM[26], ICAM[27], CCL, CXCL[28] and TGFβ signalling pathways as differentially abundant along the crypt–villus axis, revealing their potential role as upstream regulators of the heterogeneity observed in T$_{RM}$ cell populations (Fig. 3f). These pathways encompass individual ligand–receptor interactions weighted by their prevalence (Supplementary Table 4). To obtain a more comprehensive view of cytokine gradients and validate our findings through an orthogonal approach, we profiled a mouse SI at 8 d.p.i. with LCMV infection using whole-genome spatial transcriptomics at 2 μm resolution (VisiumHD)[29] (Extended Data Fig. 4a). Deconvolution of the crypt–villus axis revealed similar cytokine profiles to those found by Xenium, in addition to new ones not included in our Xenium gene panel (Extended Data Fig. 4b–d). To perform ligand–receptor analysis at single-cell resolution, we binned the 2-μm-resolution capture areas into H&E-segmented nuclei (Extended Data Fig. 4e). After cell typing and classification of CD8αβ T cells by their location in the villus, we observed differential incoming signals similar to those observed in Xenium mouse data, including the MADCAM, ICAM, CCL and CXCL families (Extended Data Fig. 4f and Supplementary Table 5).

These data indicate that signals in the SI architecture are poised to influence the influx of new CD8 T cells and steer their differentiation course. To test whether these signalling gradients exist before the infection onset, we performed Xenium profiling using a custom 480-gene panel (expanded from our original 350 genes; Supplementary Table 4) on two uninfected SIs. The data were highly correlated between the two biological duplicates (Extended Data Fig. 5a). Uninfected SIs showed a similar distribution of key cytokine expressions, including *Tgfb1*, *Tgfb2*, *Tgfb3*, *Il18* and *Il15* (Extended Data Fig. 5b), a similar connectome (Extended Data Fig. 5c) and a similar spatial polarization of CD8αβ T cells based on the expression of differentiated T$_{RM}$ cells (*Gzma*, *Gzmb* and *Itgae*) and progenitor-like T$_{RM}$ cells (*Tcf7* and *Id3*) (Extended Data Fig. 5d–f). Together, these data show that niche-dependent signals are poised to contribute both to differentiating incoming CD8 T cells as well as to maintaining the two polarized T$_{RM}$ cell states: a more differentiated T$_{RM}$ cells state at the top of the villus (high expression of *Gzma*, *Gzmb* and *Itgae*) and a progenitor-like state around the crypt area (high expression of *Tcf7* and *Slamf6*). These two distinct functional states are the opposite ends of a bimodal differentiation space and appear to be selected for and reinforced by their respective microenvironments over time (Fig. 3g).

## TGFβ directs SI CD8 T cell positioning

TGFβ is an immunomodulatory cytokine that affects intestinal CD8 T cell homing and retention[10,11] and instructs T$_{RM}$ cell differentiation and maintenance[4,30–32]. Yet, how TGFβ signalling is spatially orchestrated in the SI is not well understood. Differential cellular communication analysis comparing P14 CD8 T cells at the crypt versus the top of the villus revealed a segregation of TGFβ isoform expression available to P14 CD8 T cells based on location (Extended Data Fig. 6a). As P14 CD8 T cells showed similar expression levels of *Tgfbr1* and *Tgfbr2* in the villus (Extended Data Fig. 6b), we posited that TGFβ signalling by CD8 T cells depends on their exposure to sources of TGFβ and/or cells that trans-present the active form of TGFβ on their surface through αβ integrins, such as αV (*Itgav*) coupled to β6 (*Itgb6*) or β8 (*Itgb8*)[11].

For example, upper enterocytes (enterocyte type 1) did not express TGFβ isoforms but could, in theory, present TGFβ owing to expression of *Itgav* and *Itgb6* (Extended Data Fig. 6c). To mechanistically explore this idea, spatial transcriptomics profiling was conducted at 8 d.p.i. for the SI of mice that received wild-type or TGFβ receptor II (TGFβRII) knockout P14 CD8 T cells (Extended Data Fig. 6d). To incorporate more elements of the TGFβ signalling pathway, we designed an expanded 494-gene panel and used the MERSCOPE platform (Vizgen) (Extended Data Fig. 6e and Supplementary Table 7). This experiment generated a 470,000-cell dataset with an average of 186 transcripts per cell. TGFβRII knockout P14 CD8 T cells, labelled by the expression of *Xist* and *Cd8a* and *Cd8b* transcripts, had an overall spatial distribution shifted towards the bottom of the villus and muscularis area and lower *Itgae* transcripts than wild-type P14 CD8 T cells (Extended Data Fig. 6f,g). Their total numbers were not changed at this time after infection, as previously observed[10] (Extended Data Fig. 6h). Differential gene expression between wild-type and TGFβRII knockout P14 CD8 T cells showed global downregulation of the core T$_{RM}$ cell signature[5], as well as the TGFβ program (Extended Data Fig. 6i), including *Itgae*, *Cxcr6*, *Cd160* and *P2rx7*, and upregulation of *Klf2*, *Il18rap*, *S100a4* and *Mki67*, consistent with published regulation by TGFβ signalling[33] (Extended Data Fig. 7a). Notably, *Mki67* was upregulated by TGFβRII knockout P14 CD8 T cells, and *Mki67*[+] TGFβRII knockout cells were preferentially located lower in the villus compared with wild-type cells, suggesting that the loss of adequate TGFβ programming caused an increased frequency of mislocalized proliferating cells (Extended Data Fig. 7b). To understand how P14 CD8 T cells engage in the TGFβ program on the basis of their proximity to other cell types, we used the top eight differentially expressed genes as a TGFβ signature and calculated a correlation with the nearest distance to different cell types (Extended Data Fig. 7c,d). Wild-type P14 CD8 T cells had the highest expression of the TGFβ program when closest to enterocytes and other cells at the top of the villus, and lowest when wild-type P14 CD8 T cells were in the proximity of cells located at the bottom of the villus (Extended Data Fig. 7d). TGFβRII knockout P14 CD8 T cells had a global loss of the transcription–distance correlation, consistent with the loss of TGFβ signals (Extended Data Fig. 7d). Gene-expression visualization of TGFβ isoforms and components of the trans-presentation machinery showed fibroblast populations to be producers, consistent with our Xenium dataset (Extended Data Figs. 6c and 7d). Next, we analysed differences in the physical distances between wild-type and TGFβRII knockout P14 CD8 T cells to each nearest cell type. These analyses showed an increased accumulation of TGFβRII knockout P14 CD8 T cells near fibroblasts and a concomitant distancing from enterocytes in the villus (Extended Data Fig. 7d,e and Supplementary Table 8). These changes were not explained by the compensatory expression of TGFβ molecules (Extended Data Fig. 7f). These data add to the current model of T$_{RM}$ cell differentiation by showing that CD103[+] T$_{RM}$ cell precursors, the population depleted by TGFβRII loss[4], preferentially occupy the intravillous region and upper half of the villus, where they receive both TGFβ signalling and access to important cytokines for T$_{RM}$ cell differentiation and survival, such as IL-7 and IL-15 (refs. 32,34) (Extended Data Fig. 7g).

## CXCR3-mediated SI CD8 T cell deployment

CXCR3 ligands, CXCL9 and CXCL10, are known T cell chemoattractants that contribute to T cell positioning and maturation in the SI[28]. However, it is unclear how these signals are distributed in the SI to program CD8 T cell fate. By looking at the expression of these two chemokines over the course of infection, we found that, although *Cxcl9* and *Cxcl10* signals were heavily enriched in the top half of the lamina propria during homeostasis, their expression was strongly induced after infection in C3-expressing fibroblasts (complement fibroblasts or adventitia fibroblasts), which are located at the bottom of the muscularis, creating a second potential attraction point for *Cxcr3*-expressing CD8 T cells (Fig. 4a,b). To test the role of these gradients in CD8 T cell location

and differentiation, we used a CRISPR–Cas9 approach to induce *Cxcr3* deletion in Cas9 P14 CD8 T cells. To optically identify single guide RNA (sgRNA)-containing cells using the Xenium assay, we introduced a pseudogene barcode in the 3′ untranslated region of the mCherry reporter used to sort the modified Cas9 P14 CD8 T cells. We also included Cas9 P14 CD8 T cells containing *Cd19*- and *Thy1*-targeting sgRNAs with different pseudogene barcodes as controls in the same experiment (Supplementary Table 9). After validation of their respective target genes by flow cytometry (Extended Data Fig. 8a), sgCxcr3, sgCd19 and sgThy1 P14 CD8 T cells were pooled at equal frequencies and transferred into mice, which were subsequently infected with LCMV (Fig. 4c). The pseudogene barcodes, unique for each sgRNA, used the same landing sequences as 3 genes included in the 350-gene panel that were the least expressed across all cells and undetectable in T cells (Extended Data Fig. 8b). At 8 d.p.i., the SI from two different mice that received the targeted T cells were profiled using the Xenium platform. Unique pseudogene barcodes were found almost exclusively in CD8 T cells (perturbed P14 CD8 T cells), and the number of barcodes detected per P14 cell was uniform across the crypt–villus axis (Fig. 4e and Extended Data Fig. 8b–d). sgRNA-transduced P14 CD8 T cells showed a significant decrease in their respective target genes (Fig. 4d). sgCxcr3-targeted P14 CD8 T cells had a marked upward shift in their villus positioning, with an enrichment in the top intraepithelial area of the villus, reduction in the lamina propria and almost no presence in the muscularis (Fig. 4e, Extended Data Fig. 8e and Supplementary Table 11). Analysis of gene expression by spatial gates showed that *Cxcr3*-deficient P14 CD8 T cells had increased *Gzma* and *Itgae* expression and lower *Klf2* than control cells in certain regions (Fig. 4f). Thus, the loss of *Cxcr3* induced a preferential accumulation of cells in the top intraepithelial area, and this positional shift further reinforced a more differentiated $T_{RM}$ cell phenotype (Fig. 4g). Loss of *Cxcr3* has been shown to deplete short-lived effector populations and to be required for the formation of CD103⁻ $T_{RM}$ cells in the lamina propria without affecting intraepithelial $T_{RM}$ cell numbers[28]. Our data add to this model of differentiation by showing that short-lived effector cells are probably preferentially located in the lower half of the villus, crypt and stroma areas, attracted by CXCL9- and CXCL10-expressing immune and stromal cells in the lamina propria and muscularis early after infection. Together, these data highlight the causal connection between CD8 T cell location and phenotype and introduce an expandable approach for the spatial screening of perturbed T cells.

## Spatial CD8 T cell programs in the human SI

To put the relevance of these findings into the context of human intestinal immunity, we profiled two terminal ilea from healthy donors using the Xenium platform with a custom immune-focused 422-gene panel (Extended Data Fig. 9a and Supplementary Table 10). Combined, we generated a dataset comprising 214,546 cells with an average of 89 median transcripts per cell and identified 38 different cell types (Fig. 5a and Extended Data Fig. 9b). Technical replicates (consecutive slides) were nearly identical, and the observed cell frequencies were similar between biological samples, except for B cells owing to differences in the abundance of Peyer's patches (Fig. 5b and Extended Data Fig. 9c). IMAPs were computed by calculating the crypt–villus axis using spatial transcriptional neighbourhoods around each cell as a predictor for distance to the basal membrane and by calculating epithelial distance axes using the normalized distance to the nearest epithelial cells (Fig. 5c). First, to explore whether spatial distribution differences were observed between T cell populations, we focused on T cell types that emerged from unsupervised clustering (Extended Data Fig. 9d). CD8 T cells, proliferating T cells and γδ T cells were predominant in the top intraepithelial area, whereas effector T cells, including *GZMK*⁺ CD8 T cells, were predominantly localized in the lamina propria and more evenly distributed along the crypt–villus axis (Extended Data Fig. 9e). These differences in location were reflected by their distinct

interactions with nearby cells (Extended Data Fig. 9f). Next, we focused our analysis on CD8αβ T cells outside the Peyer's patches (Fig. 5c). Projecting the transcriptional signatures of the two spatially segregated P14 CD8 T cell populations we observed at 90 d.p.i. in mice, we identified similarly segregated subpopulations in human CD8αβ T cells (Fig. 5d). Furthermore, spatial differential expression analysis between human CD8αβ T cells in different morphological regions recapitulated the mouse findings, as many genes were spatially regulated, often similarly to their mouse homologues. These include *GZMA* and *ITGAE*, which localize near the epithelium at the top of the villus, and *KLRG1* and *TCF7*, which localize to the lamina propria and bottom of the villus as they do in mouse P14 CD8 T cells (Fig. 5e,f and Extended Data Fig. 9g). These data show that human intestinal CD8 T cells are similarly imprinted by their location in the tissue. Correlation analysis of gene expression in CD8 T cells with their relative distances to other cell types highlighted effector molecules *ITGAE* and *GZMA* to be most highly expressed by CD8αβ T cells when in proximity to enterocytes and the long-lived memory-associated molecule *TCF7* to be most highly expressed by CD8αβ T cells when in proximity to CD4 T and B cells (Fig. 5g). Finally, differential cellular communication analysis between intestinal CD8 T cells, classified by the mouse spatial signatures (as in Fig. 5d), identified differential incoming signalling (Fig. 5h). Among these, ICAM (communication with vascular endothelial cells) was more prevalent in the progenitor-like-signature-enriched CD8αβ T cells, whereas C-type lectins and CC chemokine signalling (CCL, communication with other immune cells) were favoured for differentiated-signature-enriched CD8αβ T cells and TGFβ signalling was similar for both CD8αβ T cell subtypes (Fig. 5h and Supplementary Table 12). In summary, these data suggest that the heterogeneity in the phenotypes and gene expression observed in CD8 T cells in the SI is imprinted by their intratissue location, especially along the crypt–villus axis, which, through differential cellular interactions and exposure to chemokines and cytokine sources, such as TGFβ, maintains functionally different populations of CD8αβ T cells in both human and mouse.

## Discussion

High-resolution spatial transcriptomics interrogation of mouse models of viral infection and human samples revealed that the intestinal cellular architecture shapes CD8 T cell diversity in the intestine. Our findings support a model in which the intestinal architecture creates localized instructive signals after infection through regionalization of cytokine secretion and distinct cellular interactions to produce fate-specifying areas that reinforce $T_{RM}$ cell precursor or short-lived effector CD8 T cell differentiation. Furthermore, spatial patterning throughout the crypt–villus and epithelial-distance axes reveals a distinct signalling potential that divides $T_{RM}$ cells into at least two distinct functional states. We observed similar signalling gradients and differentiation states in both mice and humans, epitomized by differentiated $T_{RM}$ cells at the intraepithelial top of the villi and progenitor-like $T_{RM}$ cells enriched in the lamina propria at the base of the villi.

Our data provide an understanding of how the TGFβ program is imparted on intestinal CD8 T cells to promote $T_{RM}$ cell programs in situ. Although many cells, both in mice and in humans, can express TGFβ isoforms (Extended Data Fig. 9h), the loss of TGFβRII leads to a preferential accumulation of CD8 T cells around fibroblasts in the lower villus area. This suggests that TGFβ signalling is required for $T_{RM}$ cells to find pro-survival and differentiation cues in the mid and upper areas of the villus where *Il7* and *Il15* are more abundantly expressed. Of note, although enterocytes do not express detectable TGFβ in mice or humans, they do express ITGAV, which is required for the presentation of the active TGFβ signal (Extended Data Fig. 9h). In the skin, migratory dendritic cells use ITGAV to provide TGFβ stimulation to naive CD8 T cells to precondition a skin $T_{RM}$ cell fate, whereas keratinocytes maintain epidermal residence[11,31]. In the future, it will be important to

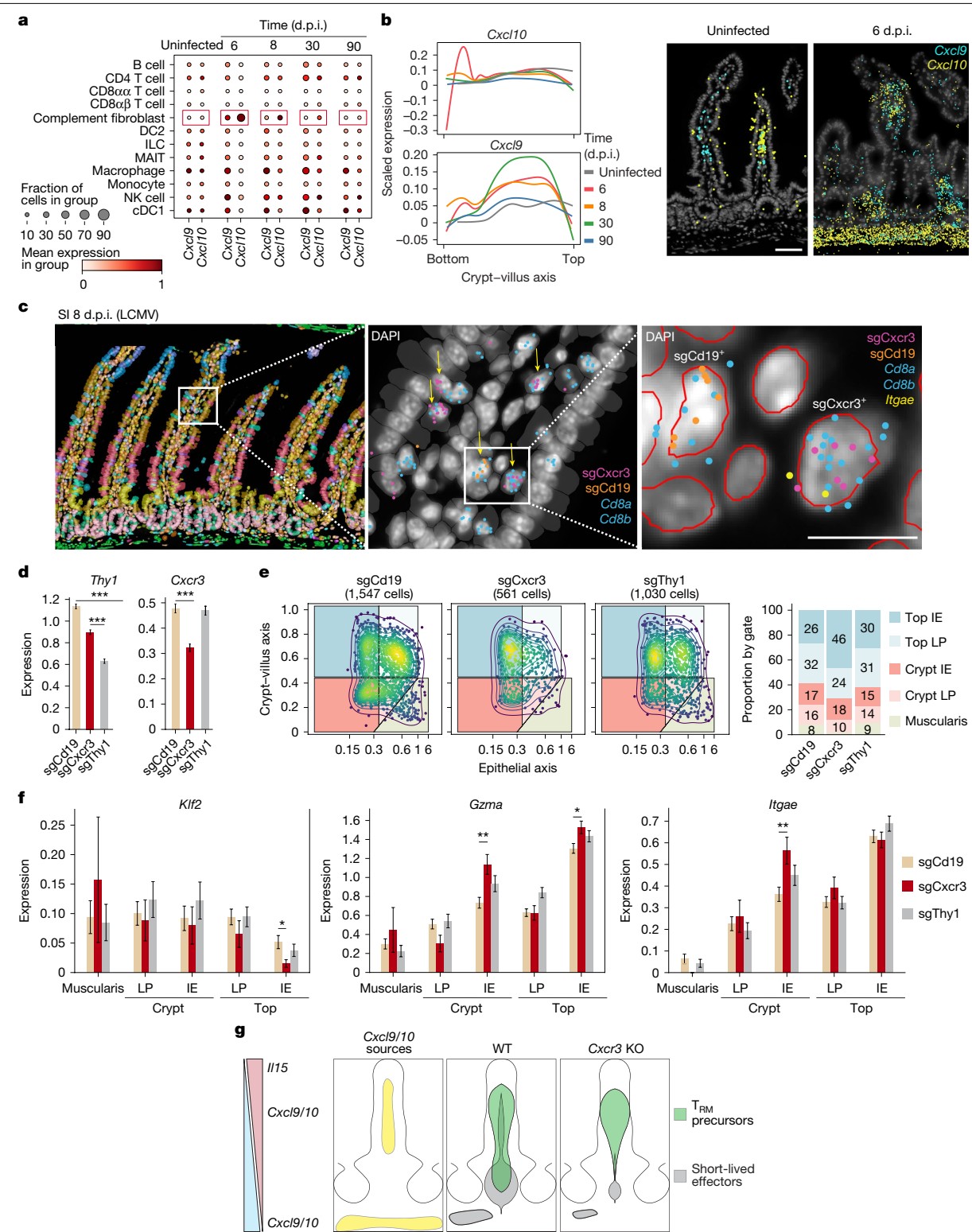

**Fig. 4 | CXCR3 promotes the early accumulation of short-lived effector cells in the lamina propria, lower villus area and muscularis. a**, A dot plot of *Cxcl9* and *Cxc10* expression in the indicated cells for time courses with and without infection. **b**, Gene expression trends for *Cxcl9* and *Cxcl10* (*Cxcl9/10*) separated by timepoint (*n* = 2 biological replicates pooled), with representative expression depicted spatially at their positions on the villus. Scale bar, 50 μm. **c**, sgRNA-containing P14 CD8 T cells (yellow arrows, middle image) shown spatially in the intestinal villus at three levels of magnification as detailed by white rectangles from left to right. The left image is coloured by graph-based clustering. The red line (right image) indicates nuclear segmentation. Scale bar, 5 μm. **d**, The gene expression of each sgRNA-containing P14 CD8 T cell. *n* = 2 pooled biological

replicates with 1,548 sgCd19, 1,030 sgThy1 and 562 sgCxcr3 cells. Pairwise two-sided *t*-tests with Benjamini–Hochberg test correction. \*\*\**P* < 0.001. **e**, An IMAP representation of each sgRNA-containing P14 CD8 T cell population with annotated percentages in each gate. **f**, The gene expression of each sgRNA-containing P14 CD8 T cell split by spatial gate. *n* = 2 pooled biological replicates with cell numbers per gate shown in Supplementary Table 12. Pairwise two-sided *t*-test of the mean expression levels, with Benjamini–Hochberg correction. \**P* < 0.05, \*\**P* < 0.01. Data are presented as mean ± s.e.m. (**d**,**f**). **g**, The proposed mechanism of CXCR3-dependent CD8 T cell villus distribution. KO, knockout; WT, wild type.

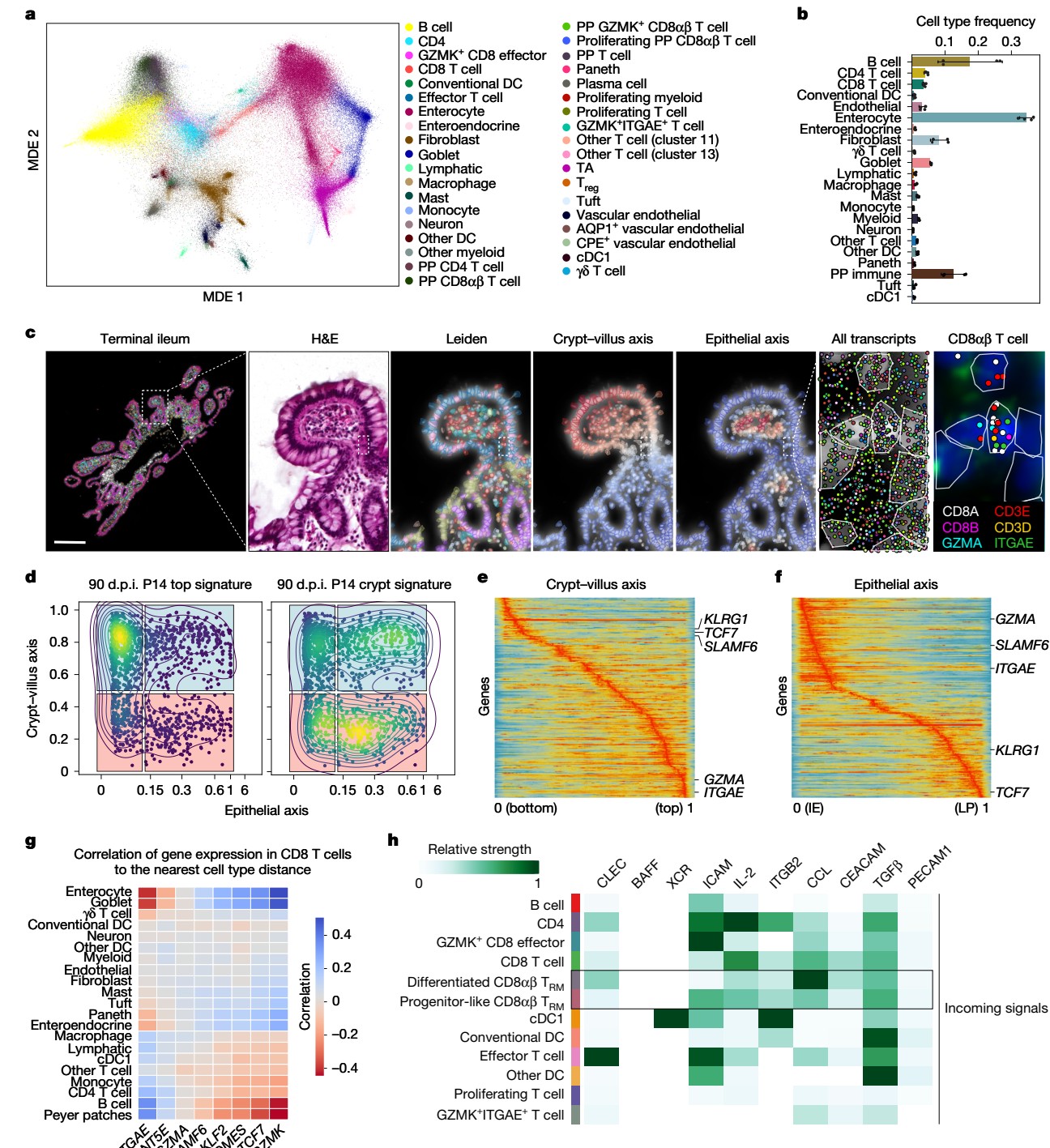

**Fig. 5 | CD8 T cell phenotypic diversity in the human ileum is spatially imprinted. a,b**, Spatial transcriptomics of two human terminal ileum sections using 10x Xenium: joint MDE embedding coloured by cell type (**a**) and mean relative frequencies of each cell type pooled across all sections (**b**). Data show mean ± s.e.m. Two adjacent sections per donor. **c**, An overview of the human Xenium data. From left to right: terminal ileum with cell masks coloured by cell type; villus magnification showing H&E staining and Xenium DAPI staining with cell boundaries overlaid and coloured by Leiden cluster, crypt–villus axis and epithelial distance; further zoom-in showing Xenium DAPI with cell masks and detected transcripts and select transcripts overlaid over DAPI staining (1 representative of $n = 1,423$ CD8αβ T cells). Scale bar, 300 μm. **d**, An IMAP representation of CD8αβ T cells coloured by kernel density estimates weighted by mouse P14 cell signatures at the top of the villus (left) or or crypt (right). The human IMAP gates define the top villus (blue) and crypt (red), split into intraepithelial (left) and lamina propria (right). CD8αβ T cells were pooled

across all replicates ($n = 1,423$); Peyer's patches (PP) excluded. **e,f**, The convolved gene expression of CD8αβ T cells along the crypt–villus axis (**e**) and epithelial axis (**f**). All human samples were pooled ($n = 2$ donors, two adjacent sections each), excluding Peyer's patches. **g**, The expression of select genes in CD8 T cells are Spearman rank correlated with distances to other cell types. Red indicates that expression increases when CD8 T cells are near, whereas blue indicates that expression decreases. Correlations calculated per sample ($n = 4$); mean coefficient shown. **h**, A heat map showing the top pathways contributing to incoming signalling of different immune cell groupings. Relative strengths calculated using spatial CellChat on all human samples. The heat map was column-normalized across all cell subtypes; only specific immune subtypes are shown. CD8αβ T cells grouped as effector or stem-like on the basis of enrichment of mouse-derived UCell signatures. Enrichment $z$-scored before classifying CD8αβ T cells. CLEC, C-type lectins; $T_{reg}$, T regulatory cell.

determine to what extent TGFβ presentation by specific cell types is needed to maintain intraepithelial CD8 T cell populations.

Newly infiltrated effector CD8 T cells responding to an infection in the SI display high interstitial motility, which becomes more restricted as cells differentiate into $T_{RM}$ cells[35]. Thus, signals received during the high-motility phase, such as short- and long-range sensing of chemokines and cytokines, might ultimately condition the destination of $T_{RM}$ cells in the tissue. Increased chemotactic signals, such as *Cxcl9* and *Cxcl10*, at the base of the muscularis might be responsible for the accumulation of short-lived effector cells observed in the few days after infection. The presence of inducible chemotactic areas rather than stable gradients in the villus creates a network that can steer incoming new CD8 T cell infiltrates while maintaining the polarization of tissue-resident populations.

Our in vivo approaches, the optically encoded perturbations and computational analyses to systematically study the spatial positioning of $T_{RM}$ cells in the SI could also be applied to other tissues with functional repetitive structures, such as the nephron in the kidney, glandular structures or hepatic lobule and similarly provide a framework for the study of other immune cell populations in tissues. Insights from this approach inform avenues to selectively target tissue-specific immune populations, functional subsets and the interactions driving immune cell function in a given tissue[30,36,37].

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

# Methods

## Mice

Mice were maintained in specific-pathogen-free conditions at a temperature between 18 °C and 23 °C with 40–60% humidity and a 12 h–12 h light–dark cycle in accordance with the Institutional Animal Care and Use Committee of the University of California, San Diego (UCSD). All mice were of C57BL/6J background and bred at the UCSD or purchased from The Jackson Laboratory. $R26^{creERT2}$ (stock no. 008463, The Jackson Laboratory), $Tgfbr2^{fl/fl}$ (stock no. 012603, The Jackson Laboratory), P14 Cas9–eGFP (stock no. 026179, The Jackson Laboratory), P14 and CD45.1 congenic mice were bred in-house. Male recipient mice were used for adoptive transfer experiments, and females were used as P14 CD8 T cell donors. In the spatial pooled CRISPR knockout experiment, a male was used as a donor. To delete floxed alleles using Cre-ERT2, 1 mg of tamoxifen (Cayman Chemical) emulsified in 100 µl of sunflower seed oil (Sigma-Aldrich) was administered by intraperitoneal injection for five consecutive days to P14 $R26^{creERT2}Tgfbr2^{WT}$ (WT) and P14 $R26^{creERT2}Tgfbr2^{fl/fl}$ (TGFβRII knockout) mice before P14 CD8 T cell isolation. All mice were between 1.5 and 6 months old at the time of infection and randomly assigned to experimental groups. No statistical methods were used to predetermine sample sizes, but our sample sizes are similar to those reported in previous publications from our laboratory and others. No blinding was performed during mouse experiments. Investigators were not blinded to group allocation during data collection and/or analysis. Mice were fed ad libitum for the specified amount of time. All animal studies were approved by the Institutional Animal Care and Use Committee at the UCSD and performed in accordance with UCSD guidelines.

## Adoptive cell transfer of naive P14 CD8 T cells and LCMV infection in mice

A total of $5 \times 10^4$ female naive P14 CD8 T cells isolated by negative enrichment using magnetic activated cell sorting (MACS) and resuspended in PBS were transferred intravenously into congenically distinct male recipient mice. Recipient mice were subsequently infected intraperitoneally with $2 \times 10^5$ plaque-forming units of the Armstrong strain of LCMV.

## Preparation of single-cell suspensions for flow cytometry

The isolation of CD8 T cells was performed as described previously[38]. SI intraepithelial lymphocytes and lamina propria lymphocytes were prepared by removing Peyer's patches and the luminal contents from the entire SI. The SI was then cut longitudinally into 1 cm pieces and incubated at 37 °C for 30 min in HBSS with 2.1 mg ml$^{-1}$ sodium bicarbonate, 2.4 mg ml$^{-1}$ HEPES, 8% bovine growth serum and 0.154 mg ml$^{-1}$ dithioerythritol (EMD Millipore). The samples were passed through a 70-µm cell strainer, and the supernatant constituted the intraepithelial lymphocyte compartment of the SI. The remaining tissue fragments of the SI were further incubated in RPMI with 1.2 mg ml$^{-1}$ HEPES, 292 µg ml$^{-1}$ L-glutamine, 1 mM MgCl$_2$, 1 mM CaCl$_2$, 5% fetal bovine serum (FBS) and 100 U ml$^{-1}$ collagenase (Worthington) at 37 °C for 30 min. After enzymatic incubation, samples were filtered through a 70-µm nylon cell strainer (Falcon). Tissue preparations were separated on a 44%/67% Percoll density gradient.

The following antibodies were used for flow cytometry: CD3 (PE clone 145-2C11, eBioscience 12-0031-83, 1:200 dilution), TCR αβ (APC clone H57-597, eBioscience 17-5961-83, 1:200 dilution), NK1.1 (FITC clone PK136, eBioscience 11-5941-81, 1:400 dilution), CD19 (PerCP-Cy5.5 clone eBio1D3, eBioscience 45-0193-82, 1:200 dilution), CD8b (BV421 clone H35-17.2, eBioscience 48-0083-82, 1:400 dilution), CD45.1 (BV510 clone A20, BioLegend 110741, 1:200 dilution), TCR γδ (BV711 clone GL3, BioLegend 118149, 1:200 dilution), CD4 (BV786 clone GK1.5, BioLegend 100453, 1:400 dilution), CD8a (PE-Cy7 clone 53-6.7, eBioscience 25-0081-82, 1:400 dilution), fixable viability dye (APC-Cy7 eBioscience 65-0865-14, 1:1,000), CD11b (PE clone M1/70, eBioscience

12-0112-82, 1:200 dilution), CD11c (APC clone N418, BioLegend 117310, 1:200 dilution), Ly6C (FITC clone AL-21, BD 553104, 1:200 dilution), Ly6G (PerCP-Cy5.5 clone 1A8, BioLegend 127615, 1:200 dilution), XCR1 (BV421 clone ZET, BioLegend 148216, 1:200 dilution), CD45 (BV510 clone 30-F11, BD 561487, 1:200 dilution), F4/80 (BV711 clone BM8, BioLegend 123147, 1:200 dilution), MHC II (BV786 clone M5/114.15.2, BioLegend 107645, 1:200 dilution) and B220 (PE-Cy7 clone RA3-6 B2, BioLegend 103222, 1:200 dilution).

## Sample preparation for histology of the mouse SI

For fresh frozen samples, mouse SIs were collected, retaining the proximal–distal orientation. After discarding the first 3 cm proximal section, approximately 10 cm of mouse proximal SI (containing duodenum and proximal jejunum) was rinsed in ice-cold PBS and the lumen contents were flushed with 20 ml of ice-cold PBS using a gavage syringe. The SI was then loaded onto a 3.25-mm-diameter knitting needle premoistened with cold PBS and placed directly on thick blotting paper. The Mouse Intestinal Slicing Tool[39] was used as a guide for the scalpel to cut the intestine longitudinally along the knitting needle. The Mouse Intestinal Slicing Tool and needle were removed, and the SI was spread open and rolled using a wood autoclaved round toothpick, embedded in OCT in plastic moulds, frozen in dry ice (Tissue-Tek Cryomold) and kept at −80 °C until cryosectioning. For fixed frozen samples, the opened cleaned SIs were fastened to blotting paper by minutien pins in each corner and fixed in 4% paraformaldehyde solution in PBS at 4 °C for 16 h, followed by incubation in 70% ethanol at 4 °C for a minimum of 3 h. The SI samples were then rolled using a wood autoclaved round toothpick, snap-frozen in OCT in plastic moulds for cryosection (Tissue-Tek Cryomold) and kept at −80 °C until processing. For formalin-fixed paraffin-embedded (FFPE) samples, fixed SIs were rolled, mounted on 2% agar round moulds and placed on histology cassettes for paraffin embedding.

## Histology and immunofluorescence staining of fresh frozen mouse tissues

After OCT block equilibration at −20 °C, 10-mm slices were obtained using a cryostat, mounted on glass slides, dried for 20 min at −20 °C and fixed in ice-cold acetone at −20 °C for 20 min. After fixation, the slides were dried briefly at room temperature and stored at −80 °C until stained or used immediately. For staining, the slides were equilibrated at room temperature, washed in 4 °C PBS twice for 5 min, blocked in serum-free blocking reagent overnight (Dako) at 4 °C, followed by staining with CD45.1–AF594 (BioLegend, clone A20, 110756, 1:50 dilution) and E-cadherin-APC (BioLegend, clone DECMA-1, 147312, 1:200 dilution) and CD8a–FITC (BioLegend, clone 53-6.7, 35-0081-U500, 1:50 dilution) diluted in antibody diluent solution (Dako, S080983-2) overnight at 4 °C, stained with DAPI and mounted with coverslips using Vectashield Vibrance Antifade mounting medium (VectorLabs, H-1700). Images were acquired on an Olympus VS200 Slide Scanner (UCSD Microscopy Core) or on a Zeiss LSM700 confocal microscope. P14 CD8 T cell distances for IMAP representation over time were quantified using a groovy script on QuPath (https://github.com/Goldrathlab/Spatial-TRM-paper).

## Single-nucleus RNA-seq of mouse SI

Female CD45.1$^+$CD8$^+$ P14 T cells were adoptively transferred into male CD45.2$^+$ recipients ($1 \times 10^5$ cells per mouse) 30 min before infection with the Armstrong strain of LCMV. At 28 d.p.i., the mice were euthanized and the SI was dissected, flayed and washed in cold PBS; Peyer's patches were excised from the SI. The SI was divided into three equal sections designated as the proximal, middle and distal SI. Tissue sections were cut into pieces of approximately 3 mm and flash-frozen in liquid nitrogen for 2 min. Nucleus isolation was performed with 10x Genomics Chromium Nuclei Isolation Kit per the manufacturer's instructions. In brief, 30–50 mg of flash-frozen tissue per sample was dissociated

with a pestle, incubated for 10 min on ice and washed. Dissociated tissue was passed through a nucleus isolation column, and flowthrough nuclei were washed in debris removal buffer and wash and resuspension buffer. Nuclei were quantified with a Nexcelom Bioscience Cellometer. For maximum targeted recovery, 40,000 nuclei per sample were loaded for Gel Bead-In Emulsion generation. Samples were processed by the Chromium Next GEM Single Cell 3′ HT Dual Index v3.1 protocol and sequenced to a depth of 550 million read pairs per sample (around 23,000 read pairs per nucleus) on a NovaSeq 6000 system (Illumina).

## Spatial transcriptomics analysis using whole-genome spatial transcriptomics (VisiumHD)

A 7-µm-thick section from an 8 d.p.i. FFPE sample was placed in a histology slide, dried at 42 °C for 3 h, dehydrated overnight, baked at 60 °C for 30 min and deparaffinized following the VisiumHD standard protocol (CG000685). The mouse reagents (VisiumHD, Mouse Transcriptome, 6.5 mm, four reactions, PN-1000676) were used to obtain high-quality H&E images, perform hybridization, RNA removal and probe amplification, carry out analyte transfer to the VisiumHD slide using the Cytassist with the 2.0 software, and generate libraries, which were sequenced following the recommended configuration in a NovaSeq 6000 Illumina instrument. Binary base call files were demultiplexed into FASTQ files using spaceranger mkfastq followed by spaceranger count to generate the spatial representation of gene counts by matrix at 2-mm-resolution and 8-mm-resolution binning. A Cellpose[40] cell-segmentation model was fine-tuned to segment nuclei in the high-resolution VisiumHD H&E image. A cell-by-gene matrix was created by summing transcript counts in all 2 mm bins overlapping each cell in the segmentation mask. gimVI was used to jointly embed cells from an 8 d.p.i. Xenium dataset with those from 8 d.p.i. VisiumHD[41]. The crypt–villus axis and cell types for VisiumHD were imputed by using the three closest Xenium neighbours in the joint latent space. The crypt–villus axis values were calculated as the mean of the neighbours' values, whereas the cell type was determined by the most frequently occurring type among the neighbours. Further details of the downstream analysis of the VisiumHD dataset are provided at GitHub (https://github.com/Goldrathlab/Spatial-TRM-paper).

## Spatial transcriptomics analysis using multiple error-robust fluorescence in situ hybridization

Fresh frozen tissue was sectioned according to standard histology procedures to a thickness of 10 µm. The sections were adhered to the MERSCOPE slides (Vizgen, 20400001) coated with fluorescent beads by storing them in the cryostat at −20 °C for at least 5 min. The samples were fixed in 5 ml of fixation buffer containing 4% paraformaldehyde in 1× PBS that was preheated to 47 °C and incubated for 30 min at 47 °C, according to the MERSCOPE Quick Guide Modified Fixation for Fresh Frozen Samples. The samples were then washed three times with 5 ml PBS, 5 min each time. The samples were permeabilized in 5 ml 70% ethanol at 4 °C in parafilm-sealed dishes overnight and stored in these conditions for up to a month. Samples were then prepared according to Vizgen's protocols, starting from the cell-boundary protein-staining step. The samples were hybridized with a custom 500-gene panel that included 5 sequential genes, as well as several blank barcodes that do not encode a gene and used for measuring the background signal. To clear the samples of lipids and proteins that interfere with imaging, 5 ml of Clearing Premix (Vizgen, 20300003) was mixed with 100 µl of proteinase K for each sample, and the samples were placed at 47 °C in a humidified incubator overnight (or for a maximum of 24 h) and then moved to 37 °C. The samples were stored in the clearing solution provided with the MERSCOPE kit in the 37 °C incubator before imaging for up to a week. The samples were imaged on the MERSCOPE according to the MERSCOPE Instrument User Guide. Seven 1.5-µm-thick z planes were imaged for each field of view at 60× magnification. Images were decoded to RNA spots with *xyz* and gene ID using the Merlin software

of Vizgen. Cell segmentation was performed using the Cellboundary algorithm, relying on the Cellboundary 2 stain and DAPI nuclear seeds.

## Spatial transcriptomics analysis using 10x Xenium

FFPE tissues were sectioned to a thickness of 5 µm onto a Xenium slide, followed by deparaffinization and permeabilization following the 10x user guides CG000578 and CG000580. Probe hybridization, ligation and amplification were done following the 10x user guide CG000582. In brief, probe hybridization occurred at 50 °C overnight with a probe concentration of 10 nM using a custom gene panel designed to detect 350 different mRNAs. After stringent washing to remove unhybridized probes, probes were ligated at 37 °C for 2 h. During this step, a rolling circle amplification primer was also annealed. The circularized probes were then enzymatically amplified (2 h at 37 °C), generating multiple copies of the gene-specific barcode for each RNA binding event. After washing, background fluorescence was quenched chemically. The sections were placed into an imaging cassette to be loaded onto the Xenium Analyzer instrument following the 10x user guide CG000584.

## Spatial data processing

For 10x Xenium spatial transcriptomics data, nuclei were segmented using a fine-tuned Cellpose[40] model on maximum-projected DAPI-staining images. Baysor[42] was used to predict cell-boundary segmentations using transcript identity and positions, and the prior Cellpose nuclei segmentation or Cellboundary 2 segmentation for 10x Xenium or MERSCOPE, respectively. The parameter prior-segmentation-confidence was set to 0.95 for 10x Xenium and to 0.9 for MERSCOPE, and min-molecules-per-cell was set to the median nucleus transcript count (https://github.com/Goldrathlab/Spatial-TRM-paper#preprocessing). Baysor segmentations containing no nuclei were filtered out, and segmentations containing multiple nuclei were split by assigning transcripts to the nearest nucleus centroid in the segmentation boundary. Cell boundaries are visualized as polygons using the alphashape Python package. All cells with $n < 8$ nuclear transcripts, $n < 20$ total transcripts or $n > 800$ total transcripts were filtered out before downstream processing. To integrate spatial replicates into a joint embedding, scVI[43] was used with n_layers of 2 and n_latent of 30. The joint embedding was projected into two-dimensional space using scVI.model.utils.mde. Leiden clustering was performed on the scVI learned embeddings using scanpy.tl.leiden with a resolution of 1, and every Leiden cluster was further subclustered at a resolution of 1.2. Celltypist[44] and GeneFormer[45] were used for a first-pass cell type assignment, with further manual refinement based on the expression of cell type marker genes to define cell types in a class > type > subtype hierarchy. The Anndata[46] format was used for all further processing. To align histology images with Xenium spatial coordinates, we used an OpenCV Oriented FAST and Rotated BRIEF[47] object to detect key points in the DAPI channel of both histology and Xenium images. These key points were then matched using an OpenCV DescriptorMatcher, enabling the computation of a homography matrix based on the top matches using cv2.findHomography. Subsequently, histology images across all channels underwent warping using this homography matrix with cv2.warpPerspective. To align H&E images with Xenium spatial coordinates, we trained a pix2pix generative adversarial network[48,49] to predict DAPI images from H&E as an intermediate state before finding key points and matching, as previously mentioned. To visualize mouse transcriptional signatures onto human datasets, all ($n = 8$) mice time course samples were used to find the top 15 differentially expressed genes (Scanpy rank_genes_groups and method = 'wilcoxon') between P14 cells gated to the crypt and P14 cells gated to the top of villi. Human homologues of these 15 genes are defined as UCell[50] signatures and mapped to human CD8αβ T cells. Human CD8αβ T cells are positioned on IMAPs and coloured by their enrichment of the (left) top mouse signature and (right) crypt mouse signature. All codes to analyse the spatial datasets are available at https://github.com/Goldrathlab/Spatial-TRM-paper.

## Histological staining of mouse intestinal tissue after Xenium analysis

After the Xenium run, slides were kept hydrated in PBS-T (0.05% Tween-20 in PBS) at 4 °C. For post-Xenium immunofluorescence staining, PBS-T was removed and samples were blocked using universal blocking reagent CAS-Block and stained with anti-CD8a antibody (Abcam, EPR21769) 1:50 dilution) overnight at 4 °C, followed by three washes with PBS. Anti-rabbit AF594 secondary antibody (Invitrogen, A-11012, 1:200 dilution) in Dako antibody diluent was then added for 1 h at room temperature in the dark, followed by three washes with PBS. The slides were then stained using WGA–FITC followed by DAPI staining and then mounted with a coverslip using Vectashield mounting medium. The slides were dried for 1 h at room temperature in the dark before imaging with an Olympus VS200 Slide Scanner at 20×. The slides were then soaked in PBS at 4 °C overnight to dismount the coverslip and subsequently washed three times in PBS and twice in ddH$_2$O before proceeding with H&E staining. The coverslip was mounted using xylene-based mounting medium (Cytoseal XYL Epredia), and the slides were dried for 1 h and imaged used VS200 Slide Scanner at 20×.

## Human samples

Deidentified human FFPE samples from healthy participants were acquired from the San Diego Digestive Diseases Research Center. Slices of 5 mm were obtained using a microtome, deparaffinized and H&E stained according to common histology practices or processed according to the 10x Xenium protocol.

## Human participants and ethical statement

The Human Research Protection Programs at the UCSD reviewed and approved the protocol, including a waiver of consent. Submucosal ileal biopsies were obtained from patients who underwent colonoscopies to rule out inflammatory bowel diseases. Ileal biopsies were evaluated by a pathologist and found to be normal without histological inflammation. Samples were deidentified and processed for the study.

## Gene panel design for probe-based spatial transcriptomics profiling of mouse SI

The gene panel design made use of Predictive and Robust Gene Selection for Spatial Transcriptomics (PERSIST)[19], a deep learning model that uses single-cell RNA sequencing (scRNA-seq) data to learn a binary mask for the identification of a subset of genes that best predict cell type from gene expression (supervised) or for the reconstruction of whole-transcriptome gene expression (unsupervised). For the Xenium mouse 350-gene panel, 79 SI canonical cell type marker genes were compiled from existing literature and *Xist*, a marker for transferred female P14 CD8 T cells. An additional 158 genes from a Nichenet database[21] of ligand–receptor pairs were included. Next, supervised PERSIST was run on an immune-enriched gut scRNA-seq dataset[51] with the previously compiled set of genes as prior information, adding an additional 70 genes. Finally, supervised PERSIST was run on a SI scRNA-seq dataset to capture 59 cell type marker genes for 350 total targets. To create the Xenium human 422-gene panel, we created a base set of canonical immune marker genes, ligand–receptor pairs, spatially differentially expressed genes in mouse P14 CD8 T cells, and the 10x Genomics base human colon panel totalling 343 genes. Using this set as prior information for PERSIST, and a reference human immune cell scRNA-seq dataset[52], unsupervised PERSIST filled in the remaining 79 genes. To create the 494-gene panel for MERSCOPE, we compiled 18 published bulk RNA-seq datasets profiling different immune populations in different disease settings[5,53–66], including the ImmGen RNAseq database (immgen.org), and manually curated metadata attributes for each sample for the following categories: cell. main, cell.type, cell.subtype, cell.state, model and tissue. The annotated integrated dataset compilation was used as input for feature selection with XGBoost[67]. Top genes in each attribute were incorporated as the panel backbone. Furthermore, gene markers of intestinal cell markers from PanglaoDB and ref. 68, ImmGen CITE-seq protein markers and 159 genes from the ligand–receptor database NicheNet[21] were added. A total of 494 final genes passed the quality-check filtering of Vizgen for transcript length and expression levels. The Xenium mouse 480-gene panel was designed using the 350-gene panel but removing genes that were minimally informative based on the time course data and adding genes that were relevant based on manual curation and a prioritization score that evaluates their ability to differentiate clusters in a scRNA-seq dataset containing most cell types of the mouse SI[69].

## Defining structural axes in spatial transcriptomics datasets of the SI

To calculate the longitudinal axis, a multiline segment was initially labelled across the base of the basal membrane, using labelme[70], starting from the outermost section of the roll. For each cell, the nearest neighbour was calculated from the set of all locations on the multiline segment positioned closer to the centre of the SI roll. The relative position of the nearest neighbour along the length of the entire multiline segment was used as the longitudinal position. In datasets with atypical morphology, longitudinal axis values for each cell were predicted using a deep neural network trained on a feature space of transcriptional neighbourhood decomposition latent factors and multiline segments marking the base of the basal membrane, the top of each villus and the middle of each villus. Transcriptional neighbourhood decomposition was performed using Scikit-learn[71] non-negative matrix factorization on a matrix of the summed transcript count values for the ten nearest neighbours of each cell, calculated with a SciPy[72] $K$-dimensional tree, to create a transformed data matrix $W$ with 15 latent factors. To calculate the crypt–villus axis, a fine-tuned Cellpose model was used to segment villi on the basis of WGA staining in Xenium samples and cell spatial positions in MERSCOPE samples. The distance between each cell and the base of the basal membrane was calculated as before, and these values were $z$-score scaled among cells in the same segmented villus. For datasets with poorly defined morphology, crypt–villus axis values for each cell were predicted using a TensorFlow v2.18.0 (https://www.tensorflow.org/) deep neural network trained on a feature space comprising a decomposition of latent factors for epithelial and stromal transcriptional neighbourhoods. Transcriptional neighbourhood decomposition was performed by fitting a non-negative matrix factorization model on a matrix of the summed transcript count values (10, mouse; 30, human) for the nearest epithelial and stromal neighbours of each cell in all datasets to avoid the influence of variability in immune populations at different infection states. Crypt–villus axis predictions for each cell were smoothed over their nearest 150 neighbouring cell predictions in human data. To calculate the epithelial axis, the mean distance from each cell to the five nearest epithelial cells was divided by the mean distance to the five nearest cells of any cell type. The resulting values were $z$-score scaled and clipped at an upper bound to align epithelial distances between the villus and basal membrane.

## Generation of IMAPs and transcriptional IMAPs

The epithelial IMAP axis values were computed through a biexponential transformation applied to the clipped epithelial axis values across all cells. Each cell was positioned on the IMAP according to its corresponding crypt–villus axis and transformed epithelial axis values. The density in the scattered point cloud was visualized using colour-mapped scipy. stats.gaussian_kde values, with density lines overlaid using seaborn. kdeplot for enhanced clarity and interpretation. Gate boundaries were drawn manually to distinguish the muscularis, villus crypt and villus top by observing the IMAP locations of cell types known to localize to each region. Transcriptional IMAPs were coloured by adding an array of gene expression counts as a point weight parameter to the scipy. stats.gaussian_kde function. Similarly, gene signature IMAPs were coloured using the squared UCell signature enrichment scores as the

point weight parameter. In human IMAPs, signature and gene expression point weights were squared to overcome bias in CD8αβ[+] physical density in IMAP visualization.

## Statistical analysis

In Fig. 2e, significance cut-offs are set at absolute Spearman $\rho > 0.05$, an arbitrary threshold used to highlight differences between axes. The Spearman coefficients for each gene and their corresponding $P$ values are documented in Supplementary Table 3 across $n = 87,387$ P14 T cells. In Extended Data Fig. 7a, the top four differentially expressed genes per condition across P14 cells ($n = 4,135$ WT, $n = 4,161$ TGFβR2 knockout) are calculated using Wilcoxon testing with Benjamini–Hochberg correction in the function scanpy.tl.rank_genes_group. In Extended Data Fig. 7d,e, a non-parametric two-sample Kolmogorov–Smirnov statistic is used to calculate the significance of the difference between P14 cell type proximity distributions in wild-type and TGFβR2 knockout conditions. A cut-off of similarity is arbitrarily positioned at a Kolmogorov–Smirnov statistic of 0.08, corresponding to a corrected $P$ value of approximately $1 \times 10^{-12}$ to $1 \times 10^{-10}$ (Kolmogorov–Smirnov $P$ values vary with the number of samples in the compared distributions). Kolmogorov–Smirnov tests are documented in Supplementary Table 8. In Fig. 4d,f and Extended Data Fig. 1h, gene-expression counts were log-normalized, and two-sample, two-sided $t$-tests were used to test for significant differences in mean gene expression pairwise between perturbation groups at a significance level of 0.05 and Benjamini–Hochberg correction for $P$ values in each pairwise group. A similar approach was used in Extended Data Fig. 8e using raw detected barcode numbers without normalization. For each subplot in Extended Data Fig. 1g, we applied one-way and two-way analyses of variance, with Dunnett's method for multiple comparisons. In Extended Data Fig. 1j,k, we used a one-way analysis of variance followed by Tukey's honestly significant (HSD) difference tests to create confidence intervals. In Extended Data Fig. 9g, human gene expression counts were log-normalized before differentially expressed genes were calculated between human CD8αβ T cells gated to different regions using a two-tailed Wald test in the Python package diffxpy. $P$ values were adjusted using a Benjamini–Hochberg correction. Data are mean ± s.e.m. in all the figures.

## Cloning and making retrovirus

The LsgC plasmid was generated by using PCR to linearize the backbone of the LsgA plasmid and exclude the Ametrine reporter gene[73]. NEBuilder HiFi DNA Assembly was then used to insert a HA-tagged mCherry sequence, synthesized by Integrated DNA Technologies (IDT), into the open site. ChopChop was used to design sgRNAs targeting mouse *Cd19*, *Thy1* and *Cxcr3* (ref. 74). Forward (5′-CACCN) and reverse (5′-AAACN) primers forming the sgRNAs were synthesized by IDT. Each sgRNA was assigned a 388-bp or 390-bp barcode, containing seven or eight probe hybridization sites, respectively. These sites corresponded to one of the three lowest-expressed genes in SI P14 CD8 T cells from the SI spatial transcriptomics time course: *Muc5ac*, *Neurog3* and *Fer1l6* (Supplementary Table 7). Each 40-bp hybridization site was separated by at least 10 bp containing no homology with the mouse transcriptome. Barcodes were ordered from IDT as gBlocks Gene Fragments in tubes and were cloned into the LsgC vectors on the 3′ side of the mCherry using NEBuilder HiFi DNA Assembly. sgRNAs were inserted into their corresponding LsgC-barcode vector by digesting BbsI restriction sites, followed by room-temperature ligation (T4 DNA ligase, NEB) with the annealed forward and reverse sgRNA primers. LsgC barcodes were transformed into DH5α competent cells (Thermo Fisher). The three unique LsgC barcodes were separately transfected into platinum-E (PlatE) cells (Cell Biolabs, no authentication or mycoplasma contamination test) to make retrovirus. One day before the transfections, $2.5 \times 10^5$ PlatE cells were plated on 10-cm dishes in PlatE medium (89% DMEM, 9% FBS, 1% HEPES 1 M, 1% penicillin-streptomycin-glutamine (PSG) (100×, Thermo Fisher) and

0.1% 2-mercaptoethanol (BME)). PlatE cells were transfected using a mix containing 10 μg of LsgC-barcode vector, 5 μg of PCL-Eco (Addgene, 12371) and TransIT-LT1 (Mirus). Retrovirus was collected at 48 h and 72 h after transfection and stored at −80 °C until use.

## Transductions and spatial transcriptomics with pooled perturbations

One day before transduction, splenic P14 CD8 T cells were isolated from a Cas9–eGFP[+] donor mouse through negative enrichment, and plated in T cell medium (TCM) (89% RPMI, 9% FBS, 1% HEPES 1 M, 1% PSG (100×, Thermo Fisher) and 0.1% BME) containing 1:500 anti-CD3e (Fisher Scientific, 50-112-9591) and CD28 (Fisher Scientific, 50-112-9711) on a six-well plate precoated with 1:30 goat anti-hamster IgG (H+L; Thermo Fisher Scientific) in PBS and stored at 37 °C overnight. Furthermore, an untreated six-well plate was coated with 15 μg ml$^{-1}$ of retronectin (Takara Bio) in PBS and stored in the dark at 4 °C overnight. During transduction, the retronectin was removed and the plates were coated with TCM and incubated at 37 °C for 30 min. After removal, the three treated plates were coated with a corresponding LsgC-barcode retrovirus over two successive 30-min incubations. Activated cells were resuspended in a 1:1,667 IL-2 in TCM mixture and spread equally across the three retronectin-treated plates. Corresponding retroviruses were added to each well, and the plate was centrifuged at 2,000 rpm for 40 min at 37 °C. The sgRNA knockouts were validated by performing flow cytometry on the transduced cells 2 days after transduction using anti-THY1.2 antibody (30-H12, BioLegend, 1:200 dilution) and anti-CXCR3 antibody (CXCR3-173, eBiosciences, 1:200 dilution) with mCherry[+] anti-CD8a[+] (53-6.7 BioLegend, 1:200 dilution) cells gated as successfully transduced. One day after transduction, mCherry[+] GFP[+] cells were sorted from each of the three transduced populations and pooled 1:1:1, then $1 \times 10^5$ cells were transferred into each recipient mouse. Recipient mice were immediately infected with LCMV and euthanized at 7 d.p.i. for spatial transcriptomics.

## Computational analysis of pooled perturbations in spatial transcriptomics

To stringently identify perturbed CD8 T cells in the spatial transcriptomics datasets, we identified all cells for which the sum of raw transcripts for *Cd8a*, *Cd8b1* and *Cd3e* was greater than or equal to 3, that had at least one barcode detected, that belonged to a CD8 T cell cluster and that had ≤1 *Muc2* transcript. *Fer1l6*, the pseudogene for sgCXCR3, shows low expression in goblet cells, requiring a stringent filtering of *Muc2* to minimize the possibility of goblet transcript bleed-over falsely marking a CD8 T cell as perturbed.

## Reporting summary

Further information on research design is available in the Nature Portfolio Reporting Summary linked to this article.

## Data availability

Sequencing data and spatial transcriptomics data are deposited in the Gene Expression Omnibus (GEO) at GSE279254 (VisiumHD), GSE279255 (single-nucleus RNA-seq) and GSE280895 (Xenium and MERSCOPE). Source data are provided with this paper.

## Code availability

Data-processing pipelines and the code to reproduce all figures for this project are available on GitHub (https://github.com/Goldrathlab/Spatial-TRM-paper).

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

**Acknowledgements** We thank P. Kosuri, N. Ahmed, C. Mah, C. Marinas, R. Maimon, E. Molina, Z. Mikulski, the members of the histology core at La Jolla Institute for Immunology, the Gene Yeo laboratory and the ImmGen consortium for helpful discussions and feedback on data analysis. This work was supported by R01AI179952 (A.W.G.), R37AI067545 (A.W.G.), P01AI132122 (A.W.G. and J.T.C.), R01AI072117 (A.W.G.), R01AI150282 (A.W.G.) and R01CA273432 (G.W.Y.). This work was supported by the National Institute of Diabetes and Digestive and Kidney Diseases (NIDDK)-funded San Diego Digestive Diseases Research Center P30DK120515 and US Department of Veterans Affairs Clinical Science Research and Development CRSD Service – I01 CX002396 (J.T.C.). The microscopy core is funded by NINDS (P30NS047101), and the NovaSeq 6000 at the sequencing core at LJI was acquired with S10OD025052 (S.B.). N.E.S. is supported by the NCI Predoctoral to Postdoctoral Fellow Transition (F99/K00) Award (K00CA222711). K.K.T. is supported by a National Institutes of Health (NIH) F31 Award (F31AI176705). Y.H.L. is supported by the Canadian Institutes of Health Research Doctoral Foreign Study Award and G.G. by a Cancer Research Institute Postdoctoral Fellowship (CRI4145).

**Author contributions** Conceptualization: M.R.-C., M.H. and A.W.G. Methodology: M.R.-C., A.M., A.F., V.L., B.B., J.T.C., M.H., H.N., S.A. and A.W.G. Investigation: M.R.-C., A.F., V.L., K.P.C., G.G., N.E.S., K.K.T., Y.H.L., W.H.W., C.S.I., M.H., P.P.C., S.Q., H.N. and A.M. Visualization: M.R.-C., M.H. and A.M. Funding acquisition: G.W.Y., J.T.C. and A.W.G. Project administration: M.R.-C., M.H. and A.W.G. Supervision: M.R.-C., M.H. and A.W.G. Writing—original draft: M.R.-C., A.M., M.H. and A.W.G. Writing—review and editing: M.R.-C., A.M., M.H. and A.W.G.

**Competing interests** M.R.-C. is a co-founder, scientific adviser and board member of TCura Bioscience, Inc. A.F. is a co-founder, CEO and board member of TCura Bioscience. B.B. receives consulting fees from Bristol Myers Squibb and Pfizer and research grants from Merck and Gilead. A.W.G. is a co-founder of TCura Bioscience, Inc. and serves on the scientific advisory board of ArsenalBio and Foundery Innovations. The other authors declare no competing interests.

**Additional information**
**Correspondence and requests for materials** should be addressed to Maximilian Heeg or Ananda W. Goldrath.

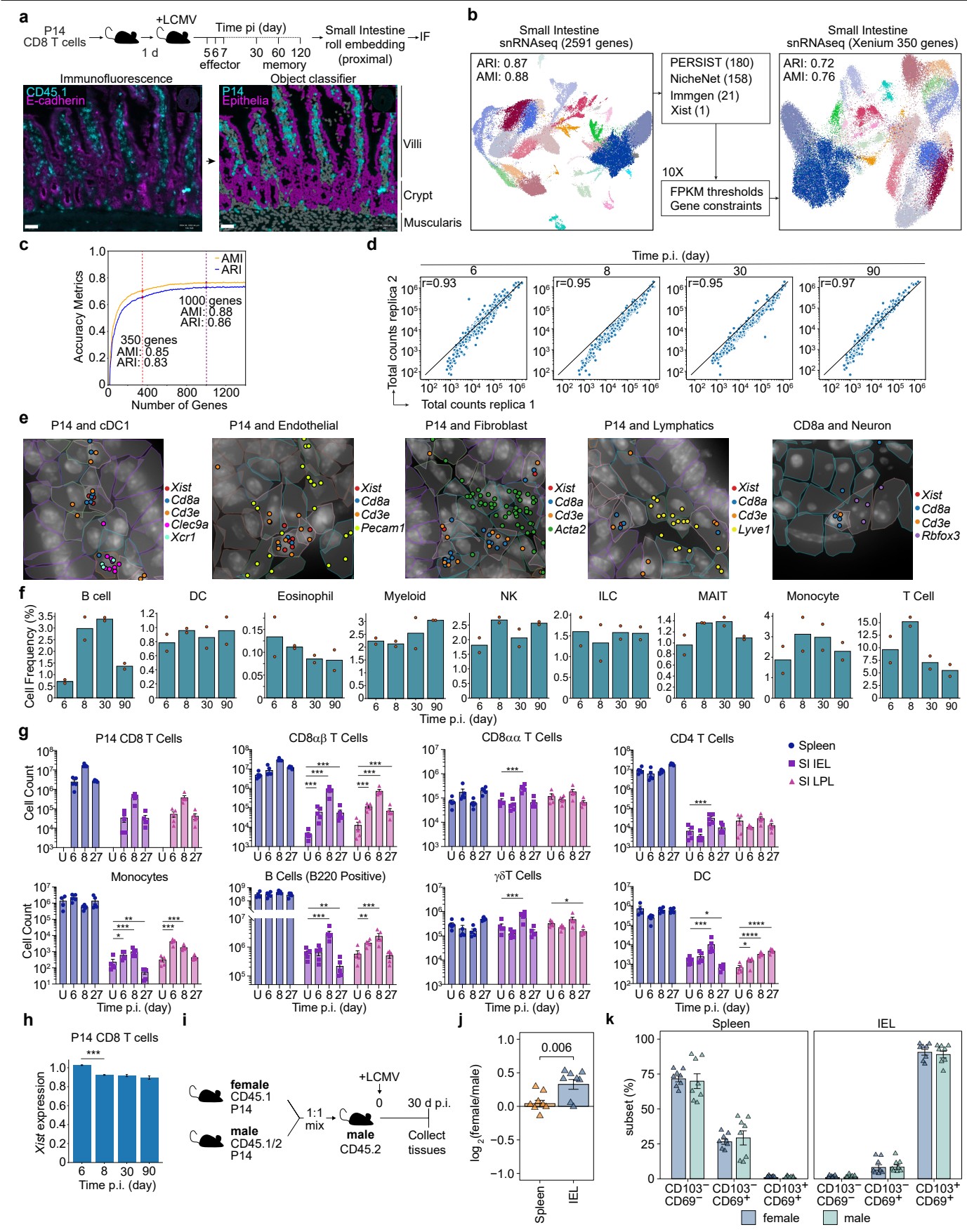

**Extended Data Fig. 1** | See next page for caption.

**Extended Data Fig. 1 | Related to Fig. 1. Targeted detection of LCMV-specific CD8 T cell responses in the mouse small intestine with spatial transcriptomics. a**, Schematic of the experimental workflow for mouse takedown at progressing timepoints post infection (p.i.) with LCMV. An object classifier in QuPath is used to identify P14 and epithelial cells from IF staining of intestinal sections. (1 representative field of view out of n > 20 similar) **b**, Diagram of the methodology used to design the Xenium mouse SI probe panel. Using snRNA-seq data from the mouse small intestine, a 350 gene set was designed to maximize Adjusted Rand Index (ARI) and Adjusted Mutual Information (AMI) scores of classifier-derived Leiden cluster predictions. **c**, Genes least informative for predicting cell type are continuously pruned using recursive feature elimination with ARI and AMI of classifier-derived Leiden cluster predictions calculated at each pruning step. **d**, Pearson residual correlations of total gene abundances between timepoint biological replicates. **e**, Snapshots from a Xenium spatial transcriptomics day 6 small intestine show unique cell types in close spatial proximity. Canonical cell type marker gene transcript positions are colored to show a (from left to right) P14 T Cell ($Xist^+Cd8\alpha^+Cd3e^+$) and cDC1 ($Clec9a^+Xcr1^+$), P14 T Cell and Endothelial ($Pecam1^+$), P14 T Cell and Fibroblast ($Acta2^+$), P14 T Cell and Lymphatic ($Lyve1^+$), and Cd8α T Cell ($Cd8\alpha^+Cd3e^+$) and Neuron ($Rbfox3^+$). Predicted cell segmentation boundaries are colored by the "Type" annotation. **f**, Cell type frequency percentages across n = 2 replicates per timepoint. **g**, Absolute cell numbers quantified by flow cytometry for the indicated cell types and time points after LCMV infection ($n = 5$). Two-way ANOVA with Dunnett's method. Bars indicate the mean +/- SEM. ***p-value < 0.001. **h**, $Xist$ expression within detected P14 CD8 T cells ($Xist^+$) at each time point ($n = 13000$, $n = 8412$, $n = 1155$, $n = 433$ cells in 6 dpi, 8 dpi, 30 dpi, and 90 dpi respectively across 2 biological replicates per time point). Pairwise t-tests on the mean expression values with Benjamini-Hochberg correction applied ***p-value < 0.001. **i**, Experimental design for a female:male P14 CD8 T cell transfer in B6 mice. **j**, Relative frequencies of female to male ratios in the indicated tissues 30 days p.i. Unpaired two-sided t-test with Tukey's HSD, *p-value < 0.05. Data are presented as the mean +/− Tukey's HSD confidence interval. **k**, Frequency of P14 CD8 T cells by sex and CD103 and CD69 expression analyzed by flow cytometry in the indicated tissues 30 days p.i. Unpaired t-test with Tukey's HSD. No significant p-values detected. Data are presented as the mean +/− Tukey's HSD confidence interval. 2 independent biological duplicates of n = 3 and n = 5 specimens are pooled (**j** and **k**). Panels **a** and **i** reproduced from ref. 5.

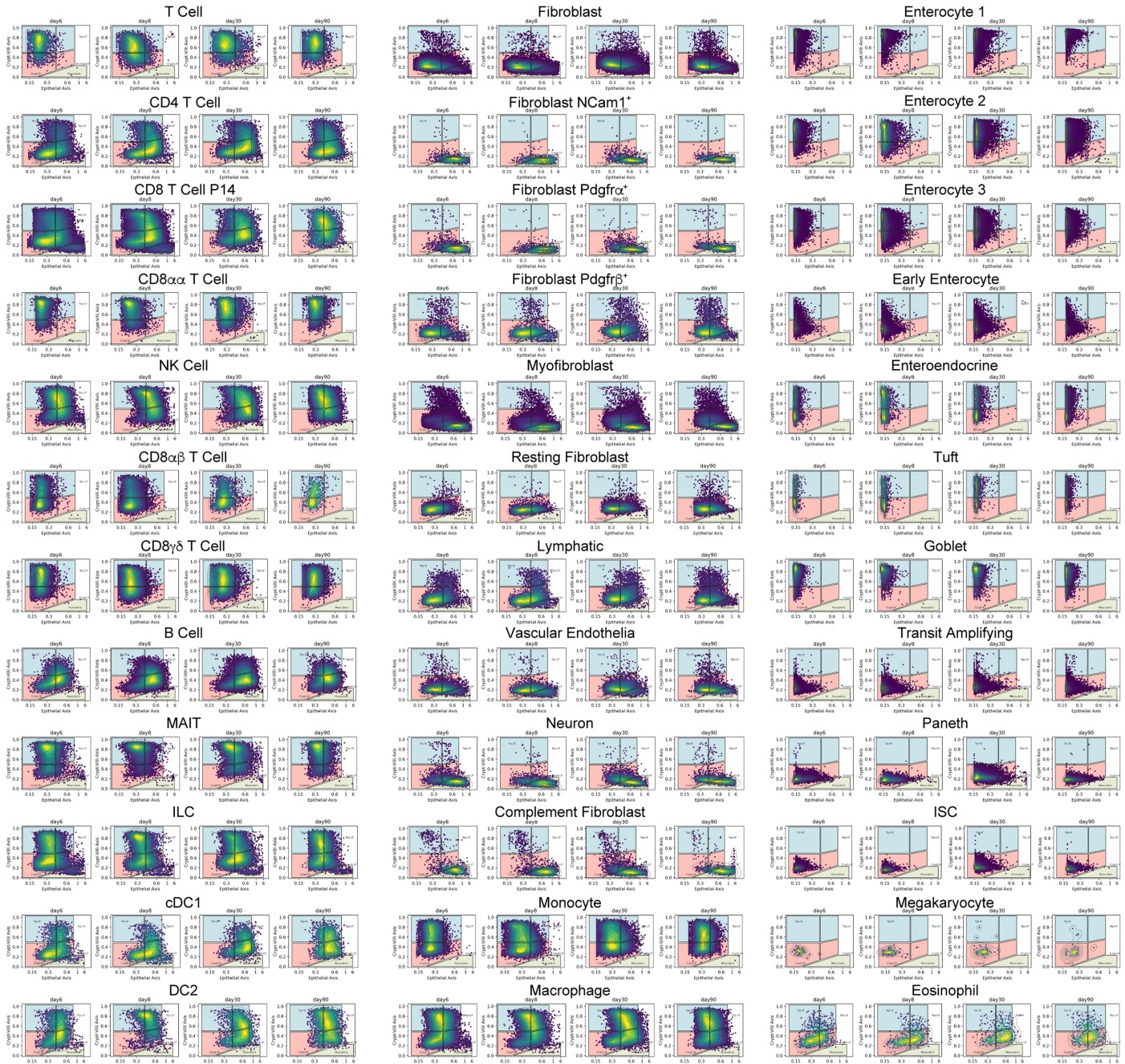

**Extended Data Fig. 2 | Related to Fig. 2. Spatial framework of the mouse intestinal villus shows cell positioning dynamics over time after an LCMV infection.** IMAPs of all cell subtypes from each time point (two biological replicates for each time point combined), with colored gates dividing the top IE, top LP, crypt IE, crypt LP, and muscularis.

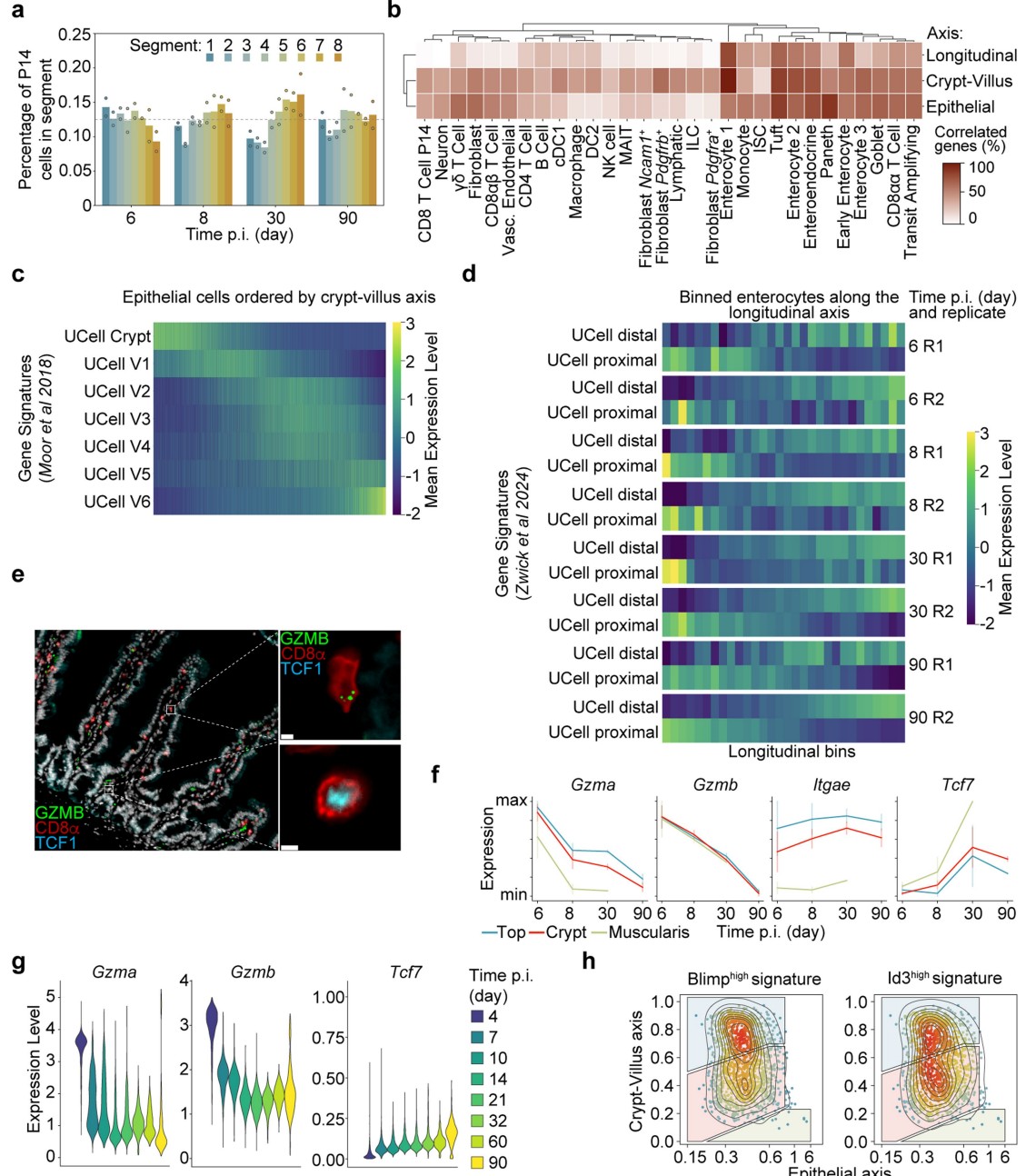

**Extended Data Fig. 3 | Related to Fig. 2. Spatial transcriptional patterning of mouse intestinal cells over the course of an LCMV infection revealed by spatial transcriptomics. a**, Frequency of P14 CD8 T cells present in binned segments of equal length of the longitudinal axis for n = 2 replicates per time point. **b**, Heatmap depicting the percentage of genes in every cell subtype correlated with each axis using all combined time course samples (n = 8). Heatmap colors indicate for all genes expressed in a particular cell type, few of them correlate with the corresponding axis (white), or most of them correlate with the corresponding axis (dark red). **c**, UCell enrichment of epithelial zonation signatures[22] in epithelial cells from pooled Xenium replicates ordered by crypt-villus axis position. Signatures are the top 30 differentially expressed, overlapping genes per zone. **d**, UCell enrichment

of proximal and distal epithelial signatures[15] in epithelial cells from each Xenium replicate, binned and ordered by longitudinal position. Zwick signatures include all overlapping genes with Spearman's ρ > 0.5 between expression and longitudinal segment order. **e**, IF staining of CD8α, TCF1 and GZMB of a mouse intestine 30 days after LCMV infection. Representative picture of 3 independent samples. Scale bars are 2 μm. **f**, Gene expression of indicated genes for P14 CD8 T cells grouped by the spatial gates shown in Fig. 2d over time; n = 2 replicates per time point. **g**, Gene expression of indicated genes for intestinal P14 CD8 T cells over time after an LCMV infection profiled by scRNA-seq. **h**, IMAPs of day 90 P14 CD8 T cells (one of two biological replicates) colored by Blimp1[high] differentiated and Id3[high] progenitor-like T[RM] signatures enrichment derived from Milner et al.

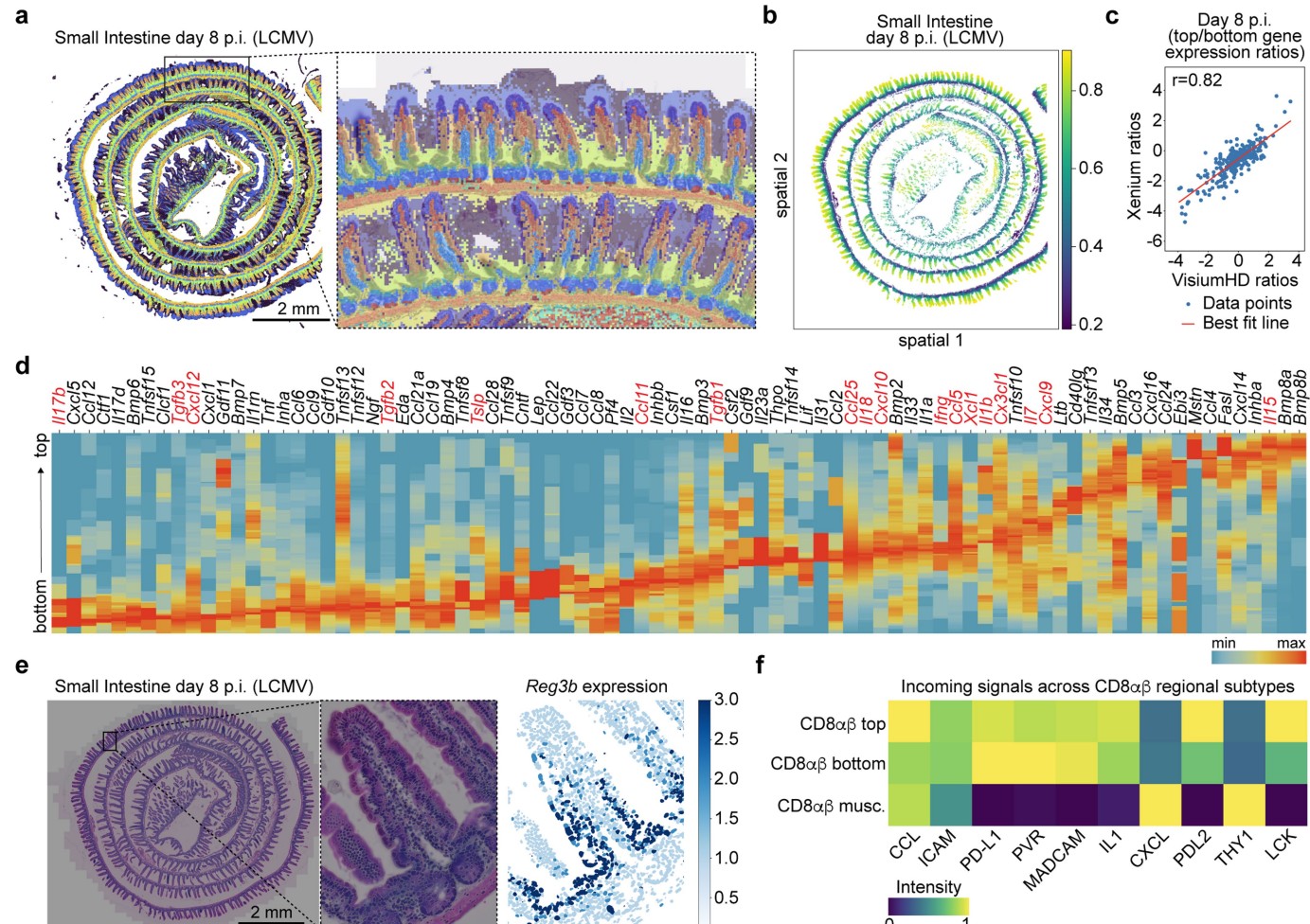

**Extended Data Fig. 4 | Related to Fig. 3. Immune response to LCMV profiled by whole-transcriptome spatial sequencing of the mouse small intestine.** **a**, VisiumHD results on mouse small intestine roll 8 days after LCMV infection colored by graph-based clustering. **b**, VisiumHD spots colored by imputed crypt-villus axis values. **c**, Ratios of overlapping gene expression at the top vs. bottom in epithelial cells from day 8 pi Xenium and VisiumHD dataset. r: Pearson correlation coefficient. **d**, Convolved gene expression of cytokines along the imputed crypt-villus axis in the VisiumHD dataset. Red labels indicate genes that were included in the Xenium gene panel. **e**, Transcript reassigning based on H&E nuclei segmentation to achieve single-nuclei level gene expression data (left) and example (right). *Reg3b* is plotted to showcase the single-nuclei level data. **f**, Incoming signals across CD8αβ regional subtypes in the VisiumHD dataset.

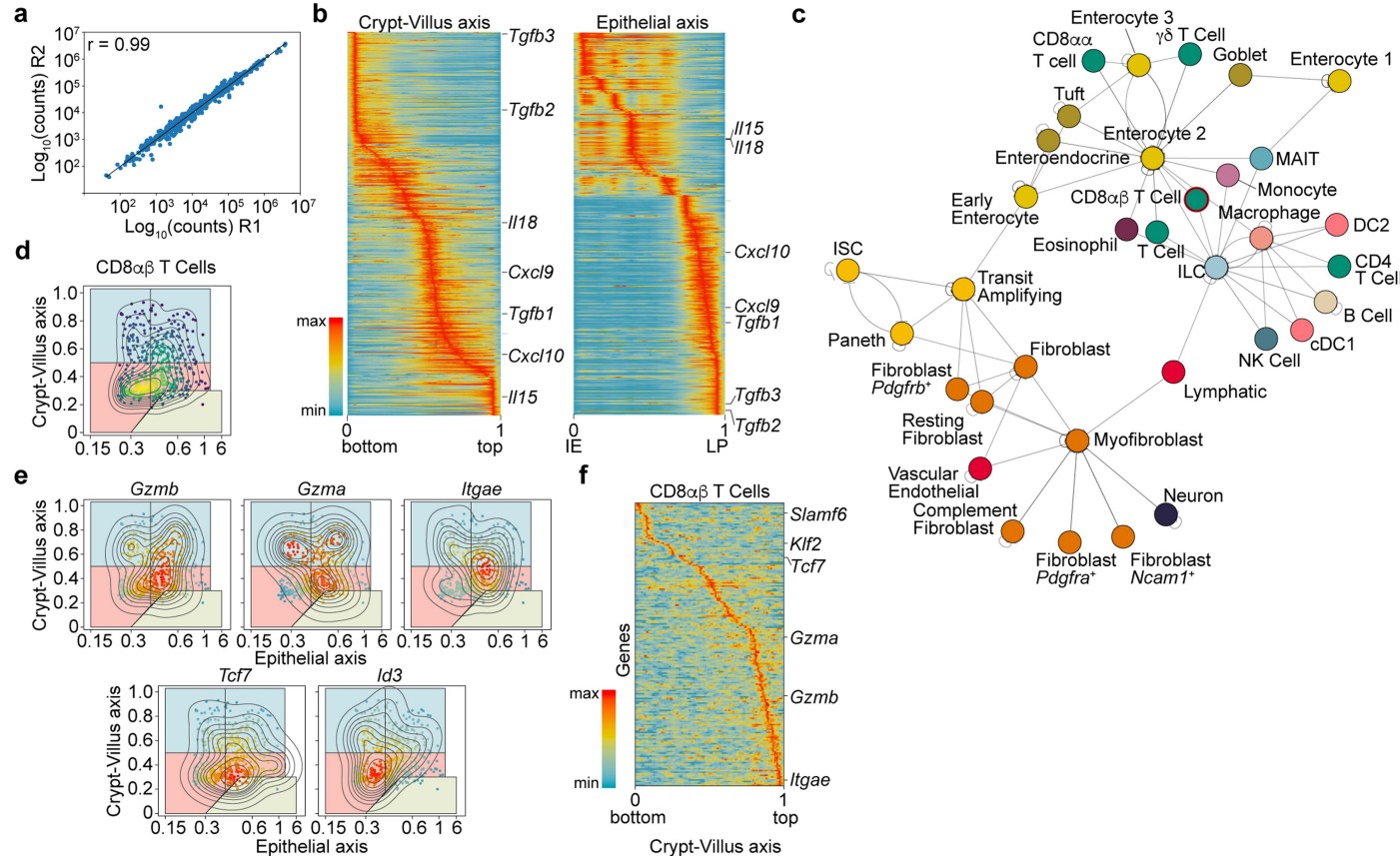

**Extended Data Fig. 5 | Related to Fig. 3. Spatial immune landscapes of the healthy mouse small intestine. a**, Pearson residual correlations of total gene abundances between biological replicates of an uninfected mouse intestine profiled by Xenium. **b**, Convolved gene expression by all cells along the crypt-villus and epithelial axes. 2 biological replicates combined. **c**, Cell type interaction igraph of the uninfected mouse small intestine. 2 biological replicates combined. **d**, IMAP of CD8αβ T cells in the uninfected small intestine. 2 biological replicates combined. **e**, expression IMAP for indicated genes in the uninfected intestine. 2 biological replicates combined. **f**, Convolved gene expression within uninfected CD8αβ T cells along the crypt-villus axis.

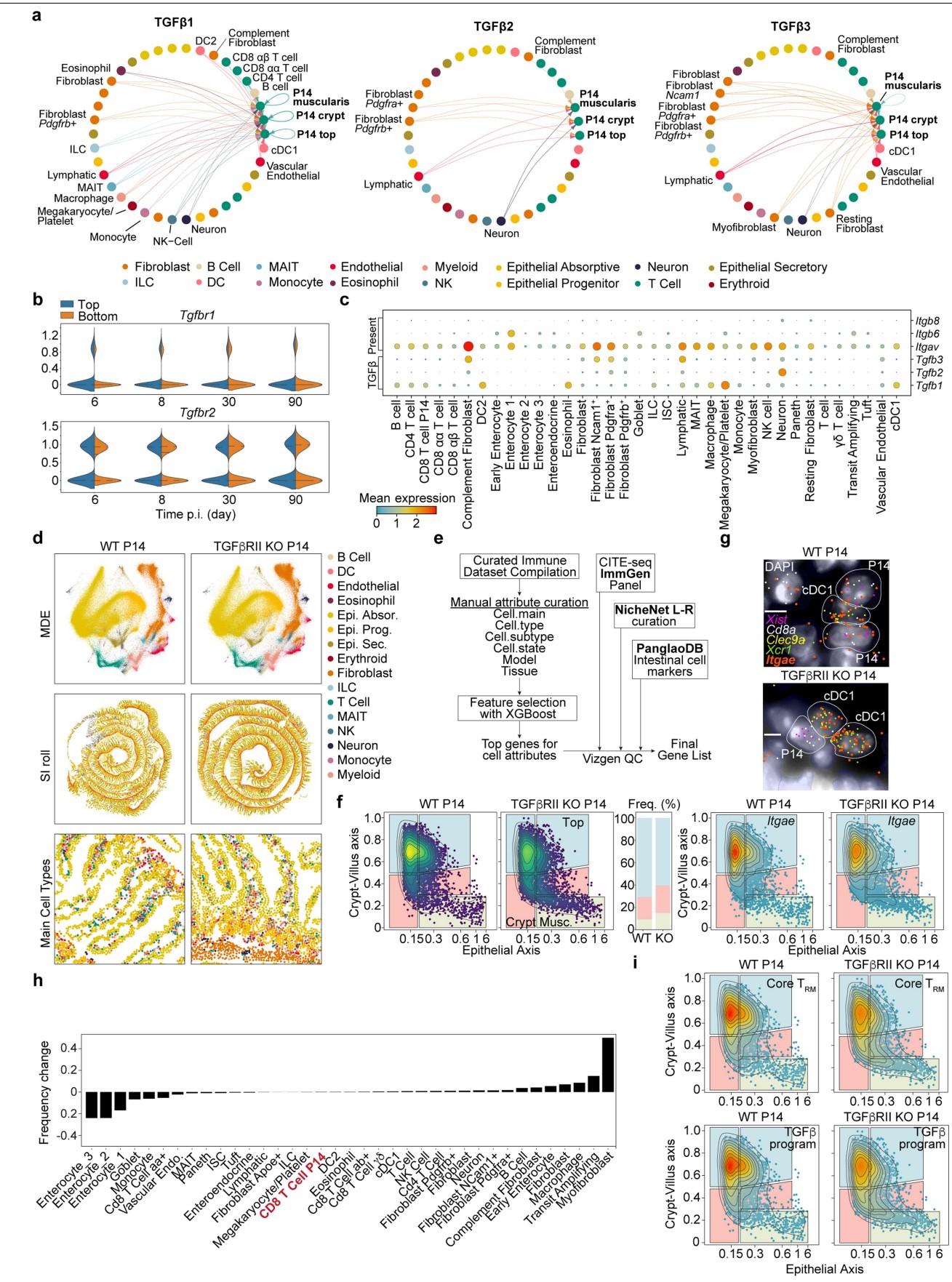

**Extended Data Fig. 6** | See next page for caption.

**Extended Data Fig. 6 | Spatial control of TGFβ signaling controls SI CD8 T cell positioning and differentiation. a**, CellChat circle plots showing top enriched interactions between TGFβ isoform senders and regionally gated P14 receivers across all 8 SI samples (4 timepoints, 2 replicates each). Each cell subtype is represented by a node, and a directed edge is displayed from a top sender subtype to a receiver P14 regional subtype for significant TGFβ sender-P14 interactions. **b**, Violin plots depicting the log-normalized TGFβRI and TGFβRII expression counts within P14s across each timepoint. Violins are plotted with Scanpy, scaled by width, and black dotted lines mark expression quartiles. **c**, The expression of TGFβ isoforms and genes involved in TGFβ presentation as measured by Xenium. **d**, Female wild type or TGFβR2 KO P14 cells were transferred into male C57BL/6 recipients. MERSCOPE-based spatial transcriptomics of the SI was done on day 8 after LCMV infection. One WT and one TGFBR2KO SI were profiled from one biological replicate with 3 mice per condition. Plots show the joint MDE embedding colored by cell type (top), in situ spatial positioning of the cells (middle), and close-ups (bottom).

**e**, Schematic for MERSCOPE gene panel design process. Most important genes for defining cell types were identified using XGBoost on a group of immune datasets, before adding biologically important genes and filtering out MERSCOPE-incompatible genes. **f**, (left) IMAP positioning and kernel density estimate (weighted by gene expression, bottom) coloring of WT and TGFβR2 KO P14 cells. Cells are gated into top intraepithelial, top lamina propria, crypt intraepithelial, crypt lamina propria, and muscularis. (middle) Quantification of P14 cells localized in the muscularis, crypt or top of villus. (right) IMAP colored by kernel density estimates weighted by expression counts of *Itgae*. **g**, Contacting P14s and conventional dendritic cells in zoomed-in regions of the SI. DAPI staining is overlaid with scattered points representing the positions of select transcripts. **h**, Change in frequency of each cell type between WT and TGFβR2 KO conditions. Frequency values reflect proportional increases or decreases of TGFβR2 KO cell type counts relative to WT. **i**, IMAPs of WT and TGFβR2 KO P14 T cells colored by enrichment of the core $T_{RM}$ signature from Milner et al.[5], and the TGFβ program derived from Nath et al.[75].

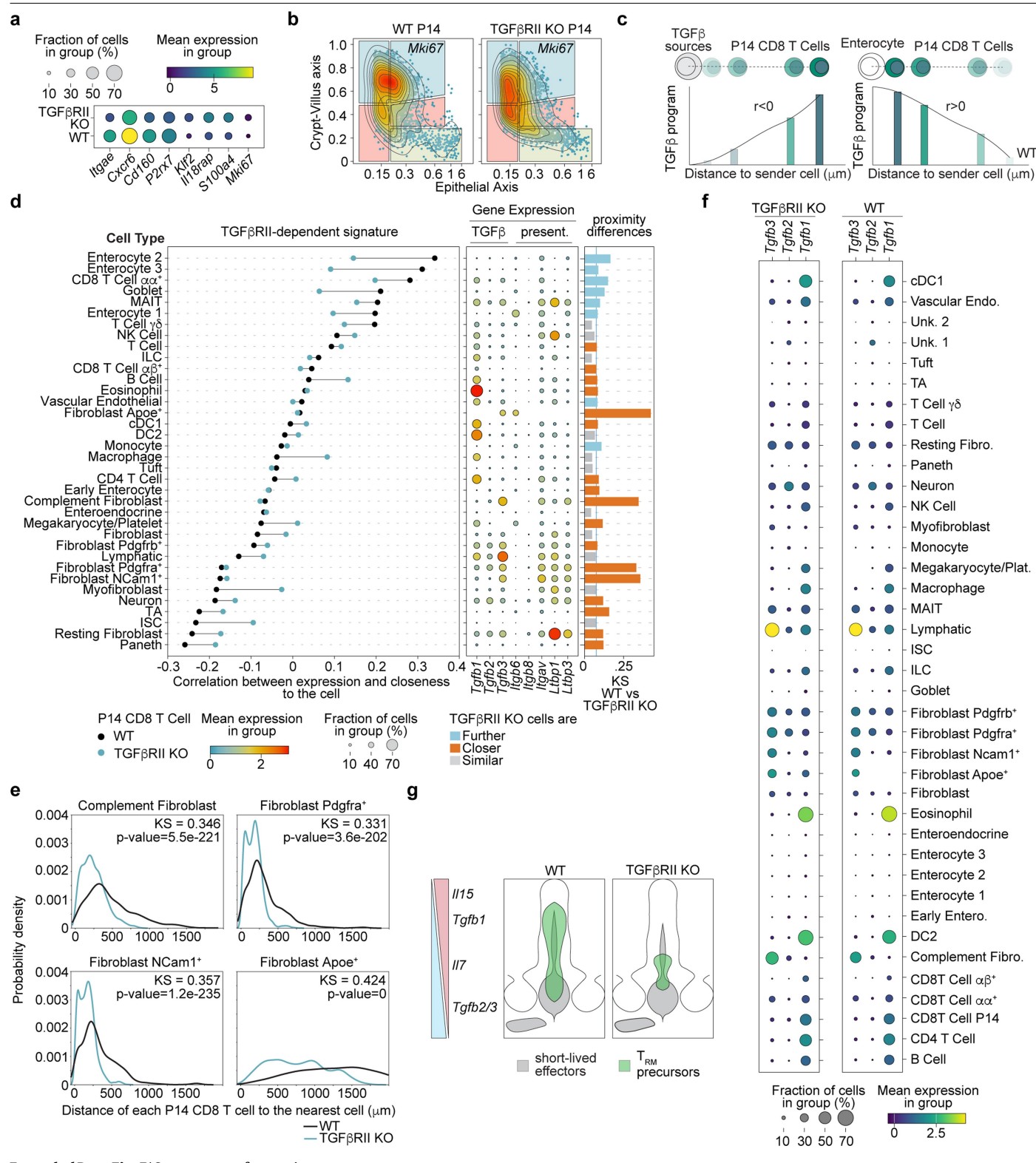

**Extended Data Fig. 7** | See next page for caption.

**Extended Data Fig. 7 | Spatial control of TGFβ signaling controls SI CD8 T cell positioning and differentiation. a**, Top differentially expressed genes between WT and TGFβR2 KO P14 cells. The dot plot is colored by the mean expression of each gene, and the dot size reflects the percentage of P14 cells in which the corresponding gene is expressed. **b**, IMAP representations of WT and KO P14 cells colored by kernel density estimates weighted by expression counts of the proliferation marker *Mki67*. **c** and **d**, TGFβR2 dependent signature enrichment and distance to each cell subtype were calculated for all P14 cells, and Spearman rank correlated against each other (**c**). (**d**) For every subtype, (left) the correlation coefficients between signature enrichment and P14 cell proximity to the subtype among both WT and TGFβR2 KO P14 CD8 T cells, (middle) the expression of TGFβ isoforms and genes involved in TGFβ presentation in the WT sample, and (right) a non-parametric two-sided Kolmogorov–Smirnov statistic indicating the significance of difference of the distance distributions between P14 CD8 T cells and the corresponding cell type in both WT and TGFβR2 KO. The color of the bars indicates whether P14 CD8 T cells are closer to a given cell type in WT (blue) or TGFβR2 KO (red), and a line indicating effect relevance is positioned at 0.08. Supplemental Table 8 presents the cell counts used in the statistical test for the n = 1 experiment across each condition. **e**, Comparisons of the distance between WT or TGFβR2 KO P14 cells and selected other cell subtypes. A two-sided Kolmogorov–Smirnov statistic indicates the difference between the WT and KO distributions for each subtype. The plotted lines show the positional density using a 1D kernel density estimate. **f**, A comparison of TGFβ isoform expression between cell subtypes in WT and TGFβR2 KO. **g**, Proposed mechanism of TGFβ-dependent upward $T_{RM}$ differentiation.

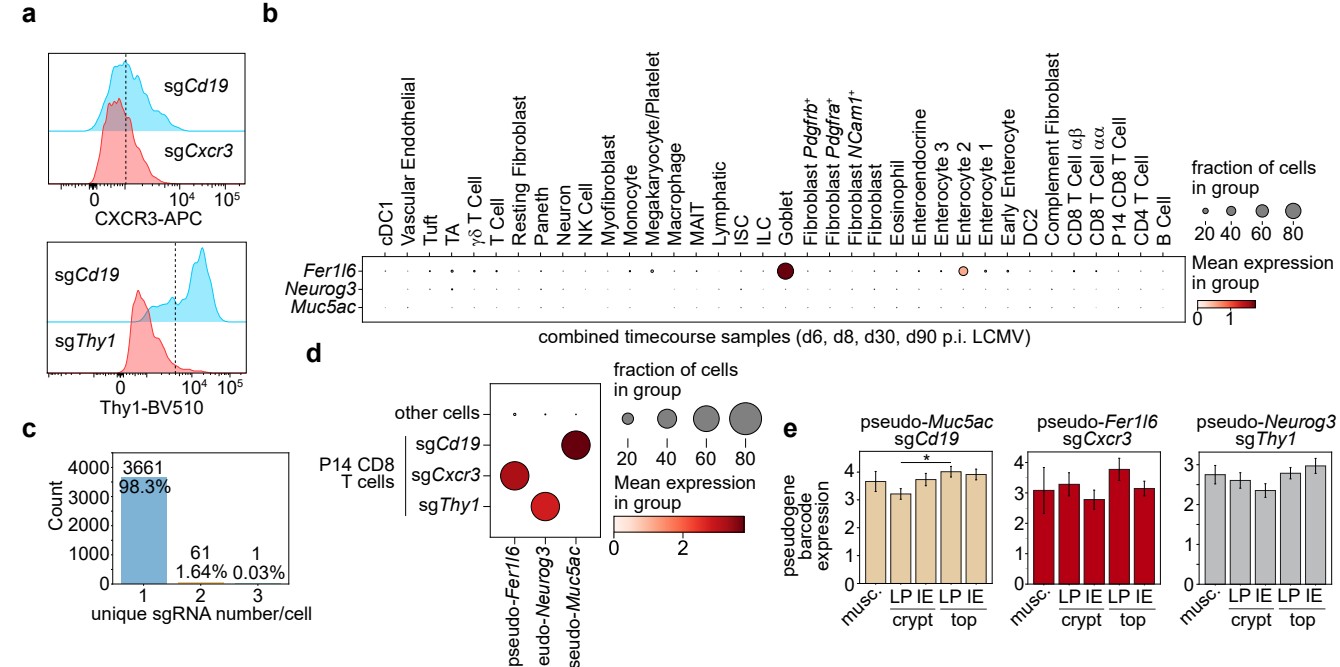

**Extended Data Fig. 8 | Related to Fig. 4. Optical readout of sgRNA-containing antigen-specific CD8 T cells in the mouse intestine integrated in the Xenium assay. a**, Flow cytometry histograms showing quantification of Thy1 and CXCR3 of Cas9eGFP P14 CD8 T cells transduced with Thy1 and Cxcr3-targeting sgRNAs. Representative of two independent biological replicates. **b**, Dot plot gene expression of the three least expressed genes in the 350 gene panel for all cells in the small intestine, all time points and replicates (n = 8) combined. **c**, Frequency of CD8 T cells containing one, two or three different sgRNAs. 2 biological duplicates combined. **d**, Expression of pseudo-gene barcodes in the perturbed day 8 small intestine. **e**, Capture of pseudo-gene barcodes along the spatial areas of the small intestine defined by IMAP for each perturbation. Two-sample t-test of the mean expression levels, with Benjamini-Hochberg correction applied, *p-value < 0.05. Sample sizes shown in Supplemental Table 12.

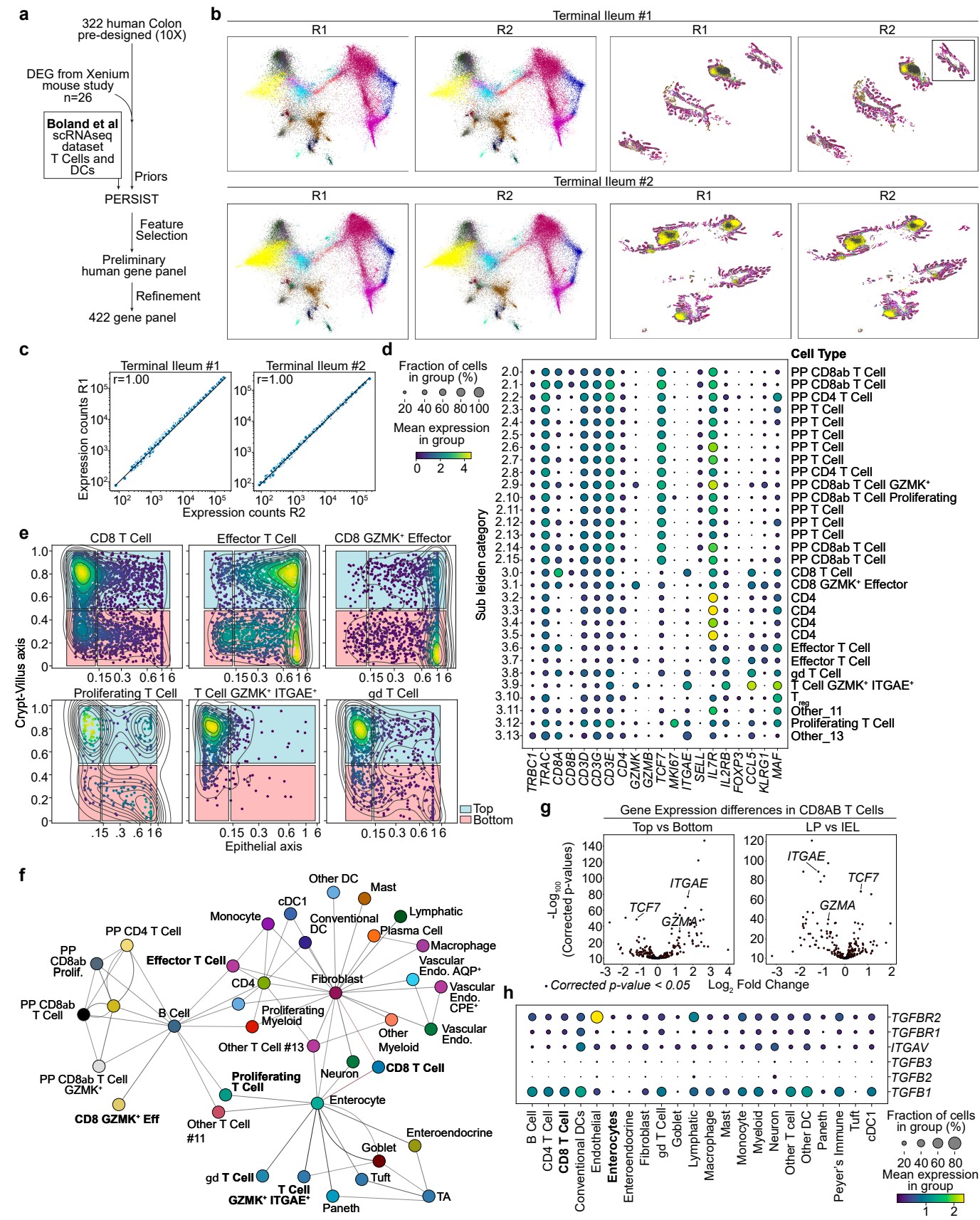

**Extended Data Fig. 9** | See next page for caption.

**Extended Data Fig. 9 | Related to Fig. 5. Immune landscape of the human small intestine revealed by spatial transcriptomics. a**, Schematic for designing the Xenium human SI gene panel. The Xenium base human colon panel was expanded with canonical immune genes, the human homologs of top spatially differentially expressed genes from the Xenium mouse data, and computationally derived genes that best capture the heterogeneity within immune cell types found in scRNA-seq data from Boland et al. **b**, Xenium processed terminal ileum samples divided into two rows corresponding to the two human donors. Adjacent tissue sections were taken from both donors and are positioned side-by-side within the joint MDE embedding (left) and spatially (right). Cells are colored by their annotations in Fig. 5a. **c**, Scattered raw gene expression abundances between the technical replicates of both human ileums overlayed with a line of best fit. The Pearson residual correlation coefficient (r) is calculated between the gene abundances of both samples. **d**, Expression of genes used to annotate immune subtypes. Colors of dots indicate the mean expression of the gene in each subcluster, and size of the dots correspond to the percentage of cells in each subcluster expressing the gene. The final cell subtype annotations of each subcluster are shown as y-ticks along the right side of the plot. **e**, IMAP positioning of select T-Cell subtypes within all (n = 4) human sections (Peyer's Patches excluded). Cells are colored by kernel density estimates of their coordinate location within the IMAP. IMAP gates are positioned as in Fig. 5d. **f**, Aggregated physical interaction network where edges between nodes represent a normalized Squidpy interaction score lying above a 0.1 threshold (10% of the connections). Nodes are positioned using a Kamada-Kawai layout algorithm on the averaged interaction matrix of all human sections. **g**, Differential expression testing of all genes expressed in at least 5% of human CD8αβ T Cells using diffxpy. A two-tailed Wald test yielded a fold change and adjusted p-value (padj) for each gene (X) between human CD8αβ T cells gated in the crypt versus those gated in the top of the villus, and (X) human CD8αβ T cells gated intraepithelial versus those gated in the lamina propria. All genes are plotted by their log2 fold change and -log100(padj), and significantly differentially expressed genes (padj <0.05) are colored red. **h**, Expression of TGFβ isoforms and genes involved in TGFβ presentation across cell types after pooling the cells from all human sections (n = 4).

# Reporting Summary

## Statistics

For all statistical analyses, confirm that the following items are present in the figure legend, table legend, main text, or Methods section.

| n/a | Confirmed | |
|---|---|---|
| ☐ | ☒ | The exact sample size (*n*) for each experimental group/condition, given as a discrete number and unit of measurement |
| ☐ | ☒ | A statement on whether measurements were taken from distinct samples or whether the same sample was measured repeatedly |
| ☐ | ☒ | The statistical test(s) used AND whether they are one- or two-sided *Only common tests should be described solely by name; describe more complex techniques in the Methods section.* |
| ☐ | ☒ | A description of all covariates tested |
| ☐ | ☒ | A description of any assumptions or corrections, such as tests of normality and adjustment for multiple comparisons |
| ☐ | ☒ | A full description of the statistical parameters including central tendency (e.g. means) or other basic estimates (e.g. regression coefficient) AND variation (e.g. standard deviation) or associated estimates of uncertainty (e.g. confidence intervals) |
| ☐ | ☒ | For null hypothesis testing, the test statistic (e.g. *F*, *t*, *r*) with confidence intervals, effect sizes, degrees of freedom and *P* value noted *Give P values as exact values whenever suitable.* |
| ☒ | ☐ | For Bayesian analysis, information on the choice of priors and Markov chain Monte Carlo settings |
| ☒ | ☐ | For hierarchical and complex designs, identification of the appropriate level for tests and full reporting of outcomes |
| ☐ | ☒ | Estimates of effect sizes (e.g. Cohen's *d*, Pearson's *r*), indicating how they were calculated |

*Our web collection on statistics for biologists contains articles on many of the points above.*

## Software and code

Policy information about availability of computer code

| Data collection | Histology and Immunofluorescence  images were acquired on an Olympus VS200 Slide Scanner (UCSD Microscopy CORE) or on a ZEISS LSM700 confocal microscope. Spatial transcriptomics data was acquired on the 10x Xenium and Vizgen MERSCOPE. |
|---|---|
| Data analysis | Data processing pipelines and the code to reproduce all figures for this project will be made publicly available on Github (https://github.com/Goldrathlab/Spatial-TRM-paper) after publication. <br><br> The following packages have been used: <br><br> Cellpose 2.2.3 <br> Baysor 0.6.2 <br> MERLIN 232.230125.316b <br> scipy        1.10.1 <br> torch        2.1.0 <br> labelme        5.5.0 <br> tensorflow      2.15.0 <br><br> [dependencies] <br> scanpy = "==1.9.5" <br> scikit-learn = "==1.1.3" <br> scvi-tools = "==1.1.1" |

```
tqdm = ">=4.66.5,<5"
r = ">=4.3,<4.4"
r-tidyverse = ">=2.0.0,<3"
jupyterlab = ">=4.2.5,<5"
r-irkernel = ">=1.3.2,<2"
bioconductor-zellkonverter = ">=1.12.1,<2"
radian = ">=0.6.13,<0.7"
bioconductor-genomeinfodbdata = ">=1.2.11,<2"
r-biocmanager = ">=1.30.25,<2"
bioconductor-scater = ">=1.30.1,<2"
bioconductor-scran = ">=1.30.0,<2"
bioconductor-scuttle = ">=1.12.0,<2"
bioconductor-singlecellexperiment = ">=1.24.0,<2"
imageio = ">=2.35.1,<3"
alphashape = ">=1.3.1,<2"
opencv = ">=4.10.0,<5"
tifffile = ">=2024.8.28,<2025"
ipywidgets = ">=8.1.5,<9"
pre_commit = ">=3.8.0,<4"
openpyxl = ">=3.1.5,<4"
scvelo = "==0.3.1"
squidpy = ">=1.2.3,<2"
optax = "0.2.0.*"
libegl = ">=1.7.0,<2"
libopengl = ">=1.7.0,<2"
libgl = ">=1.7.0,<2"
python = ">=3.10.14,<3.11"
r-remotes = ">=2.5.0,<3"
bioconductor-complexheatmap = ">=2.18.0,<3"
r-lme4 = ">=1.1_35.5,<2"
r-future = ">=1.34.0,<2"
r-patchwork = ">=1.2.0,<2"
r-svglite = ">=2.1.3,<3"
r-rstatix = ">=0.7.2,<0.8"
r-shiny = ">=1.9.1,<2"
r-pbkrtest = ">=0.5.3,<0.6"
r-plotly = ">=4.10.4,<5"
r-ggsci = ">=3.2.0,<4"
r-ggpubr = ">=0.6.0,<0.7"
r-pbapply = ">=1.7_2,<2"
r-nmf = ">=0.21.0,<0.22"
r-ggalluvial = ">=0.12.5,<0.13"
r-network = ">=1.18.2,<2"
r-ggnetwork = ">=0.5.13,<0.6"
r-rspectra = ">=0.16_2,<0.17"
r-presto = ">=1.0.0,<2"

[pypi-dependencies]
ucell = { git = "https://github.com/maximilian-heeg/UCell.git", rev = "3d29122" }
```

For manuscripts utilizing custom algorithms or software that are central to the research but not yet described in published literature, software must be made available to editors and reviewers. We strongly encourage code deposition in a community repository (e.g. GitHub). See the Nature Portfolio guidelines for submitting code & software for further information.

# Data

Policy information about availability of data

All manuscripts must include a data availability statement. This statement should provide the following information, where applicable:

- Accession codes, unique identifiers, or web links for publicly available datasets
- A description of any restrictions on data availability
- For clinical datasets or third party data, please ensure that the statement adheres to our policy

The code to reproduce the analysis presented in this manuscript is available on GitHub (https://github.com/Goldrathlab/Spatial-TRM-paper). Sequencing data and spatial transcriptomics data are deposited in GEO: GSE279254 (VisiumHD), GSE279255 (snRNA sequencing) and GSE280895 (Xenium & Merscope).

# Research involving human participants, their data, or biological material

Policy information about studies with human participants or human data. See also policy information about sex, gender (identity/presentation), and sexual orientation and race, ethnicity and racism.

Reporting on sex and gender | 2/2 healthy donors were female.

| | |
|---|---|
| Reporting on race, ethnicity, or other socially relevant groupings | No information about ethnicity or race available. |
| Population characteristics | Both healthy donors were adults, no further age characteristics are available. |
| Recruitment | Deidentified human FFPE samples from healthy subjects were acquired from the San Diego Digestive Diseases Research Center (SDDRC). |
| Ethics oversight | The Human Research Protection Programs at the University of California, San Diego reviewed and approved the protocol, including a waiver of consent. |

Note that full information on the approval of the study protocol must also be provided in the manuscript.

# Field-specific reporting

Please select the one below that is the best fit for your research. If you are not sure, read the appropriate sections before making your selection.

☒ Life sciences     ☐ Behavioural & social sciences     ☐ Ecological, evolutionary & environmental sciences

For a reference copy of the document with all sections, see nature.com/documents/nr-reporting-summary-flat.pdf

# Life sciences study design

All studies must disclose on these points even when the disclosure is negative.

| | |
|---|---|
| Sample size | No statistical methods were used to pre-determine sample sizes, but our sample sizes are like those reported in previous publications (e.g. Milner et al, Nature 2017) from our laboratory and others. |
| Data exclusions | No data were excluded |
| Replication | All mouse experiments were successfully repeated ≥2 times and where possible quantification and statistics were run on combined replicate experiments, when possible. |
| Randomization | All mice were between 1.5 and 6 months old at the time of infection and randomly assigned to experimental groups. Human samples were not randomized. |
| Blinding | No blinding was performed during mouse experiments. The experimental observations presented would be consistent irrespective of blinding and therefore blinding was not relevant in this study. |

# Reporting for specific materials, systems and methods

We require information from authors about some types of materials, experimental systems and methods used in many studies. Here, indicate whether each material, system or method listed is relevant to your study. If you are not sure if a list item applies to your research, read the appropriate section before selecting a response.

## Materials & experimental systems

| n/a | Involved in the study |
|---|---|
| ☐ | ☒ Antibodies |
| ☒ | ☐ Eukaryotic cell lines |
| ☒ | ☐ Palaeontology and archaeology |
| ☐ | ☒ Animals and other organisms |
| ☒ | ☐ Clinical data |
| ☒ | ☐ Dual use research of concern |
| ☒ | ☐ Plants |

## Methods

| n/a | Involved in the study |
|---|---|
| ☒ | ☐ ChIP-seq |
| ☐ | ☒ Flow cytometry |
| ☒ | ☐ MRI-based neuroimaging |

# Antibodies

| | |
|---|---|
| Antibodies used | Marker Color  Clone # Vendor Catalog # Dilution Validation<br>CD3 PE 145-2C11 eBioscience 12-0031-83 200 flow cytometric analysis of mouse thymocytes and splenocytes<br>TCR ab APC H57-597 eBioscience 17-5961-83 200 flow cytometric analysis of mouse thymocytes and splenocytes<br>NK1.1 FITC PK136 eBioscience 11-5941-81 400 flow cytometric analysis of C57Bl/6 mouse splenocytes<br>CD19 PerCP-Cy5.5 eBio1D3 eBioscience 45-0193-82 200 flow cytometric analysis of mouse splenocytes<br>CD8b BV421 H35-17.2 eBioscience 48-0083-82 400 flow cytometric analysis of mouse splenocytes<br>CD45.1 BV510 A20 Biolegend 110741 200 Verified Reactivity: Mouse |

TCR gd BV711 GL3 Biolegend 118149 200 Verified Reactivity: Mouse
CD4 BV786 GK1.5 Biolegend 100453 400 Verified Reactivity: Mouse
CD8a PE-Cy7 53-6.7 eBioscience 25-0081-82 400 flow cytometric analysis of mouse thymocytes and splenocytes
CD11b PE M1/70 eBioscience 12-0112-82 200 flow cytometric analysis of mouse splenocytes or bone marrow cells
CD11c APC N418 Biolegend 117310 200 Verified Reactivity: Mouse
Ly6C FITC AL-21 BD 553104 200 "Reactivity: Mouse (QC Testing)
Application:Flow cytometry (Routinely Tested)"
Ly6G PerCP-Cy5.5 1A8 Biolegend 127615 200 Verified Reactivity: Mouse
Xcr1 BV421 ZET Biolegend 148216 200 Verified Reactivity: Mouse, Rat
CD45 BV510 30-F11 BD 561487 200 "Reactivity: Mouse (QC Testing)
Application:Flow cytometry (Routinely Tested)"
F4/80 BV711 BM8 Biolegend 123147 200 Verified Reactivity: Mouse
MHC II BV786 M5/114.15.2 Biolegend 107645 200 Verified Reactivity: Mouse
B220 PE-Cy7 RA3-6 B2 Biolegend 103222 200 "Verified Reactivity: Mouse, Human
Reported Reactivity: Cat"
Fixable Viability Dye APC-Cy7  eBioscience 65-0865-14 1000 flow cytometric analysis of mouse thymocytes
anti-hamster IgG  n/a polyclonal Thermo Fisher Scientific PI31115 50
CD3e n/a 145-2C11 Fisher Scientific 50-112-9591 1000 flow cytometric analysis of mouse splenocytes
CD28 n/a 37.51 Fisher Scientific 50-112-9711 1000 flow cytometric analysis of mouse splenocytes
Thy1.2  BV510 30-H12 BioLegend 105335 200 flow cytometric analysis of mouse splenocytes
Cxcr3 APC Cxcr3-173 eBiosciences 17-1831-82 200 flow cytometric analysis of mouse splenocytes
CD45.1 AF594 A20 BioLegend 110756 50 IHC-F - Quality tested
E-cadherin APC DECMA-1 BioLegend 147312 200 FC - Quality tested
CD8a FITC 53-6.7 cytekbio 35-0081-U500 50 Flow cytometry
CD8a n/a/ EPR21769 abcam ab217344 50 Suitable for IP, Flow Cyt, WB, IHC-Fr, IHC-P and reacts with Mouse samples
Anti-rabbit AF594 Polyclonal Invitrogen A-11012 200

| Validation | All antibodies were obtained from commercial vendors.<br>Marker Color  Clone # Vendor Catalog # Dilution Validation<br>CD3 PE 145-2C11 eBioscience 12-0031-83 200 flow cytometric analysis of mouse thymocytes and splenocytes<br>TCR ab APC H57-597 eBioscience 17-5961-83 200 flow cytometric analysis of mouse thymocytes and splenocytes<br>NK1.1 FITC PK136 eBioscience 11-5941-81 400 flow cytometric analysis of C57Bl/6 mouse splenocytes<br>CD19 PerCP-Cy5.5 eBio1D3 eBioscience 45-0193-82 200 flow cytometric analysis of mouse splenocytes<br>CD8b BV421 H35-17.2 eBioscience 48-0083-82 400 flow cytometric analysis of mouse splenocytes<br>CD45.1 BV510 A20 Biolegend 110741 200 Verified Reactivity: Mouse<br>TCR gd BV711 GL3 Biolegend 118149 200 Verified Reactivity: Mouse<br>CD4 BV786 GK1.5 Biolegend 100453 400 Verified Reactivity: Mouse<br>CD8a PE-Cy7 53-6.7 eBioscience 25-0081-82 400 flow cytometric analysis of mouse thymocytes and splenocytes<br>CD11b PE M1/70 eBioscience 12-0112-82 200 flow cytometric analysis of mouse splenocytes or bone marrow cells<br>CD11c APC N418 Biolegend 117310 200 Verified Reactivity: Mouse<br>Ly6C FITC AL-21 BD 553104 200 "Reactivity: Mouse (QC Testing)<br>Application:Flow cytometry (Routinely Tested)"<br>Ly6G PerCP-Cy5.5 1A8 Biolegend 127615 200 Verified Reactivity: Mouse<br>Xcr1 BV421 ZET Biolegend 148216 200 Verified Reactivity: Mouse, Rat<br>CD45 BV510 30-F11 BD 561487 200 "Reactivity: Mouse (QC Testing)<br>Application:Flow cytometry (Routinely Tested)"<br>F4/80 BV711 BM8 Biolegend 123147 200 Verified Reactivity: Mouse<br>MHC II BV786 M5/114.15.2 Biolegend 107645 200 Verified Reactivity: Mouse<br>B220 PE-Cy7 RA3-6 B2 Biolegend 103222 200 "Verified Reactivity: Mouse, Human<br>Reported Reactivity: Cat"<br>Fixable Viability Dye APC-Cy7  eBioscience 65-0865-14 1000 flow cytometric analysis of mouse thymocytes<br>anti-hamster IgG  n/a polyclonal Thermo Fisher Scientific PI31115 50<br>CD3e n/a 145-2C11 Fisher Scientific 50-112-9591 1000 flow cytometric analysis of mouse splenocytes<br>CD28 n/a 37.51 Fisher Scientific 50-112-9711 1000 flow cytometric analysis of mouse splenocytes<br>Thy1.2  BV510 30-H12 BioLegend 105335 200 flow cytometric analysis of mouse splenocytes<br>Cxcr3 APC Cxcr3-173 eBiosciences 17-1831-82 200 flow cytometric analysis of mouse splenocytes<br>CD45.1 AF594 A20 BioLegend 110756 50 IHC-F - Quality tested<br>E-cadherin APC DECMA-1 BioLegend 147312 200 FC - Quality tested<br>CD8a FITC 53-6.7 cytekbio 35-0081-U500 50 Flow cytometry<br>CD8a n/a/ EPR21769 abcam ab217344 50 Suitable for IP, Flow Cyt, WB, IHC-Fr, IHC-P and reacts with Mouse samples<br>Anti-rabbit AF594 Polyclonal Invitrogen A-11012 200 |
| --- | --- |

# Animals and other research organisms

Policy information about studies involving animals; ARRIVE guidelines recommended for reporting animal research, and Sex and Gender in Research

| Laboratory animals | Mice were maintained in specific-pathogen-free conditions at a temperature between 18°C and 23°C with 40–60% humidity and a 12h-light and 12h-dark light cycle in accordance with the Institutional Animal Care and Use Committees (IACUC) of the University of California San Diego (UCSD). All mice were of C57BL/6J background and bred at the University of California San Diego (UCSD) or purchased from the Jackson Laboratory. R26Cre-ERT2 (stock no. 008463, Jackson Laboratory), Tgfbr2fl/fl (stock no. 012603, Jackson Laboratory), P14, and CD45.1 congenic mice were bred in-house. All mice used were between 1.5 and 6 months old. |
| --- | --- |

| Wild animals | No wild animals were used. |
|---|---|
| Reporting on sex | For immunofluorescence male and female mice were used for infection experiments and adoptively transferred P14 cells were either sex matched or male cells were transferred into female recipients.<br>For spatial transcriptomic experiments male TCR-transgenic T cells were adoptively transferred into female C57BL/6 recipient mice. |
| Field-collected samples | The study did not involve samples collected from the field. |
| Ethics oversight | Institutional Animal Care and Use Committee of the University of California San Diego |

Note that full information on the approval of the study protocol must also be provided in the manuscript.

## Plants

| Seed stocks | *Report on the source of all seed stocks or other plant material used. If applicable, state the seed stock centre and catalogue number. If plant specimens were collected from the field, describe the collection location, date and sampling procedures.* |
|---|---|
| Novel plant genotypes | *Describe the methods by which all novel plant genotypes were produced. This includes those generated by transgenic approaches, gene editing, chemical/radiation-based mutagenesis and hybridization. For transgenic lines, describe the transformation method, the number of independent lines analyzed and the generation upon which experiments were performed. For gene-edited lines, describe the editor used, the endogenous sequence targeted for editing, the targeting guide RNA sequence (if applicable) and how the editor was applied.* |
| Authentication | *Describe any authentication procedures for each seed stock used or novel genotype generated. Describe any experiments used to assess the effect of a mutation and, where applicable, how potential secondary effects (e.g. second site T-DNA insertions, mosiacism, off-target gene editing) were examined.* |

## Flow Cytometry

### Plots

Confirm that:

☒ The axis labels state the marker and fluorochrome used (e.g. CD4-FITC).

☒ The axis scales are clearly visible. Include numbers along axes only for bottom left plot of group (a 'group' is an analysis of identical markers).

☒ All plots are contour plots with outliers or pseudocolor plots.

☒ A numerical value for number of cells or percentage (with statistics) is provided.

### Methodology

| Sample preparation | Isolation of CD8 T cells was performed similarly as described (Steinert at al, 2015). Small intestine (SI) intra-epithelial lymphocytes (IEL) and lamina propria lymphocytes (LPL) were prepared by removing Peyer's patches and the luminal contents from the entire SI. The SI was then cut longitudinally and into 1 cm pieces, then incubated at 37°C for 30 minutes in HBSS with 2.1 mg/mL sodium bicarbonate, 2.4 mg/mL HEPES, 8% bovine growth serum, and 0.154 mg/mL of dithioerythritol (EMD Millipore). Collection of the supernatant through a 70 µM constituted the IEL compartment of the SI. The remaining tissue fragments of the SI were further incubated in RPMI with 1.2 mg/mL HEPES, 292 µ/mL L-glutamine, 1 mM MgCl2, 1 mM CaCl2, 5% fetal bovine serum, and 100 U/mL collagenase (Worthington) at 37°C for 30 min. After enzymatic incubation, tissues were filtered through a 70-µm nylon cell strainer (Falcon). Tissue preparations were separated on a 44%/67% Percoll density gradient. |
|---|---|
| Instrument | For flow cytometry, all events were acquired on a BD LSRFortessa X-20 or a BD LSRFortessa. |
| Software | FlowJo v10.10.0 |
| Cell population abundance | Sorted transduced samples had a purity of > 95%. |
| Gating strategy | The gating strategy is shown in supplemental figure 1. |

☒ Tick this box to confirm that a figure exemplifying the gating strategy is provided in the Supplementary Information.

