## [Peer Review File · Nature]

Tissue-Resident Memory CD8 T Cell Diversity is Spatiotemporally Imprinted

Corresponding Author: Professor Ananda W Goldrath

Version 0:

Reviewer comments:

Referee #1

(Remarks to the Author)

The study shows that within a given organ there exists location-dependant variation in tissue-resident T cells that is (at least partly) controlled by differential TGFb signalling. The researchers exploit spatial transcriptomics to show that the Trm population in the small intestine is not binary – in other words, they do not consist of only immature polyfunctional Trm and fully mature effector Trm, but a spectrum of cells that exist along a progressing maturation continuum.

It is known that TGFb is an environmental factor required for driving Trm development and maturation in some tissues, although in organs like the liver, an absence of TGFb signalling results in a Trm population that is immature-like with more reprogramming flexibility when compared to those that have undergone full TGFb-driven differentiation. However, what is novel and important here is the demonstration that the SI Trm population exists along a TGFb-controlled maturation continuum with the most immature Trm cells found in the crypts while the most mature Trm localise to the tip of the villi – with those cells 'in between' varying in their state of maturation. This maturation continuum is not just spatially dependent – but also time dependent. All tissues are going to have cells at different stages of maturity early in the primary or challenge responses. Based on this, the demonstration that the Trm feature this temporal variation is of major importance. Altogether, this paper it argues that even within a given tissue not all Trm are the same and they can't be assumed to be a homogeneous population across space and time – an assumption that dominates the literature to this point in time.

Detailed points:

1) I found the proposed nomenclature confusing. The authors use the term “polyfunctional memory Trm” for those in the crypt and “effector Trm” for those at or near the villus tip. I don't think the Authors are suggesting that the later are traditional tissue circulating effector memory T cells (quite the opposite), but this nomenclature may confuse the reader and/or lead to conclusions that these cells are Tem-like. Also I do not understand how the 'in between' cells fit within this functionality-based paradigm. A nomenclature that may better reflect the concept of a continuum as seen in Figure 2g,h is to call the cells in the crypt “immature Trm” since they lack many of the canonical Trm markers (such as high CD103 and low Klf2) - and those in the villi “mature Trm” since they manifest the full TGFb-driven transcriptional signature.

2) It remains unclear whether this type of variation is found in other tissues. As mentioned above, it's been shown that liver lacks TGFb-imprinted CD103+ Trm cells. As a consequence, might this population be relatively more homogeneous?

3) TGFb is important in Trm development – but the results in Figures 3 and 4 and their interpretation is confusing. TGFb concentration declines towards the top of the villus (Fig 3d-g), yet the authors argue that TGFb program is highest for the T cells are closest to enterocytes at the top of the villus and lowest for those near the muscularis (Fig 4g). The speculation that the Trm are programmed at the base of the villus to be positioned at the top seems like a long bow to draw.

(Remarks on code availability)

Referee #3

(Remarks to the Author)

This is a very data dense manuscript examining microanatomical P14 CD8 T cell localization and phenotype in the small intestine. The authors use cutting edge techniques, such as spatial transcriptomics, to simultaneously score, phenotype and locate P14 within the small intestine at different times after LCMV-Armstrong infection. Human ileum were also analyzed which adds an important component to this manuscript. I understand that the prepping of samples and data analysis is a huge effort and, to my knowledge, this is the first time these sort of approaches have been used to examine Ag-specific CD8 T cells in the small intestine. This makes this manuscript and the data very innovative and significant.

However, there are some important considerations in their studies that are missing, which make it hard to evaluate the data.

- 1) Most of the data is based on 2 biological replicates. While I understand that this is likely due to the amount of time and effort it takes to produce data from one n, reproducibility can't be forgotten just because of the technique. Also, because of this, there are not a lot of statistical analyses. The authors state that their n=2 are reproducible, but Ext Fig 2b shows that is not true, especially at day 6 and 8. The authors also state that P14 are equally distributed longitudinally (line 116), but this figure leads one to question that statement.
- 2) Many techniques used here are not standard for most investigators. Uninfected or no transfer control mice are not shown. Using Xist as a marker of T cells is clever, but no data are presented to demonstrate that Xist expression is constant during T cell activation.
- 3) In many figures, it is hard to understand what was really done, what time points the data are referring to, etc. E.g., in Fig 2d, are the cell numbers just in the one slice of the entire Swiss roll? Is there a calculation to account for the entire intestine? Fig 2g – legend says all cells in the timecourse. Does that mean that P14 at all timepoints are analyzed together here? If so, why? Fig 4a legend does not say it is day 8, only in text.
- 4) There is a lot of speculation in this manuscript, e.g. line 196 and lines 282-285 about TGFb. The data show that cells localize differently when P14 do not sense TGFb, but the authors build a whole pathway of differentiation from this observation.
- 5) There is little validation of what the authors discovered using more standard methods, such as flow or immunofluorescence on tissue sections. While Ext Fig 1, shows % cells after enzymatic digestion (again n=2), there is no quantification of cell numbers here to try to match or understand the numbers of P14 in Fig 2d.
- 6) Some of the discoveries in this manuscript are surprising, e.g., stable CD103 expression over the day 6-day 90 timepoints, as there are published flow data that CD8 T cells enter the SI as CD103 low and upregulate CD103 over time. Low expression of granzyme at late timepoints post infection is also surprising, as ex vivo lytic activity of CD8 T cells in SI has been demonstrated previously. Equal numbers of P14 at day 6 and day 8 are documented in this manuscript with their techniques, yet that would not match published data. While what is discovered here may be correct and data from others may be the result of sub-optimal tissue isolation techniques, there should be some validation done as part of this study.
- 7) The differentiation pattern of P14 is assumed to be driven just by the genes/proteins they find in their assays. However, they do not discuss the impact of Ag recognition in the SI by P14. While virus is thought to be cleared by day 6/8, Ag may still be present. P14 in different locations appear to differentially regulate Ki67, Thy1 expression and type II interferon which are associated with T cell signaling, proliferation and effector function due to Ag. It would be interesting to do these same experiments using a system where Ag is not present in the SI.

-Refs 15, 16 seem to be in the wrong place in paragraph 1.

-Line 150 – did they mean “with” and not “within”?

-The title states functional diversity, but no actual functions are tested here, as stated in line 104 - they capture transcriptional regionalization (going back to point 4).

-The authors often state that cells at the top are in contact with epithelial cells (e.g., line 157). While that may be true for IEL, there is a basement membrane under the epithelium. It is not clear that cells in the LP in that area would contact epithelial cells.

-The “top” and “bottom” designations are not very specific (Fig 2a) and depending on the length of the villus (very long on day 8), half of the villus could be “top” (red). Unclear how these were designated.

-The authors assume that once a T cell enters a particular villus, it does not leave that villus (to go to another villus – I do not mean back into the circulation). The perceived differentiation pathways are based on this assumption, but it has not been shown that this single villus residency is the case and some do speculate that T cells move between villi.

(Remarks on code availability)

Referee #4

(Remarks to the Author)

In this study, the Goldrath lab makes use of spatial transcriptomics to examine TRM differentiation in the small intestine. They use human samples and a murine model of acute systemic viral infection to profile the location and transcriptome of TRM cell differentiation at single-transcript resolution. They develop computational approaches to capture cellular locations and visualize the spatiotemporal distribution of cell types and gene expression. They show TRM populations were spatially segregated, with more effector and memory-like TRM preferentially localized at the villus tip or crypt, respectively. Using a computational approach of modeling ligand-receptor activity suggests patterns of cellular interactions and cytokine signaling pathways that initiate and maintain TRM differentiation and functional diversity, including different TGFb sources. Alterations

in the cellular networks induced by loss of TGFb receptor expression revealed a model consistent with TGFb promoting progressive TRM maturation towards the villus tip. Overall, the authors have developed a framework for the study of immune cell interactions with the spectrum of tissue cell types, revealing that T cell location and functional state are fundamentally intertwined.

This manuscript builds on a long list of studies by the Goldrath lab that have examined TRM differentiation in great detail, in particular in the small intestine. Here the authors take their studies to the next level using new tools that allow at an unprecedented depth an illustration of the processes that results in TRM differentiation and diversification. Although one could argue that novel insights into TRM differentiation are limited, it is clear that the authors make exemplary use of state-of-the-art technology and analysis methods, which are likely to be of great interest to many research groups.

While this study overall is great, it would be important to provide additional biological data to underpin the main conclusions. Specifically, the authors state "Our findings support a model in which the cellular organization of the intestine provides localized instructions through regionalization of cytokine secretion and distinct cellular interactions that lead TRM differentiation from the lower villus to the upper intraepithelial area, where TRM are exposed to cytokines that promote their persistence". While this reviewer agrees that the data support such a model, it needs to be shown in a more direct manner and with more experimental detail. This would be important to take the study from a technical tour-de-force to include more generalizable biological insights.

Main points

1. A general concern I have with this and similar studies is the limited depth of transcriptional profiling. Only 191 genes were detected in P14 CD8 T cells that were expressed in more than 5% of the cells. Considering that many thousand genes are thought to be expressed in any given cell, this is an exceedingly small number which casts some doubts about the power of such analyses. Having said that, this study reflects (or exceeds!) the state of the field and has to be considered in this context.
2. The authors find that 87 genes (46%) had significant changes in expression along the crypt-villus axis, 76 (40%) changed along the epithelial axis, and 8 (4%) along the longitudinal axis (Fig. 2e). Their interpretation is that these results "suggested strong transcriptional imprinting based on the intratissue location of P14 CD8 T cells, influenced primarily by the crypt-villus and epithelial axes". However, I am not sure this interpretation is justified. Firstly, as outlined above, it is unclear how important "significant changes in expression" are in a situation in which only a very small fraction of transcripts is captured. Could these changes be random? Furthermore, the authors show that distribution of TRM along the crypt-villus axis changes over time post infection. Thus, another interpretation of their finding is that "time in the villus" drives differential expression, not location.
3. Overall, I have some problems understanding some of the spatio-temporal conclusions. The authors show that developing TRM cells appear first at the base of the villi before moving up to the tip over time (Fig. 2d). *Gzmb*, for example, is enriched in the TRM at the tip, while *Tcf7* is enriched in TRM at the base (Fig. 2h). That all makes sense. However, the authors also show that overall expression of *Gzmb* goes down over time, while *Tcf7* goes up. That seems counterintuitive and hard to reconcile with the first two points (Fig. 2i). How does this work?
4. Fig. 3a is hard to understand. What does it mean? What is what? Fig. 3b is meant to show cellular interaction; yet, if I understand it correctly, it simply shows proximity of cells. If that is correct, perhaps it should be described as such. Some of the labelling in this figure is a little unclear (at least to me). Does 'bottom' equal 'crypt'? What is the meaning of cytokines (Fig. 3g) for which the 'incoming signal' is 0 in all three locations (eg IL-6). Why are they part of the figure? How was the "incoming signal" determined (it would help to provide a little more detail in the main body of the text)? What is "CCL" and "CXCL"?
5. Given that multiple important cytokines, including Il10, Il7, Il21, and Il15 were found to show distinct spatial expression patterns, it is not clear why the study examines solely the role of TGFb. Presumably because TGFb is already well-known to be required for intestinal TRM. However, precisely for this reason, it appears that other cytokine signals should have been explored in some depth. This would add important novelty to the study. This also applies to CXCR6 which is mentioned in the study but not explored further.
6. The authors justifiably explore the role of TGFb in more detail. Using TGFbR-KO P14 T cells they find that TGFb plays a role in driving the TRM maturation process (shown previously) and the progression of TRM into the villi tip. While this adds a novel aspect to the well-known critical role of TGFb in TRM differentiation, the study would benefit a lot if this could be expanded. For example, the authors show that TGFb isoform expression is locally different. It would be interesting if the authors could address whether these different isoforms play distinct roles in TRM differentiation. This reviewer, however, appreciates that this may not be possible. It may, however, be easier to address the role of TGFb processing more directly. The authors show that TGFb-processing integrins are differentially expressed across distinct anatomical location and associated with specific cell types. What happens if these processing integrins are deregulated or blocked?
7. Technical point: the authors use a system in which they use Xist transcripts to track female P14 cells in a male host. While this is a clever way of doing it, it also prompts the question whether the localization and persistence of female cells in a male

host is normal. It would be re-assuring if the authors could provide evidence that this is the case. I would also suggest mentioning this in the manuscript.

(Remarks on code availability)

Version 1:

Reviewer comments:

Referee #1

(Remarks to the Author)

The authors have added a wealth of new data to the revised manuscript, with the VisiumHD and spatial CRISPR experiments taking the study to new technological heights. My previous comments have been adequately addressed and I am satisfied with the definition of Trm populations as "progenitor-like" or "differentiated". I have one remaining query based on the authors now proposing that the differentiation is "bimodal" (ie progenitor-like vs differentiated) rather than "linear" – specifically, where do recirculating memory cells fit into this bimodal mix. The model in new Fig 3g denotes that CD8 T cells in the SI early post infection are either SLEC or Trm precursors. Are Trm precursors/recirculating T cells and progenitor cells one and the same - with the circulating T capable of converting to "differentiated Trm" if they receive TGFb signals in timely/spatial fashion.

(Remarks on code availability)

Referee #3

(Remarks to the Author)

The revised version of this manuscript is easier to understand, has more clear-cut results/outcomes with the removed speculations and still presents very interesting data detailing microenvironments and their impact on CD8 T cell differentiation in a way that has not been done before. I was happy to see that some of the control experiments with uninfected mice were able to demonstrate that the microenvironments uncovered in the original manuscript were not just caused/changed by the infection, but these were the similar in uninfected animals. I am still wary about the $n = 2$ experiments with spatial transcriptomics. The other data that were provided to shore up this low n (but not an actual repeat) is great, but does not really change my concerns. The authors do point out that there is animal to animal variability in other assays and data they present, so why not in this assay? Publishing a very visible and high impact paper, as this would likely be in Nature, could be detrimental to the field if the n of 2 is not actually representative.

(Remarks on code availability)

Referee #4

(Remarks to the Author)

The authors have done a great job addressing my comments. They not only added confirmatory data but also expanded their study substantially. I am satisfied with the state of the manuscript.

A small point: perhaps the title should be adjusted to "Tissue-resident memory CD8 T Cell Diversity is Spatiotemporally Imprinted"

(Remarks on code availability)

Referee #5

(Remarks to the Author)

This very interesting manuscript describes the dynamic zonation of CD8 T cells along the villus axis following infection. By combining spatial transcriptomics, elegant data analyses and several knock-out strategies the authors identify TGFb-related signals that are responsible for homing of distinct Trm subsets to the villus bottom and top. They also decipher the signaling

molecules and central ligands responsible for the differential programming of bottom and tip T cells. The approaches are original, and the findings are intriguing and thought-provoking. The authors have done a tremendous amount of work to strengthen the paper during the revision process. I have a few comments that could improve the work:

- The authors present ample zonated signatures largely based on Xenium and VisiumHD. It would be important to add at least some QCs and comparisons to zonation patterns previously observed in the mouse epithelium along the crypt-villus axis ([https://www.cell.com/cell/fulltext/S0092-8674\(18\)31164-4](https://www.cell.com/cell/fulltext/S0092-8674(18)31164-4)) and along the duodenum-ileum axis (<https://www.nature.com/articles/s41556-023-01337-z>). It would also be important to compare the profiles between the xenium experiments and the respective genes in the VisiumHD experiment. The authors can correlate the tip/bottom ratios in expression of the genes that appear in both the VisiumHD and Xenium panel.
- There is some unclarity regarding the intestinal segments analyzed, the methods describe Swiss rolls prepared from 10cms of proximal small intestine, after removal of 3cms. This means the authors analyze duodenum and proximal jejunum. The human validation is performed on ileal biopsies. The authors should be clear regarding the segments analyze and discuss potential confounders (e.g. the possibility that the mouse ileum could exhibit different patterns than those shown).
- Line 190-192 – The uninfected intestines are claimed to also show zonation of Gzma, Gzmb and Itgae, however the shown results suggest otherwise (Figure 3g clearly shows the tip zonation of Gzma, Gzmb and Itgae and the bottom zonation of Tcf7 in the post-infection mouse intestines, whereas EDF5e does not show the same trends in the uninfected intestines. What are the patterns of these genes in the VisiumHD experiment? The authors may also want to validate on another published mouse VisiumHD dataset (<https://www.10xgenomics.com/datasets/visium-hd-cytassist-gene-expression-libraries-of-mouse-intestine>).
- The analysis of differences in localization and expression in the TGFbr2-KO cells uses 8 differentially expressed genes between the KO and WT to define a TGFb signature. Why were these 8 genes chosen, and how robust is this analysis to the gene set choice? The robustness of this analysis can be assessed by using different numbers of genes, and explicitly using clear criteria of DGE including genes with defined fold-change and q-value.
- The authors denote three enterocyte cell types (enterocyte 1,2,3), what are their gene expression markers and how do they relate to the zonated enterocyte populations described in [https://www.cell.com/cell/fulltext/S0092-8674\(18\)31164-4](https://www.cell.com/cell/fulltext/S0092-8674(18)31164-4)? Defining these according to their location along the villus axis (tip,middle,bottom) might increase clarity in some of the figures.
- Line 86-87 – provide details in Methods on the deep-learning prioritization.
- Refs 15 and 23 are duplicates.
- EDF6 - add specific references to Milner et al and Nath et al.
- The authors need to ensure all data is deposited upon publication (the current data for review did not include the Visium HD experiments). Uploading Loupe files would facilitate easier exploration.

(Remarks on code availability)

Version 2:

Reviewer comments:

Referee #3

(Remarks to the Author)

The authors have adequately addressed my previous comments.

(Remarks on code availability)

Referee #5

(Remarks to the Author)

The authors have satisfactorily addressed all of my comments.

(Remarks on code availability)

We thank the referees for their time and for their encouraging comments and constructive criticism. We are thrilled to read there is an overall consensus and excitement on the novelty and strength of the technical approach and the importance of the demonstration that T_{RM} diversity in the small intestine originates through spatial positioning over time, which we show for the first time in mice and humans.

Our spatiotemporal profiling of SI T_{RM} differentiation has numerous novel findings, which we further highlight in our revised manuscript, including: 1) previously unknown cellular sources and gradients of key immune cytokines, 2) previously unknown shifts in immune cell populations over the course of an immune response, 3) a comprehensive description of immune and non-immune cell interactions, and 4) a computational framework for analyzing spatial transcriptomics data that leverages anatomical landmarks and relationships to generate tissue-wide quantitative measures of cellular location, cell-cell interactions and gene expression. In addition, we now provide: 1) the introduction of novel optically-encoded functional perturbation method which we use to show newly appreciated biological function revealed by our spatial transcriptomic analysis, 2) the profiling of two new uninfected small intestines with an expanded gene panel (480 genes), 3) the whole-transcriptome spatial transcriptomics profiling of a mouse small intestine at 2 μm resolution, 4) immunofluorescence and flow cytometry validations supporting our key findings as well as additional statistical tests and quantifications, and 5) a streamlined results description and a refined working model of the spatiotemporal differentiation of CD8 T cells in the small intestine.

We have provided a point-by-point response to the Referee's comments below highlighting the new additions and observations that further illustrate the significant biological, technological, and conceptual advances of our work. Importantly, using our optically-encoded functional perturbation approach, our new data show how chemokine gradients within the villus control the location and the phenotype of T_{RM} cells in the small intestine. For these new experiments, we focused on the CXCR3-Cxcl9/10 axis, as suggested by Referee #4. We first found the epithelial axis generates a gradient between the IE and LP compartments, including a very strong LP-enriched Cxcl9/10 signal (**new Figure 5 and Rev. Fig. 10**). Studying the transcriptional differences and signaling across this microgradient *in situ* had not been previously possible. Second, we found *Cxcl9* and *Cxcl10* to be induced shortly after infection in a distinct area of the intestine, the base of the muscularis, which creates a secondary focal point for T cell chemoattraction. How these signaling gradients impart CD8 T cell differentiation and positioning was unknown. To address this, we designed a new methodology enabling optically-encoded pooled CRISPR screens with spatial transcriptional profiling in antigen-specific CD8 T cells. To our knowledge, this is the first time this approach has been successfully implemented. Importantly, our data showed that the loss of *Cxcr3* in P14 CD8 T cells prevents short-lived effector T cells from occupying the lower villus and muscularis, shifting the relative frequencies of the CD8 T cell pool towards the top intraepithelial compartment. The enrichment of *Cxcr3*-deficient CD8 T cells towards the top of the villi was accompanied by a further increase in expression genes associated with T_{RM} differentiation, supporting the idea that location further reinforces the phenotype induced by CXCR3 loss, and ultimately demonstrating that CD8 T cell phenotype and location within the tissue are intertwined.

Therefore, for the first time, we can link unique functional and differentiation states to microanatomical niches. Together, these findings have considerable implications for how the functional state of T_{RM} cells can be manipulated to improve barrier immunity.

We hope the referees appreciate the tremendous effort put into the revision of this manuscript. We believe the additional data and enhancements raise the broad interest and impact of this manuscript to merit publication in *Nature*. Below is a detailed point-by-point response to the referees' comments.

Referee #1:

Overall Remarks:

“The study shows that within a given organ there exists location-dependent variation in tissue-resident T cells that is (at least partly) controlled by differential TGFb signalling. The researchers exploit spatial transcriptomics to show that the Trm population in the small intestine is not binary – in other words, they do not consist of only immature polyfunctional Trm and fully mature effector Trm, but a spectrum of cells that exist along a progressing maturation continuum.

It is known that TGFb is an environmental factor required for driving Trm development and maturation in some tissues, although in organs like the liver, an absence of TGFb signalling results in a Trm population that is immature-like with more reprogramming flexibility when compared to those that have undergone full TGFb-driven differentiation. However, what is novel and important here is the demonstration that the SI Trm population exists along a TGFb-controlled maturation continuum with the most immature Trm cells found in the crypts while the most mature Trm localise to the tip of the villi – with those cells ‘in between’ varying in their state of maturation. This maturation continuum is not just spatially dependent – but also time dependent. All tissues are going to have cells at different stages of maturity early in the primary or challenge responses. Based on this, the demonstration that the Trm feature this temporal variation is of major importance. Altogether, this paper it argues that even within a given tissue not all Trm are the same and they can’t be assumed to be a homogeneous population across space and time – an assumption that dominates the literature to this point in time.”

Detailed points:

1) *I found the proposed nomenclature confusing. The authors use the term “polyfunctional memory Trm” for those in the crypt and “effector Trm” for those at or near the villus tip. I don’t think the Authors are suggesting that the later are traditional tissue circulating effector memory T cells (quite the opposite), but this nomenclature may confuse the reader and/or lead to conclusions that these cells are Tem-like. Also I do not understand how the ‘in between’ cells fit within this functionality-based paradigm. A nomenclature that may better reflect the concept of a continuum as seen in Figure 2g,h is to call the cells in the crypt “immature Trm” since they lack many of the canonical Trm markers (such as high CD103 and low Klf2) - and those in the villi “mature Trm” since they manifest the full TGFb-driven transcriptional signature.*

We apologize if our nomenclature created confusion, and we are happy to update it as suggested by the referee. We decided to refer to these two polarized cellular states as "differentiated T_{RM}" and "progenitor-like T_{RM}". Progenitor-like T_{RM}, at the lower villus area and crypt, do not necessarily lack many of the canonical T_{RM} markers, but they do express lower levels of CD103 and granzymes, and higher *Klf2*, *Slamf6*, and *Tcf7* compared to differentiated T_{RM}, located at the top of the villus. The nomenclature chosen reflects previous publications that addressed the phenotypic diversity and functional capacity of these cells (PMID: 32433949), and falls in line with the current

nomenclature chosen to reflect progenitor vs differentiated states in other contexts, such as chronic infection and cancer.

Our new set of experiments included in the revised version of this manuscript have allowed us to refine our model and address this referee's comment on the "in between cells" (**Rev. Fig. 1**). Our data show that the intestinal architecture creates a spatial polarization of T_{RM} states, which are maximized at the top IE vs the crypt LP areas, as described. Rather than a linear continuum, we see these two populations on the opposite ends of a bimodal differentiation continuum, selected for and reinforced by their respective microenvironments over time. This model has now been included in **new Fig. 3g** of the manuscript.

2) It remains unclear whether this type of variation is found in other tissues. As mentioned above, it's been shown that liver lacks TGF β -imprinted CD103⁺ Trm cells. As a consequence, might this population be relatively more homogeneous?

As the reviewer pointed out, the liver lacks TGF β imprinted CD103⁺ T_{RM} cells. However, T_{RM} cells within the liver represent a heterogeneous population (Crowl et al., 2022), with differences in expression of some canonical T_{RM} marker genes, e.g., *Klf2*, *Tcf7*, *Gzmb*, *Ifng*, suggesting, that T_{RM} cells within the liver might also exist along a differentiation spectrum, and also possibly align with anatomical and/or histological landmarks of the liver (e.g., from the periportal area via zone 1, zone 2, and zone 3 to the central vein). Every tissue likely has its own intricacies that may imprint memory T cell diversity. Assessing these phenotypic variations in relation to anatomy across tissues would require its own dedicated project. We hope our experimental and computational approaches developed in the current study provide the framework to support the study of the immune diversity of other tissues and organs.

3) TGF β is important in Trm development – but the results in Figures 3 and 4 and their interpretation is confusing. TGF β concentration declines towards the top of the villus (Fig 3d-g), yet the authors argue that TGF β program is highest for the T cells are closest to enterocytes at the top of the villus and lowest for those near the muscularis (Fig 4g). The speculation that the Trm are programmed at the base of the villus to be positioned at the top seems like a long bow to draw.

The questions that we sought to answer with our unique dataset are: 1) what cells are sources of TGF β , and potential intermediaries in its signaling (mediated by presentation), 2) what is the influence of TGF β on transcriptional programming of SI CD8 T cells, and 3) how does the lack of this signaling (due to TGF β RII deficiency), impact the intratissue location of CD8 T cells? To our surprise, we found a clear gradient along the crypt-villus axis for TGF β isoform expression and isoforms were also produced largely by different cell types. In addition, we were able to comprehensively describe potential cell types mediating the presentation of TGF β 's bioactive form. To understand what the relevant sources for TGF β signaling in P14 CD8 T cells are, we looked

Rev. Figure 2. Proposed working model for the TGF β -dependent differentiation of CD8 T cells in the SI at an effector timepoint.

at how the TGF β programming changed in WT P14 CD8 T cells as a function of distance to every other cell type (**Fig. 4g**). This revealed that P14 CD8 T cells had the highest TGF β gene signature when found closest to enterocytes in the upper villus area. Importantly, in the absence of TGF β RII, P14 CD8 T cells are more frequently found near fibroblast populations in the muscularis and lower villus area, which are both producers and presenters of TGF β (**Fig. 4h**). Thus, we can conclude TGF β sensing is required for CD8 T cells to populate the upper area of the villus, an environment with increased abundance of pro-survival cytokines, such as Il-7 and Il-15 and where TGF β is likely provided by different cells than at the crypt or muscularis (**Rev. Fig. 2 and new Fig. 4i**). We realize that our previous text was speculative and have modified the current version to provide a more objective and balanced description of the results and moved our additional ideas to the discussion.

Genetic follow-up studies are needed to pinpoint the exact location of the physiological source of TGF β for intestinal CD8 T cell maturation and lie outside of the scope of this study. Nevertheless, in this study, we are for the first time able to show the different sources and spatial availability of TGF β isoforms, and show, that antigen specific T cells have the highest TGF β gene signature, when embedded in the intestinal epithelium, and thus, significantly advance our understanding of how TGF β isoforms instruct CD8 T cell differentiation in the small intestine.

Referee #3:

Overall remarks:

“This is a very data dense manuscript examining microanatomical P14 CD8 T cell localization and phenotype in the small intestine. The authors use cutting edge techniques, such as spatial transcriptomics, to simultaneously score, phenotype and locate P14 within the small intestine at different times after LCMV-Armstrong infection. Human ileum were also analyzed which adds an important component to this manuscript. I understand that the prepping of samples and data analysis is a huge effort and, to my knowledge, this is the first time these sort of approaches have been used to examine Ag-specific CD8 T cells in the small intestine. This makes this manuscript and the data very innovative and significant.

However, there are some important considerations in their studies that are missing, which make it hard to evaluate the data.”

Detailed points:

1) Most of the data is based on 2 biological replicates. While I understand that this is likely due to the amount of time and effort it takes to produce data from one n, reproducibility can't be forgotten just because of the technique. Also, because of this, there are not a lot of statistical analyses. The authors state that their n=2 are reproducible, but Ext Fig 2b shows that is not true, especially at day 6 and 8. The authors also state that P14 are equally distributed longitudinally (line 116), but this figure leads one to question that statement.”

We appreciate the referee for recognizing the time and effort it takes to generate and analyze these types of data, and we agree on the need to uphold statistical rigor, despite the costs. While the field has not yet decided on gold standards for the experimental design and statistical analysis of spatial transcriptomics data, we decided on biological experimental duplicates based on available data (Wang et al. 2023 BioRxiv, Cook et al. 2023 BioRxiv, Hartman et al 2023 BioRxiv) and our own data, which show very good concordance between biological and technical replicates. We have included additional QC data with: "Total counts per gene (log10) Pearson Correlation with Replicate", "Cell type frequency (log 10) Pearson Correlation with Replicate", "Median total transcripts per cell type (log) Pearson Correlation with Replicate", and "Median unique genes per cell (log)

Pearson Correlation with Replicate", and "Number of total cells profiled" in **Supplemental Table 2**. In addition, we have QC data in **new Extended Data 1d, 1h, 5a, 8b, and 8c**. Overall, biological and technical replicates had excellent Pearson correlation coefficients across multiple metrics.

The figure mentioned by the referee showed total P14 CD8 T cell numbers, which can differ from replicate to replicate due to differences in the overall tissue area imaged. **New Extended Data Fig. 3a** showing cell frequency rather than absolute cell counts for each intestinal segment shows a smaller variation across replicates.

Ultimately, we acknowledge that spatial transcriptomics, which images a 5 μm sections of tissue, is not the best technology to make claims on total cell numbers for an organ, but it does provide insights on relative cell numbers. Please note that we have added flow cytometry quantification of total cell numbers in the tissues for the reported increase in P14 CD8 T cells, and other immune cells in **new Extended Data Fig. 1g**.

2) Many techniques used here are not standard for most investigators. Uninfected or no transfer control mice are not shown. Using Xist as a marker of T cells is clever, but no data are presented to demonstrate that Xist expression is constant during T cell activation.

We have calculated *Xist* transcript expression across *Xist*⁺ Cd8 $\alpha\beta$ T cells at each timepoint normalized to the mean transcripts per cell (**Rev. Fig. 3 and new Extended Data Fig. 1h**). *Xist* expression values in *Xist*⁺ Cd8 $\alpha\beta$ T cells across time points can be used to infer the variability of P14 CD8 T cell detection efficiency due to changes in *Xist* expression. *Xist* expression in detected CD8 $\alpha\beta$ T cells does not statistically vary with time, with the exception of day 6 (**Rev. Figure 3**). We may have detected *Xist*⁺ CD8 T cells at a ~10% higher rate at day 6 than at the other time points, which does not change the main conclusions of this study.

The uninfected sample was a missing control, and we appreciate the referee's suggestion. We have now included two uninfected control samples. To do this, we developed an upgraded 480 gene panel (the previous panel was no longer available) and profiled two uninfected SI controls using Xenium. The data are in **new Extended Data 5 (Rev. Fig. 4)**. Uninfected SI showed a similar distribution of key cytokine expressions, including *Tgfb1/2/3*, *Cxcl9/10*, *Il18*, and *Il15* (**new Extended Data Fig. 5b and Rev. Fig. 4b**), a similar connectome (**Extended Data Fig. 5c and Rev. Fig. 4c**), and a similar spatial polarization of CD8 $\alpha\beta$ T cells based on the expression of differentiated T_{RM} (*Gzma*, *Gzmb*, *Itgae*), and progenitor-like T_{RM} (*Tcf7*, and *Id3*) (**Extended Data Fig. 5d and 5e and Rev. Fig. 4d and 4e**). Together, these data greatly strengthen our conclusions. In addition,

these new data have allowed us to test whether the signaling gradients observed existed prior to the infection onset or were induced. This has led to the new finding and follow up, included in **new Fig. 5**, exploring the role of inducible areas of T cell chemoattraction, further discussed to referee #4, point 5.

3) In many figures, it is hard to understand what was really done, what time points the data are referring to, etc. E.g., in Fig 2d, are the cell numbers just in the one slice of the entire Swiss roll? Is there a calculation to account for the entire intestine?

In **Fig 2d**, the cell numbers are just from one slice of the small intestine roll, as every Xenium experiment is performed on a 5 μm section of the total tissue. Obtaining a reliable estimate for the cell numbers in the entire tissue using spatial transcriptomics is not yet possible.

Fig 2g – legend says all cells in the timecourse. Does that mean that P14 at all timepoints are analyzed together here? If so, why?

Yes, that panel had all timepoints combined. We realize this was a source of confusion and have now provided the convolved gene-expression gradients for each axis and time point (**new Fig. 2f and 2i**), in addition to improving our annotation of the figures, figure call outs and figure legends.

Fig 4a legend does not say it is day 8, only in text.

We apologize for this omission, it has now been corrected.

4) *There is a lot of speculation in this manuscript, e.g. line 196 and lines 282-285 about TGF β . The data show that cells localize differently when P14 do not sense TGF β , but the authors build a whole pathway of differentiation from this observation.*

Other referees have also raised this point, and we agree that it is an important point to clarify. The questions that we sought to answer with our unique dataset are: 1) what cells are sources of TGF β , and potential intermediaries in its signaling (mediated by presentation), 2) what is the influence of TGF β on transcriptional programming of SI CD8 T cells, and 3) how does the lack of this signaling (due to TGF β RII deficiency), impact the intratissue location of CD8 T cells? To our surprise, we found a clear gradient along the crypt-villus axis for TGF β isoform expression and isoforms were also produced largely by different cell types. In addition, we were able to comprehensively describe potential cell types mediating the presentation of its bioactive form. To understand what the potentially relevant sources for TGF β signaling in P14 CD8 T cells are, we looked at how the TGF β programming changed in WT P14 CD8 T cells as a function of distance to every other cell type (**Fig. 4g**). This revealed that P14 CD8 T cells had the highest TGF β gene signature when found closest to enterocytes in the upper villus area. Importantly, in the absence of TGF β RII, P14 CD8 T cells are more frequently found near fibroblast populations in the muscularis and lower villus area, which are both producers and presenters of TGF β (**Fig. 4h**). Thus, we can conclude TGF β sensing is required for CD8 T cells to populate the upper area of the villus, an environment with increased abundance of pro-survival cytokines, such as Il-7 and Il-15 and where TGF β is likely provided by different cells than at the crypt or muscularis (**Rev. Fig. 2 and new Fig. 4i**). We realize that our previous text was speculative and have modified the current version to provide a more objective and balanced description of the results and moved our additional ideas to the discussion.

5) *There is little validation of what the authors discovered using more standard methods, such as flow or immunofluorescence on tissue sections. While Ext Fig 1, shows % cells after enzymatic digestion (again n=2), there is no quantification of cell numbers here to try to match or understand the numbers of P14 in Fig 2d.*

We agree with the reviewer and have addressed this point in two key areas.

First, we have performed flow cytometry-based quantification of total cell numbers at multiple time points after LCMV infection in the spleen and SI (IE and LP compartments) (**new Extended Data Fig. 1g**). This will serve as validation of our Xenium cell type annotations and frequencies of the reported immune cell types.

Second, we have performed immunofluorescence validation for TCF1 and GZMB expression in the small intestine of a mouse 30 days after LCMV infection to show TCF1⁺ and GZMB⁺ CD8 α β T cells are preferentially found at the crypt and upper villus area, respectively (**new Extended Data Fig. 3c and Rev. Fig. 5**).

Rev. Figure 5. Immunostaining of CD8 α , TCF1 and GZMB in day 30 p.i small intestine.

6) *Some of the discoveries in this manuscript are surprising, e.g., stable CD103 expression over the day 6-day 90 timepoints, as there are published flow data that CD8 T cells enter the SI as CD103 low and upregulate CD103 over time.*

The reviewer raises an important point, and we do apologize that our old Figure 2i might have been misleading, as it made it look like *Itgae* expression did not increase from day 6 to day 90. It is important to note that this figure (**new Extended Data Fig. 3d**), shows the expression of *Itgae* of P14 CD8 T cells within a defined spatial gate (muscularis, crypt and top), where expression of *Itgae* by P14 T cells at the top and the crypt is relatively high already at early time points after the infection and this is maintained over time. However, as we show in the manuscript, P14 CD8 T cells have a preferential shift in location over time from the muscularis and lower villus at the effector phase to the upper villus area at the memory phase. Hence, in addition to the expression, one must take into account the changes in cell frequency within the spatial gates over time. Importantly, we see numerous P14 CD8 T cells in the muscularis at day 6, but not at later time points (**Fig. 2d**). P14 CD8 T cells in the muscularis express very low levels of *Itgae* mRNA (**Extended Data Fig. 3d**). If we compare *Itgae* expression on a total P14 CD8 T cell population level, we do see an increase from day 6 to day 8, partially mediated by the decrease in cells in the muscularis (**Rev. Fig. 6**), in line with the previously published data that this referee mentions. We apologize again, that this figure might have been insufficiently discussed. We have improved the clarity of our message in the manuscript.

Rev. Figure 6. Mean mRNA expression of *Itgae* in all P14 T cells detected in our spatial transcriptomics datasets at day 6 and 8 post LCMV Armstrong infection (two replicated per time point)

Low expression of granzyme at late timepoints post infection is also surprising, as ex vivo lytic activity of CD8 T cells in SI has been demonstrated previously.

We agree this seems paradoxical, and that the way that we plotted expression levels comes across as unclear. But, expression of numerous effector genes does go down in memory cells compared to effector cells. Levels of *Gzmb* RNA transcripts (and *Gzma*) for SI P14 T cells is in fact lower in late timepoints than early time points, as also seen in data from our previously published single-cell RNAseq time-course studies (**new Extended Data 3e and Revision Fig. 7a**). However, we realize the y-axis is standard scaled with no definition of ‘min’ and ‘max’ so the readers do not know how much lower ‘min’ is than ‘max’. We defined the axes of this plot in the legend to avoid potential confusion. In addition, we provide gene expression quantification for the cells in the IMAP gates (**new Fig. 2g, 2h, and 2j and Rev. Fig. 7b**).

Rev. Figure 7. a, Gene expression of indicated genes for intestinal P14 CD8 T cells over time after an LCMV infection profiled by scRNA-seq (Kurd et al. 2020). **b**, IMAP gene expression for day 90 pi samples, with gene expression quantification.

Equal numbers of P14 at day 6 and day 8 are documented in this manuscript with their techniques, yet that would not match published data. While what is discovered here may be correct and data from others may be the result of sub-optimal tissue isolation techniques, there should be some validation done as part of this study.

Our spatial data suggests – in agreement with the literature – an increase of total T cell numbers from day 6 to day 8 (**Extended Data figure 1f**). However, as the reviewer pointed out, we do not see a drastic increase in P14 cell numbers between day 6 and day 8. There are multiple possible explanations for this. It is possible total cell numbers can differ from time point to time point due to differences in the overall tissue area imaged. We acknowledge spatial transcriptomics is not the best approach for total cell determination, and we have provided flow cytometry quantification to support the statements in our study (please see point #5 above).

*7) The differentiation pattern of P14 is assumed to be driven just by the genes/proteins they find in their assays. However, they do not discuss the impact of Ag recognition in the SI by P14. While virus is thought to be cleared by day 6/8, Ag may still be present. P14 in different locations appear to differentially regulate *Ki67*, *Thy1* expression and type II interferon which are associated with T cell signaling, proliferation and effector function due to Ag. It would be interesting to do these same experiments using a system where Ag is not present in the SI.*

It is our goal to describe the differentiation of T_{RM} cells induced by the response to infection, thus antigen is a key component of the system. Passive differentiation of T cells or the induction of T_{RM} in response to microbiome etc. is a very intriguing separate question, yet beyond the scope of this manuscript.

- Refs 15, 16 seem to be in the wrong place in paragraph 1. Thank you, we have corrected this.

- Line 150 – did they mean “with” and not “within”? Yes.

- *The title states functional diversity, but no actual functions are tested here, as stated in line 104. they capture transcriptional regionalization (going back to point 4). We will remove “functional” from the title.*

- *The authors often state that cells at the top are in contact with epithelial cells (e.g., line 157). While that may be true for IEL, there is a basement membrane under the epithelium. It is not clear that cells in the LP in that area would contact epithelial cells. We appreciate the correction, and we have modified the wording throughout the manuscript.*

- *The “top” and “bottom” designations are not very specific (Fig 2a) and depending on the length of the villus (very long on day 8), half of the villus could be “top” (red). Unclear how these were designated.*

Top and bottom does not change based on the length of the villus. It is entirely standardized because it is transferred across experiments using a predictive model based on cellular neighborhood. We have clarified this in the main text and add more specific wording when referencing top and bottom positions. We have added labels to **Fig. 1a** to improve clarity.

-*The authors assume that once a T cell enters a particular villus, it does not leave that villus (to go to another villus – I do not mean back into the circulation). The perceived differentiation pathways are based on this assumption, but it has not been shown that this single villus residency is the case and some do speculate that T cells move between villi.*

Based on current evidence presented by the Mucida, Masopust and Vezys labs, CD8 T cells entering the small intestine are initially very motile but become increasingly restricted as they differentiate (observed up to day 30 after infection). In short, SI T_{RM} cells do not frequently change location between different areas of the small intestine, such as from one villus to another, under normal conditions (PMID: 30865878 and PMID: 28942917). While definitive proof is lacking, as these studies cannot track these cells for long periods of time, our data agree with the current observations provided by other laboratories.

Referee #4:

Overall Remarks

“In this study, the Goldrath lab makes use of spatial transcriptomics to examine TRM differentiation in the small intestine. They use human samples and a murine model of acute systemic viral infection to profile the location and transcriptome of TRM cell differentiation at single-transcript resolution. They develop computational approaches to capture cellular locations and visualize the spatiotemporal distribution of cell types and gene expression. They show TRM populations were spatially segregated, with more effector and memory-like TRM preferentially localized at the villus tip or crypt, respectively. Using a computational approach of modeling ligand-receptor activity suggests patterns of cellular interactions and cytokine signaling pathways that initiate and maintain TRM differentiation and functional diversity, including different TGF β sources. Alterations in the cellular networks induced by loss of TGF β receptor expression revealed a model consistent with TGF β promoting progressive TRM maturation towards the villus tip. Overall, the authors have developed a framework for the study of immune cell interactions with the spectrum of tissue cell types, revealing that T cell location and functional state are fundamentally intertwined.

*This manuscript builds on a long list of studies by the Goldrath lab that have examined TRM differentiation in great detail, in particular in the small intestine. Here the authors take their studies to the next level using new tools that allow at an unprecedented depth an illustration of the processes that results in TRM differentiation and diversification. Although one could argue that **novel insights into TRM differentiation are limited**, it is clear*

that the authors make exemplary use of state-of-the-art technology and analysis methods, which are likely to be of great interest to many research groups.

While this study overall is great, it would be important to provide additional biological data to underpin the main conclusions. Specifically, the authors state “Our findings support a model in which the cellular organization of the intestine provides localized instructions through regionalization of cytokine secretion and distinct cellular interactions that lead TRM differentiation from the lower villus to the upper intraepithelial area, where TRM are exposed to cytokines that promote their persistence”. While this reviewer agrees that the data support such a model, it needs to be shown in a more direct manner and with more experimental detail. This would be important to take the study from a technical tour-de-force to include more generalizable biological insights.”

Detailed points

1. A general concern I have with this and similar studies is the limited depth of transcriptional profiling. Only 191 genes were detected in P14 CD8 T cells that were expressed in more than 5% of the cells. Considering that many thousand genes are thought to be expressed in any given cell, this is an exceedingly small number which casts some doubts about the power of such analyses. Having said that, this study reflects (or exceeds!) the state of the field and has to be considered in this context.

The use of single-cell resolution spatial transcriptomics provides a substantial leap forward, as presented in our work—current approaches rely on multiplex antibody staining where 10-30 targets can be detected with spatial consideration—here, we are able to increase that number by 10-fold. While this approach does not reveal global gene expression, neither does scRNAseq where there can be significant sparsity in signal for many transcripts. Therefore, we appreciate the recognition that our approaches reflect, or even exceed, the state of the field. We do, however, acknowledge that the technology's limitation is its reduced gene panel size. We designed our gene panel to focus on immune cell heterogeneity and ligand-receptor pairs, giving us a higher-resolution look at T cells spatially relative to other spatial transcriptomics studies.

However, we have now included a higher resolved secondary whole-transcriptome sequencing-based approach (VisiumHD, **new Extended Data 4 and Rev. Fig. 8**) to validate some of our findings concerning gradients of cytokine expression along the crypt-villus axis. This additional analysis showed good concordance of gene-expression patterns with our Xenium experiments.

Rev. Figure 8. a, VisiumHD results on a day 8 p.i. mouse small intestine roll colored by graph-based clustering. **b**, VisiumHD spots colored by imputed crypt-villus axis values. **c**, Convolved gene expression of cytokines along the imputed crypt-villus axis in the VisiumHD dataset. Red labels indicate genes that were included in the Xenium gene panel. **d**, Transcript reassigning based on H&E nuclei segmentation to achieve single-nuclei level gene expression data (left) and example (right). **e**, Incoming signals across CD8αβ regional subtypes in the VisiumHD dataset.

2. The authors find that 87 genes (46%) had significant changes in expression along the crypt-villus axis, 76 (40%) changed along the epithelial axis, and 8 (4%) along the longitudinal axis (Fig. 2e). Their interpretation is that these results “suggested strong transcriptional imprinting based on the intratissue location of P14 CD8 T cells, influenced primarily by the crypt-villus and epithelial axes”. However, I am not sure this interpretation is justified. Firstly, as outlined above, it is unclear how important “significant changes in expression” are in a situation in which only a very small fraction of transcripts is captured. Could these changes be random? Furthermore, the authors show that distribution of TRM along the crypt-villus axis changes over time post infection. Thus, another interpretation of their finding is that “time in the villus” drives differential expression, not location.

It is true that the choice of gene panel will impact the percentage of significant expression changes along different axes when doing multiple test correction on Spearman rank correlation p-values. In **new Fig 2e** and **2f**, we defined a gene as “correlated” based on a correlation coefficient cutoff rather than a p-value cutoff. Adjusted p-values change with the gene panel size and composition, whereas correlation coefficients do not. We will change

the wording in this part of the text for better precision in our claim. Of note, we found similar gene-expression gradients in two additional uninfected small intestines profiled with Xenium using a newer upgraded 480 gene panel (**new Extended Data 5 and Rev. Fig. 4**), and a highly resolved sequencing-based approach (VisiumHD) (**Extended Data 4 and Rev. Fig. 8**). Together, these data show strong transcriptional differences correlated to their spatial location within the villus. The causality of these changes is explored in response to point #5.

Our new results and feedback from the referees have helped us refine our model and interpretations. We agree with the referee that time in the tissue will be important for CD8 T cells to become habituated to the new environment (as opposed to circulation or lymphoid organs) (**Rev. Fig. 1**). Our data show that the intestinal architecture creates a spatial polarization of T_{RM} states, which are maximized at the top IE vs the crypt LP areas, as described. Rather than a linear continuum, we see these two populations as a bimodal distribution on the opposite ends of a bimodal differentiation space, selected for and reinforced by their respective microenvironments over time. This model has now been included in **new Fig. 3g** of the study (**Rev. Fig. 1**).

3. Overall, I have some problems understanding some of the spatio-temporal conclusions. The authors show that developing TRM cells appear first at the base of the villi before moving up to the tip over time (Fig. 2d). Gzmb, for example, is enriched in the TRM at the tip, while Tcf7 is enriched in TRM at the base (Fig. 2h). That all makes sense. However, the authors also show that overall expression of Gzmb goes down over time, while Tcf7 goes up. That seems counterintuitive and hard to reconcile with the first two points (Fig. 2i). How does this work?

We apologize for the confusion in the way we presented our data. We realize the source of this confusion comes from conflating short-lived effector cells and T_{RM}-precursors at effector times with polarized T_{RM} states at memory time points. Our new model streamlines our conclusions, please see **Rev. Fig. 1** and **new Fig. 3g**) and our reply to the point above. Our data show a decrease of *Gzmb* expression over time, which makes sense as the infection is cleared. The overall amount of *Gzmb* and *Gzma* transcript molecules goes down in P14 CD8 T cells in the intestine over time as these cells undergo memory T cell differentiation (**Rev. Fig. 7a**). Conversely, *Tcf7* expression comes up over time as a group population. Despite these overall changes at the population level, the polarization along the crypt-villus axis for each time point was retained (**Rev. Fig. 7b**). These data have been included in **new Fig. 2g, 2h and 2j** and addressed in the main text.

4. Fig. 3a is hard to understand. What does it mean? What is what? Fig. 3b is meant to show cellular interaction; yet, if I understand it correctly, it simply shows proximity of cells. If that is correct, perhaps it should be described as such.

Figure 3a is meant to display an example of the cellular interactome analysis of spatial nearest neighbors. The reviewer is correct this is based on spatial proximity. We will clarify this point in the main text.

Some of the labelling in this figure is a little unclear (at least to me). Does 'bottom' equal 'crypt'?

Yes, we have clarified our nomenclature throughout the text and included labels to **Fig. 1a**.

What is the meaning of cytokines (Fig. 3g) for which the 'incoming signal' is 0 in all three locations (eg IL-6). Why are they part of the figure? How was the "incoming signal" determined (it would help to provide a little more detail in the main body of the text)? What is "CCL" and "CXCL"?

We apologize for the confusion. We cropped the original figure for clarity, while the entire uncropped version was in the Extended Data. We have re-done this analysis to split the incoming/outgoing signaling by time point to better capture the differences in signaling across time and space (**new Fig. 3f**) and have added better figure legends and strength. We have defined the interactions comprising each signaling pathway (CCL, CXCL, and all others) and their contribution to the pathway strength in a new **Supplemental Table 4, 5, and 11**.

5. Given that multiple important cytokines, including *Il10*, *Il7*, *Il21*, and *Il15* were found to show distinct spatial expression patterns, it is not clear why the study examines solely the role of *TGFb*. Presumably because *TGFb* is already well-known to be required for intestinal TRM. However, precisely for this reason, it appears that other cytokine signals should have been explored in some depth. This would add important novelty to the study. This also applies to *CXCR6* which is mentioned in the study but not explored further.

We believe this to be a key aspect of our revised manuscript, and we appreciate the referee for encouraging us to further explore biological relevance of our datasets.

CXCR3 ligands, *Cxcl9* and *Cxcl10*, are known T cell chemoattractants that contribute to T cell positioning and maturation in the SI (PMID: 25706747). However, how are these signals distributed in the SI to program CD8 T cell fate is unclear. By looking at the expression of these two chemokines over the course of infection, we found that while *Cxcl9* and *Cxcl10* signals were heavily enriched in the top half of the LP at homeostasis, their expression was strongly induced after infection in Complement Fibroblasts (C3-expressing fibroblasts, also termed adventitia fibroblasts), which are located at the bottom of the muscularis, creating a second potential attraction point for *Cxcr3*-expressing CD8 T cells during infection (**new Fig. 5a and 5b and. Rev. figure 10a and 10b**). To test the role of these gradients on CD8 T cell location and differentiation, we used a CRISPR Cas9 approach to induce *Cxcr3* deletion in Cas9 P14 CD8 T cells. To optically identify sgRNA-containing cells within the Xenium assay we introduced a pseudogene barcode within the 3'UTR region of the mCherry reporter used

Rev. Figure 10. a, Dotplot of *Cxcl9* and *Cxcl10* expression for indicated cells in uninfected and infection time course. **b**, Gene expression trends for *Cxcl9/10* separated by timepoint (n = 2 biological replicates pooled) with representative expression depicted spatially at their positions on villus. **c**, sgRNA-containing P14 CD8 T cells (yellow arrows, middle) shown spatially within the intestinal villus at three levels of magnification. Colors by graph-based clustering (left). Red line indicates nuclear segmentation (right). **d**, Gene expression for each respective sgRNA-containing P14 CD8 T cells. **e**, IMAP representation for each respective sgRNA-containing P14 CD8 T cell population with annotated percentages within each gate. **f**, Gene expression for each respective sgRNA-containing P14 CD8 T cells split by spatial gate.

to sort the modified Cas9 P14 CD8 T cells. We also included Cas9 P14 CD8 T cells containing *Cd19*- and *Thy1*-

targeting sgRNAs as controls (**new Supplemental Table 9**). After validation of their respective target genes by flow cytometry (**new Extended Data Fig. 7a**), sg*Cxcr3*, sg*Cd19* and sg*Thy1* P14 CD8 T cells were pooled at equal frequencies and transferred into mice followed by LCMV infection (**new Fig. 5c and. Rev. Figure 10c**). The pseudogene barcodes, unique for each sgRNA, used the same landing sequences as three genes included in the 350 gene panel that were the least expressed across all cells and undetectable in T cells (**Extended Data Fig. 7b**). At day 8 after infection, two different SI were profiled by Xenium. Unique pseudogene barcodes were found almost exclusively in CD8 T cells (perturbed P14 CD8 T cells), and detected uniformly across the crypt-villus axis (**new Fig. 5e and. Rev. figure 10e and Extended Data Fig. 7b-d**). sgRNA-containing P14 CD8 T cells showed a significant decrease in their respective targeted genes (**new Fig. 5e and. Rev. figure 10e**). sg*Cxcr3*-containing P14 CD8 T cells had a marked upward shift in their villus positioning, with an enrichment in the top IE area of the villus, reduction in the LP, and virtually no presence in the muscularis (**new Fig. 5f and. Rev. figure 10f and Extended Data Fig. 7e**). Analysis of gene expression by spatial gates showed that *Cxcr3*-deficient P14 CD8 T cells had increased *Gzma* and *Itgae* expression and lower *Klf2* than control cells in certain regions (**new Fig. 5g**). Thus, not only loss of *Cxcr3* induced a preferential accumulation of cells in the top IE area, but this positional shift further reinforced a more differentiated T_{RM} phenotype (**new Fig. 5g**). The Bevan lab has previously shown that loss of *Cxcr3* depletes short-lived effector populations and is required for the formation of CD103⁺ T_{RM} cells in the LP without affecting IE T_{RM} numbers (PMID: 25706747). Our data adds to this model of differentiation by showing that short-lived effectors, are likely preferentially located in the lower half of the villus, crypt and stroma areas, attracted by *Cxcl9/10*-expressing immune and stromal cells in the LP and muscularis early after infection. Together, these data highlight the causal connection between CD8 T cell location and phenotype and highlights a previously unknown set of spatial relationships that enables coordinating new immune infiltrate CD8 T cell recruitment to areas of inflammation while preserving the existing gradients in the villus.

We believe these new data adds additional strong conceptual advance to our study and does so by employing a first-of-its-kind optically-read pooled-perturbation approach, which, we believe, will be of high interest to the larger immunology community.

6. The authors justifiably explore the role of TGFβ in more detail. Using TGFβR-KO P14 T cells they find that TGFβ plays a role in driving the TRM maturation process (shown previously) and the progression of TRM into the villi tip. While this adds a novel aspect to the well-known critical role of TGFβ in TRM differentiation, the study would benefit a lot if this could be expanded. For example, the authors show that TGFβ isoform expression is locally different. It would be interesting if the authors could address whether these different isoforms play distinct roles in TRM differentiation. This reviewer, however, appreciates that this may not be possible. It may, however, be easier to address the role of TGFβ processing more directly. The authors show that TGFβ-processing integrins are differentially expressed across distinct anatomical location and associated with specific cell types. What happens if these processing integrins are deregulated or blocked?

We appreciate the referee's comment on the rationale for exploring the biology of TGFβ signaling on P14 CD8 T cells. The pleiotropic effects of TGFβ presentation via integrins such as *Itgav* through different cell types make studies using genetic deletion of *Itgav* unfeasible due to phenotypes such as sprouting angiogenesis and intestinal hemorrhaging. The use of cell type-specific Cre-driven KO models also induces toxicities, such as ulcerative colitis and autoimmunity, when deleted from myeloid cells (PMID: 17895374), which prevents making any conclusions on the effect of this signaling axis on P14 CD8 T cells. Alternatively, pharmacological targeting using *Itgav*-directed antibodies or peptides would have pleiotropic effects in vivo, making the analysis of specific effects on P14 CD8 T cells hard to disentangle. Targeting the receptor (TGFβRII) in P14 CD8 T cells is the cleanest approach to perturbing this signaling pathway intrinsically.

7. *Technical point: the authors use a system in which they use Xist transcripts to track female P14 cells in a male host. While this is a clever way of doing it, it also prompts the question whether the localization and persistence of female cells in a male host is normal. It would be re-assuring if the authors could provide evidence that this is the case. I would also suggest mentioning this in the manuscript.*

We agree with this referee this is an important point. Before conducting our spatial transcriptomics experiments, we tested the ability of female P14 CD8 T cells to compete with male P14 CD8 T cells in establishing immunological memory and residency in the intestine by a mixed transfer experiment in the context of LCMV infection (**new Extended Data Fig. 1j-k and Rev. Fig. 11a**). We did not observe a significant defect for female P14 CD8 T cells to become SI T_{RM} at day 30 after infection (**Rev. Fig. 11b**), nor we observed differences in the quality of female memory P14 CD8 T cells compared to male P14 CD8 T cells assessed by the expression of CD69 and CD103 in the spleen or the small intestine by flow cytometry (**Rev. Fig. 11c**). These data show that frequencies and phenotypes of female P14 CD8 T cells are comparable to male P14 CD8 T cells, supporting our use of a female into male transfer system. These data have been added to **Extended Data 1**.

Rev. Figure 11.

(a) Experimental approach using a 1:1 mixed transfer of male and female P14s into infected male recipient mice. **(b)** The log₂ female to male P14 abundance ratio in the spleen and IE after day 30 takedown. **(c)** Percentage of male and female P14s gated to the subsets CD103⁻CD69⁻, CD103⁻CD69⁺, and CD103⁺CD69⁺ in the spleen and SI IEL.

Revision 2 10/03/2024

We thank the reviewers and editors for their careful consideration and constructive comments in review of our manuscript. We are glad that the reviewers agree that the manuscript has improved significantly over the course of revision. Below we address the remaining specific points raised by the reviewers.

Referees' comments:

Referee #1 (Remarks to the Author):

The authors have added a wealth of new data to the revised manuscript, with the VisiumHD and spatial CRISPR experiments taking the study to new technological heights. My previous comments have been adequately addressed and I am satisfied with the definition of Trm populations as "progenitor-like" or "differentiated". I have one remaining query based on the authors now proposing that the differentiation is "bimodal" (ie progenitor-like vs differentiated) rather than "linear" – specifically, where do recirculating memory cells fit into this bimodal mix. The model in new Fig 3g denotes that CD8 T cells in the SI early post infection are either SLEC or Trm precursors. Are Trm precursors/recirculating T cells and progenitor cells one and the same - with the circulating T capable of converting to "differentiated Trm" if they receive TGFb signals in timely/spatial fashion.

Response: We thank the reviewer for the positive evaluation of our manuscript. We believe that the distribution that we ultimately see in the T_{RM} population at memory time points (day 30 and day 60) is best reflected by a bimodal distribution. In this setting, there is strong evidence, that the progenitor-like cells are not circulating memory cells. This is based on two observations: (1) In parabiosis experiments (*Steinert et al., 2015, Figure 3*) the vast majority (> 98%) of antigen-specific T cells in the small intestine is permanently resident. (2) In fate-mapping mice using a CD103-CreERT2 mouse (*von Hoesslin et al, 2022, Figure 4*), it was shown, that even in a rechallenge experiment (1st Lm-OVA, 2nd VSV-Ova) preexisting CD103⁺ T cells remain organ-confined during and after secondary infections. Together, these two important studies argue, that the progenitor-like T_{RM} cells that we see at day 30/90 and which are still CD103⁺ albeit lower than the differentiated T_{RM} cells, are not (re)circulating memory T cells. However, we acknowledge that there may be a fraction of circulatory memory cells (< 2%) present in the tissue at these time points. Unfortunately, we do not have a method to mark these circulatory memory cells experimentally, but given their small contribution to the overall memory T cell population in the intestinal tissue, their impact on our analysis is minimal.

Referee #3 (Remarks to the Author):

The revised version of this manuscript is easier to understand, has more clear-cut results/outcomes with the removed speculations and still presents very interesting data detailing microenvironments and their impact on CD8 T cell differentiation in a way that has not been done before. I was happy to see that some of the control experiments with uninfected mice were able to

demonstrate that the microenvironments uncovered in the original manuscript were not just caused/changed by the infection, but these were the similar in uninfected animals.

I am still wary about the $n = 2$ experiments with spatial transcriptomics. The other data that were provided to shore up this low n (but not an actual repeat) is great, but does not really change my concerns. The authors do point out that there is animal to animal variability in other assays and data they present, so why not in this assay? Publishing a very visible and high impact paper, as this would likely be in Nature, could be detrimental to the field if the n of 2 is not actually representative.

Response: We are glad the reviewer sees a pronounced improvement in our manuscript, and we agree that adding the uninfected controls has substantially strengthened our findings and interpretations.

We strongly believe that scientific rigor and reproducibility are required prerequisites for all scientific publications. We appreciate the referee for recognizing the time and effort it takes to generate and analyze these types of data, and we agree on the need to uphold statistical rigor, despite the costs. We have analyzed one additional spatial transcriptomics experiment from a mouse small intestine at day 7 after LCMV Armstrong infection that was profiled using the same 350 gene panel. The data recapitulate the findings from both our day 8 p.i. samples related to average gene expression levels (**Reviewer Figure 1**), and the polarization of P14 CD8 T cells across the crypt-villus axis (**Reviewer Figure 2**). We provide these data to the referees and ask that it not be included in the manuscript as it is part of an ongoing additional study.

Reviewer Figure 1: Correlation analysis of average gene abundances between the day 7 p.i. SI (new) and the two day 8 p.i. SI samples included in our timecourse analysis.

Reviewer Figure 2: Correlation analysis of villus top/bottom ratios between the day 7 p.i. SI (new) and the two day 8 p.i. SI samples included in our timecourse analysis.

We would like to emphasize that our experimental design's strength lies in using a time series, now also including uninfected, in addition to having biological duplicates and orthogonal validation assays such as VisiumHD, and similar observations in two individual human patient samples. We have added further validation of our data by correlating the top/bottom ratios in gene expression between Xenium and VisiumHD, as suggested by reviewer 5 (**Extended Data Figure 4c** and **Reviewer Figure 3**), which shows an excellent reproducibility between these two different approaches. Overall, biological and technical replicates had excellent Pearson correlation coefficients across multiple metrics. The conclusions that we are making are thus based on many more than an n of 2 samples; we observe spatial segregation of different T_{RM} subsets across 10 samples in our Xenium time course, our Merscope samples, VisiumHD samples, and human tissues. Thus, we are confident that our data are representative of the biology and that our study upholds the highest standards in scientific rigor.

Referee #4 (Remarks to the Author):

The authors have done a great job addressing my comments. They not only added confirmatory data but also expanded their study substantially. I am satisfied with the state of the manuscript.

A small point: perhaps the title should be adjusted to "Tissue-resident memory CD8 T Cell Diversity is Spatiotemporally Imprinted"

Response: We would like to thank the reviewer for their particularly encouraging feedback, and we have modified the title of the manuscript as per the reviewer's suggestion.

Referee #5 (Remarks to the Author):

This very interesting manuscript describes the dynamic zonation of CD8 T cells along the villus axis following infection. By combining spatial transcriptomics, elegant data analyses and several knock-out strategies the authors identify TGFb-related signals that are responsible for homing of distinct Trm subsets to the villus bottom and top. They also decipher the signaling molecules and central ligands responsible for the differential programming of bottom and tip T cells. The approaches are original, and the findings are intriguing and thought-provoking. The authors have done a tremendous amount of work to strengthen the paper during the revision process. I have a few comments that could improve the work:

Response: We would like to thank the reviewer for their time in assessing both our original and revised versions and the very encouraging positive feedback to strengthen our manuscript further.

The authors present ample zoned signatures largely based on Xenium and VisiumHD. It would be important to add at least some QCs and comparisons to zonation patterns previously observed in the mouse epithelium along the crypt-villus axis ([https://www.cell.com/cell/fulltext/S0092-8674\(18\)31164-4](https://www.cell.com/cell/fulltext/S0092-8674(18)31164-4)) and along the duodenum-ileum axis (<https://www.nature.com/articles/s41556-023-01337-z>). It would also be important to compare the profiles between the xenium experiments and the respective genes in the VisiumHD experiment. The authors can correlate the tip/bottom ratios in expression of the genes that appear in both the VisiumHD and Xenium panel.

We thank the reviewer for this suggestion. We have now mapped the published gene signatures defining the zones of the mouse epithelium on our dataset. The gene signatures from *Moor et al.*, defining the crypt-to-villus axis and the gene signature from *Zwick et al.*, defining zones along the duodenum-ileum axis, map perfectly on our datasets, thus confirming previously published results and further strengthening the validity of our datasets. We have added this important data as new **Extended Data Figures 3 c and d** (also **Reviewer Figure 3**).

As per the reviewer's suggestion, we have also correlated the top/bottom ratios in gene expression between Xenium and VisiumHD. The analysis revealed a strong correlation between the Xenium and VisumHD experiments, thus further supporting the identified zones and validating the results obtained using the Xenium and VisiumHD platforms. We have added this important additional QC figure as new **Extended Data Figure 4c** (and also **Reviewer Figure 4**).

Reviewer Figure 3: The gene signatures from Moor et al. and Zwick et al., defining zones of epithelial cells along the crypt-to-villus axis or along the longitudinal axis respectively, were mapped on epithelial cells in our datasets. Both signatures can clearly identify the different zones of epithelial cells along both axes.

Reviewer Figure 4: Pearson correlation of epithelial cell gene expression at the top of the villus divided by expression at the bottom of the villus. These ratios were compared for overlapping genes between Xenium and VisiumHD (both day 8 .p.i.).

There is some unclarity regarding the intestinal segments analyzed, the methods describe Swiss rolls prepared from 10cms of proximal small intestine, after removal of 3cms. This means the authors analyze duodenum and proximal jejunum. The human validation is performed on ileal biopsies. The authors should be clear regarding the segments analyze and discuss potential confounders (e.g. the possibility that the mouse ileum could exhibit different patterns than those shown).

Response: We acknowledge that the classic separations of the small intestine as duodenum, jejunum and ileum can be macroscopically challenging, hence the relevance of using unbiased approaches to delineate the intestinal anatomy, as elegantly done by Harnik et al 2024 Nature, Zwick et al 2024 Nature Cell Bio., and Moor et al 2018 Cell, and others. Indeed, we have profiled the duodenum and proximal jejunum. We have clarified this point in the manuscript.

Line 190-192 – The uninfected intestines are claimed to also show zonation of *Gzma*, *Gzmb* and *Itgae*, however the shown results suggest otherwise

(Figure 3g clearly shows the tip zonation of *Gzma*, *Gzmb* and *Itgae* and the bottom zonation of *Tcf7* in the post-infection mouse intestines, whereas *EDF5e* does not show the same trends in the uninfected intestines. What are the patterns of these genes in the VisiumHD experiment?

Response: To better show the polarization of CD8 T cells along the crypt-villus axis in the uninfected samples and its similarity to the phenotypes observed in our time course in mice and in human samples, we now provide scVelo heatmap plots highlighting key genes, such as *Slamf6*, *Tcf7*, *Gzma*, *Gzmb* and *Itgae*) expressed by CD8 $\alpha\beta$ T cells along the crypt-to-villus axis. This new figure has been included as new **Extended Data Figure 5f** (**Reviewer Figure 5**). Please note that some minor differences could also exist due to the fact that our analysis of the uninfected SI focuses on total endogenous CD8 $\alpha\beta$ T cells (polyclonal uncoordinated cellular estates) vs antigen-specific P14 CD8 T cells in our time course (responding to a common antigen in LCMV).

The authors may also want to validate on another published mouse VisiumHD dataset (<https://www.10xgenomics.com/datasets/visium-hd-cytassist-gene-expression-libraries-of-mouse-intestine>).

new ED5f

Reviewer Figure 5: Convolved gene expression of CD8 $\alpha\beta$ T cells along the crypt-villus axis in the uninfected SI (n=2 biological duplicates combined)

Response: We are aware of the publicly available dataset deposited by 10X Genomics. Because we cannot ascertain the conditions in which the small intestine was collected, or the conditions in which the mice were kept, or fed, or any other protocol or experimental details potentially impacting immune cell activities (as this is dataset is not part of a peer-reviewed publication), we believe these data does not meet the rigor standards for inclusion in Nature. Hence, we decided to generate our own dataset.

The analysis of differences in localization and expression in the TGF β 2-KO cells uses 8 differentially expressed genes between the KO and WT to define a TGF β signature. Why were these 8 genes chosen, and how robust is this analysis to the gene set choice? The robustness of this analysis can be assessed by using different numbers of genes, and explicitly using clear criteria of DGE including genes with defined fold-change and q-value.

Response: We thank the reviewer for this important consideration, and we agree that robustness is a key necessity for these kinds of analysis. We had initially chosen the top 4 genes, that were up- or downregulated in the differential expression analysis between TGF β KO and WT cells. To increase the robustness of the analysis, we generated another gene list, using an approach as suggested by the reviewer ($|\log_2FC| > 1.0$, q-value < 0.01). This resulted in a list of 20 differentially expressed genes. Using this gene set for the correlation analysis mirrored almost perfectly the analysis shown in Figure 4g (see

Reviewer Figure 6). We are therefore confident, that the analysis shown in Figure 4 is robust to the gene set choice.

Reviewer Figure 6: Repeat of the analysis shown in Figure 4g using a gene list that was generated based on log2-foldchange and q-value cutoffs. The results of this analysis mirror the results shown in Figure 4g, which was generated using the top 4 down- or up-regulated genes.

The authors denote three enterocyte cell types (enterocyte 1,2,3), what are their gene expression markers and how do they relate to the zoned enterocyte populations described in [https://www.cell.com/cell/fulltext/S0092-8674\(18\)31164-4](https://www.cell.com/cell/fulltext/S0092-8674(18)31164-4)? Defining these according to their location along the villus axis (tip,middle,bottom) might increase clarity in some of the figures.

Response: Our study focuses on the spatial phenotypic differences in intestinal immune cells. We would like to point out that while our 350 and 480 gene panels were designed to characterize all intestinal cellular populations, they were optimized to maximize the profiling of immune cells and their activation states. Our data are able to identify previously published spatial heterogeneity of epithelial cells (new **Extended Data Figures 3c and 3d** and **Reviewer Figures 3 and 4**) and we see, that “enterocytes 1” are predominantly found at the top of the villus,

Reviewer Figure 7: (a) Distribution of epithelial cells at day 9.0 p.i. along the crypt-to-villus axis and the longitudinal axis. “Enterocytes 1” are predominantly found at the top of the villus, whereas “enterocytes 2 & 3” are located towards the crypt and separate by the longitudinal axis. (b) Marker genes for the enterocyte cluster.

whereas “enterocytes 2 & 3” are located towards the bottom of the villus and separated by the longitudinal axis (**Reviewer Figure 7**). Since enterocytes are, however, not the main focus of our study, and other laboratories have done more comprehensive studies on enterocytes, we have refrained from annotating these cells in a more detailed level.

- Line 86-87 – provide details in Methods on the deep-learning prioritization.

Response: Added to Gene panel design for probe-based spatial transcriptomics profiling of mouse small intestine methods: “Gene panel design made use of PERSIST¹⁰, a deep learning model that uses scRNA-seq data to learn a binary mask for identification of a subset of genes that best predict cell type from gene expression (supervised), or reconstruction of the whole transcriptome gene expression (unsupervised).”

- Refs 15 and 23 are duplicates.

Response: We apologize for this oversight, and we have resolved this issue in the new submission.

- EDF6 - add specific references to Milner et al and Nath et al.

Response: We added references to Extended Data Figure 6.

The authors need to ensure all data is deposited upon publication (the current data for review did not include the Visium HD experiments). Uploading Loupe files would facilitate easier exploration.

Response: We will ensure that all data generated and used in this study, as well as the code to analyze the data, is deposited before publications.